



# Atmospheric observations with E-band microwave links – challenges and opportunities

Martin Fencl[1], Michal Dohnal[1], Pavel Valtr[2], Martin Grabner[3], Vojtěch Bareš[1]

[1]Department of Hydraulics and Hydrology, Czech Technical University in Prague, Prague 6, 166 29, Czech Republic

[2]Department of Electromagnetic Field, Czech Technical University in Prague, Prague 6, 166 29, Czech Republic

[3]Department of Frequency Engineering, Czech Metrology Institute, Brno, 638 00, Czech Republic

*Correspondence to*: Martin Fencl (martin.fencl@cvut.cz)

**Abstract.** Opportunistic sensing of rainfall and water vapor using commercial microwave links operated within cellular

networks was conceived more than a decade ago. It has since been further investigated in numerous studies predominantly concentrating on the frequency region of 15–40 GHz. This manuscript provides the first evaluation of rainfall and water vapor sensing with microwave links operating at an E-band (specifically, 71–76 GHz and 81–86 GHz), which are increasingly updating, and frequently replacing, older communication infrastructure. Attenuation-rainfall relations are investigated theoretically on drop size distribution data. Furthermore, quantitative rainfall estimates from six microwave links, operated

within cellular backhaul, are compared with observed rainfall intensities. Finally, the capability to detect water vapor is demonstrated on the longest microwave link measuring 4.86 km in path length. The results show that E-band microwave links are by one order of magnitude more sensitive to rainfall than devices operating in the 15–40 GHz range and are thus able to observe even light rainfalls, a feat practically impossible to achieve previously. The E-band links are, however, substantially more affected by errors related to variable drop size distribution. Water vapor retrieval might be possible from long E-band

microwave links, nevertheless, the efficient separation of gaseous attenuation from other signal losses will be challenging in practice.

## 1 Introduction

Electromagnetic (EM) waves in the microwave region are attenuated by water vapor, oxygen, fog, or raindrops. Measurements of microwave attenuation at different frequency bands thus represent an invaluable source of information regarding the

atmosphere. Passive and active microwave systems have become an integral part of Earth observing satellites, terrestrial remote sensing systems, and complete remote sensing methods in other spectral regions (Woodhouse, 2017). The microwave region is, however, also increasingly utilized in communication systems allowing for new possibilities to observe atmosphere with unintentional (opportunistic) sensing. Commercial microwave links (CMLs) are an excellent example of a communication system capable of providing close-to-ground observations of the atmosphere. CMLs are point-to-point line-of-sight radio

connections widely used in mobile phone backhaul for connecting hops of different lengths typically ranging from tens of



meters to several kilometers. There were about 4 million CMLs operated worldwide within cellular backhaul in 2016 (Ericsson, 2016) and about 5 million in 2018 (Ericsson, 2018). Most of these CMLs operate at frequencies between 15 and 40 GHz (Ericsson, 2016, 2018) where raindrops and, to a lesser extent, water vapor represent a significant source of attenuation (Atlas and Ulbrich, 1977; Liebe et al., 1993). Information on the attenuation of any CML within (countrywide) networks is virtually

accessible in real-time with a delay of several seconds from a remote location, typically a network operation center creating an appealing opportunistic sensing system capable of providing close-to-ground observations of rainfall intensity (Leijnse et al., 2007; Messer et al., 2006) and water vapor density (David et al., 2009).

CML rainfall retrieval methods developed over the last decade have been predominantly designed and tested for frequency bands between 15 and 40 GHz (Chwala and Kunstmann, 2019). Attenuation caused by raindrops is, in this frequency region,

almost linearly related to rainfall intensity and does not strongly depend on drop size distribution (DSD) (Berne and Uijlenhoet, 2007). Water vapor retrieval has been proposed for CMLs operating around 22 GHz (David et al., 2009) where there is a resonance line of water vapor. Increasing demands on data transfers force operators to utilize higher frequency spectra and a new generation of E-band CMLs, operating at the 71 - 86 GHz frequency band, is gradually modernizing cellular backhaul networks, especially in cities where they often replace older devices. The share of E-band CMLs in mobile phone backhaul

has already reached 20%, *e.g.,* in Poland and the Czech Republic, and it is expected to grow in other countries as E-band CMLs are considered an essential part of new 5G networks (Ericsson, 2019).

E-band CMLs should be, according to recommendations for designing CMLs (ITU-R P.838-3, 2005), more sensitive to rainfall, nevertheless, the relation between rainfall intensity and attenuation is not linear. Furthermore, E-band radio waves have two to four time's shorter wave lengths and the scattering efficiency (resonance peak) is highest for smaller raindrops.

The attenuation-rainfall relation is, thus, more sensitive to drop size distribution, which has been already demonstrated in several propagation experiments, *e.g.,* (Hansryd et al., 2010; Luini et al., 2018). Radiowave propagation at an E-band is also more sensitive to water vapor which poses a challenge when separating rainfall-induced attenuation from other sources of attenuation. On the other hand, the sensitivity to water vapor might also enable its detection or even monitoring.

This manuscript provides the first evaluation of E-band CMLs as rainfall and water vapor sensors. The capabilities of E-band

CMLs for weather monitoring are theoretically evaluated and demonstrated on attenuation data retrieved between August and December 2018 from a six E-band CML operated within cellular backhaul of a commercial provider in Prague (T-Mobile, CZ). The ultimate goal of this investigation is to provide an overview of the challenges and opportunities related to atmospheric observations with E-band CMLs. Section 2 of the manuscript summarizes the principles behind retrieving atmospheric variables from CML observations, Section 3 describes the methodology and datasets used for the CML assessment, Section 4

presents the results of the case study and evaluates the effect of DSD on the attenuation-rainfall relation. The results are further interpreted and discussed in Section 5 followed by the conclusions which are presented in Section 6.



## 2. Retrieving atmospheric variables from CMLs

### 2.1 Components of total observed loss

Standard CMLs are monitored for transmitted ($tx$) and received ($rx$) signal power and the difference between $tx$ and $rx$ is the
total observed loss ($L_t$) which can be separated into several components:

$$L_t = L_{bf} + L_m + L_{tc} + L_{rc} - G_t - G_r, \tag{1}$$

where $L_{bf}$ is free space loss, $L_m$ are losses in the medium, $L_{tc}$ and $L_{rc}$ are losses at the transmitting and receiving antennas, and
$G_t$ and $G_r$ are antenna directive gains (Internationale Fernmelde-Union, 2009). Free space loss is uniquely defined by the
distance between the transmitter and receiver, and by wavelength. The sum of antenna losses and gains is given by their
hardware and includes interference with the environment close to antennas as antenna loss can change, *e.g.*, due to the wetness
of antenna radomes. The propagation mechanisms influencing loss in the medium ($L_m$) consist of attenuation due to
atmospheric gases including water vapor, attenuation due to precipitation, attenuation due to obstacles in the wave path, and
diffraction losses causing bending of the direct wave towards the ground.

The separation of attenuation due to rainfall, resp. water vapor from other sources of attenuation, is possible to some extent,
but, firstly, dry and wet weather periods need to be identified (Overeem et al., 2011; Schleiss and Berne, 2010). Attenuation
during dry weather is assumed to be a baseline, and the difference between dry and wet weather attenuation is then attributed
to rainfall. Fluctuations in the baseline during dry weather can be attributed to water vapor, nevertheless, they can also be
caused by temperature changes, hardware instability, etc.

The correct estimation of raindrop path attenuation also requires the separation of additional attenuation caused by antenna
radome wetting, so-called wet antenna attenuation (WAA). This is especially important for shorter CMLs which are attenuated
by raindrops along the short path and the relative importance of WAA contribution is significant.

### 2.2 Wet antenna attenuation

Wet antenna attenuation (WAA) is caused by a water layer forming on antenna radomes during rainfall events or dew
occurrence. Modelling WAA is challenging as the formation of water film on antennas is a complex process dependent on
rainfall intensity, wind direction and velocity, or air and rain temperature, as well as on antenna radome hydrophobic properties.
On the other hand, WAA represents a substantial part of total attenuation (Fencl et al., 2019), especially by shorter CMLs, and
its identification and separation from attenuation caused by raindrops along a CML path is crucial when obtaining reliable
rainfall estimates.

Most of the models specifically suggested for microwave link rainfall retrieval are empirical and designed for lower frequencies
(Minda and Nakamura, 2005; Overeem et al., 2011; Schleiss et al., 2013). However, the semi-empirical model suggested by
Leijnse et al. (2008) enables WAA for an arbitrary frequency to be calculated. The Leijnse model assumes a layer of water
with constant thickness which is then assumed to be power-law related to rainfall intensity. The parameters of this relation



need to be optimized. According to the Leijnse model, WAA typical of E-band CML frequencies is about two times higher than for 38 GHz. Hong et al. (2017), however, showed on 72 and 84 GHz microwave links that WAA depends highly on specific hardware settings. An antenna without radome experienced WAA of about 7 dB during a spraying experiment with artificial rain. The antenna covered by a radome (which is a typical setting for CMLs) experienced WAA of approx. 2 dB and WAA decreased further to only approx. 0.3 dB when a radome with hydrophobic coating was used. Similarly low values of WAA at E-band CMLs have been reported by Ostrometzky et al. (2018) who observed WAAs of 0.86 +- 0.54 dB and 1.07 +- 0.75 dB at two 73 GHz CMLs. These values are significantly lower than previously observed WAA at lower frequencies (Fencl et al., 2018; Minda and Nakamura, 2005; Schleiss et al., 2013), although E-band CMLs should be, in theory, more sensitive to WAA (Leijnse et al., 2008). More extensive investigations are, therefore, needed to better understand how WAA affects E-band CMLs.

**2.3 Relation between raindrop path attenuation and rainfall intensity**

Attenuation of a direct EM wave due to raindrops can be precisely calculated using the scattering theory. Attenuation caused by a single raindrop is determined by the wavelength, refractive index of water, and shape parameters of the raindrop. The extinction cross section $C_{ext}$, which can be calculated using the T-matrix method (Mishchenko and Travis, 1998), characterizes the scattering and absorption properties of each raindrop for a given frequency and polarization. The number of drops in the unit volume per drop diameter interval $N(D)$ (m$^{-3}$ mm$^{-1}$) is relatively small for natural showers. Therefore, the contribution of scattered secondary EM waves radiated from particles to the incident field of the other particles is negligible. The specific raindrop path attenuation $k$ (dB km$^{-1}$) can be thus considered as a sum of attenuations caused by single raindrops of diameter $D$ and can be expressed in integral form:

$$k(f) = 4.343 \times 10^3 \int_{D_{min}}^{D_{max}} C_{ext}(D, f) N(D)\, dD, \tag{2}$$

The $N(D)$ also determines rainfall intensity $R$ (mm h$^{-1}$):

$$R = 0.6\, \pi\, 10^{-3} \int_{D_{min}}^{D_{max}} v(D)\, D^3 N(D)\, dD, \tag{3}$$

where $v(D)$ is the terminal velocity of raindrops given by their diameters. As both specific attenuation and rainfall intensity are moments of drop size distribution (DSD) the relation between attenuation and rainfall intensity can be approximated by a power-law:

$$k = a\, R^b, \tag{4}$$

where $a$ (mm$^{-b}$ h dB km$^{-b}$) and $b$ (-) are empirical parameters dependent on frequency, polarization, and DSD. When estimating rainfall from observed attenuation, Eq. 4 can be reformulated to:

$$R = \alpha\, k^\beta, \tag{5}$$





where $\alpha$ (mm h$^{-\beta}$ dB$^{-\beta}$ km) $= a^{-1/b}$ and $\beta$ (-) $= b^{-1}$.

The model (4) resp. (5) approximates the attenuation-rainfall relation well at frequencies around 30 GHz, nonetheless, errors increase for both lower and higher frequencies due to variable DSD (Berne and Uijlenhoet, 2007). Berne and Uijlenhoet

(2007), however, investigated sensitivity to DSD only for the frequency region of 5-50 GHz. A detailed evaluation of the attenuation-rainfall model for the E-band has not been reported, yet higher sensitivity of E-band CMLs to DSD has been demonstrated during several propagation experiments (Hansryd et al., 2010; Luini et al., 2018). Furthermore, different sensitivities of E-band CMLs to DSD, compared to lower frequencies, is also apparent from their scattering efficiency ($Q_{ext}$):

$$Q_{ext} = \frac{C_{ext}}{\sigma_{geo}}, \tag{6}$$

where $C_{ext}$ and $\sigma_{geo}$ are extinction resp. geometric cross-sections. The scattering efficiency (Fig. 1) of EM waves at the E-band is the highest for smaller raindrops, which is characteristic for stratiform rainfalls, whereas larger raindrops characteristic for convective rainfalls contribute relatively less to total attenuation.

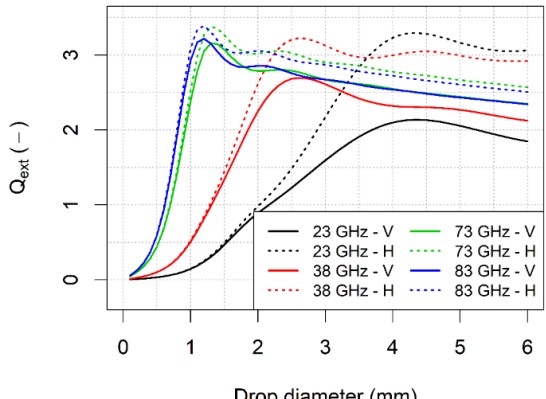

**Figure 1: Scattering efficiency of plane waves at different frequencies and polarizations (H – horizontal, V – vertical).**

**2.4 Attenuation by atmospheric gasses**

Attenuation by atmospheric gasses is caused predominantly by the interaction of an EM wave with molecules of water and oxygen. The evaluation of gas attenuation, as described in (ITU-R P.676-11, 2016) and originally in (Liebe et al., 1993), is based on the concept of the complex refractive index. In a medium with complex refractive index $n$, the intensity of EM wave $I$ (Wm$^{-2}$) is attenuated at distance $x$ (m) as:

$$I(x) = I(0)\, exp(-2\kappa\, Im(n)\, x), \tag{7}$$

where $\kappa = 2\pi f / c$ is a vacuum wave number (m), $c$ (m s$^{-1}$) speed of light and $Im(n)$ denotes the imaginary part of $n$. After introducing complex refractivity $N = (n-1)10^6$, the specific attenuation $k$ (dBkm$^{-1}$) is obtained as:



$$k = 10 \, log_{10} \left( \frac{I(0)}{I(1)} \right) = 0.1819 \, f \, Im(N), \tag{8}$$

where $f$ (GHz) is the EM wave frequency.

In the model (Liebe et al., 1993), total complex refractivity $N_t$ is divided into dispersive and nondispersive parts: $N_t = N_0 + N(f)$. Nondispersive part $N_0$ is real and does not contribute to attenuation. The dispersive part is given as $N(f) = N_L + N_d + N_c$, where $N_L$ is the contribution of spectral lines of oxygen and water, $N_d$ is from the dry air non-resonant spectrum and $N_c$ is from the water vapor continuum. The key term $N_L$ is determined as a sum of 44 and 35 spectral lines of oxygen and water respectively. The shape and amplitude of the spectral lines depend on temperature, water vapor content and atmospheric pressure.

Figure 2 shows specific attenuation by water vapor and dry air. Attenuation due to water vapor increases as the frequency increases, with the exception of the peak around 22 GHz and the depression around 60 GHz (Fig. 2a). The sensitivity of water vapor attenuation to temperature and air pressure monotonically increases as the frequency increases. The temperature and pressure also influence dry-air attenuation, especially at frequencies around 60 GHz (Fig. 2b).

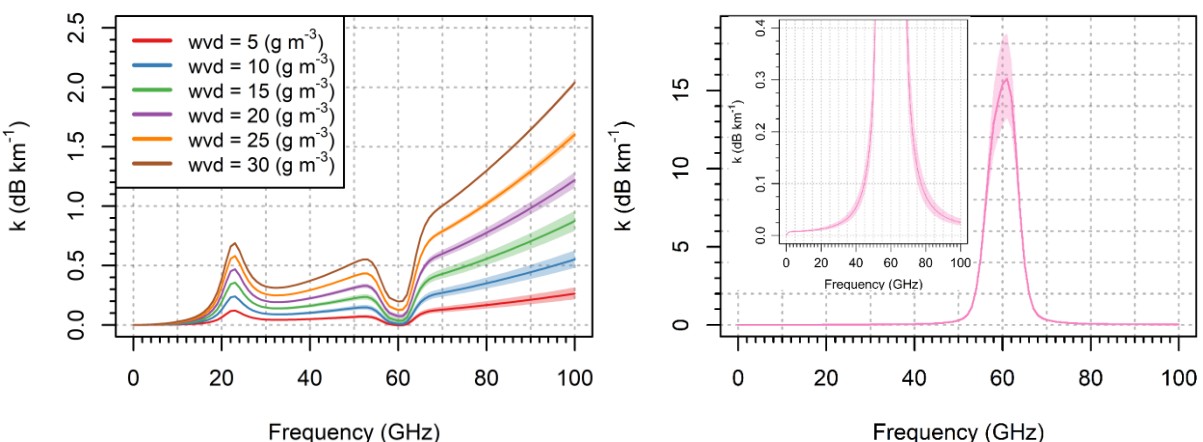

**Figure 2: (a) Attenuation of EM by water vapor for frequencies 0 to 100 GHz. The relation is shown for different water vapor densities (wvd) for temperatures -10° to +30° C and pressure 1000 to 1030 hPa (colored bands). (b) Dry air attenuation of EM for temperatures -10° to +30° C and pressure 1000 to 1030 hPa (colored band), inset: a detailed view of the lower attenuations.**

### 3. Material and Methods

The E-band evaluation concentrates on i) the relation between raindrop attenuation and rainfall intensity, ii) processing routines

for separating different attenuation components, and iii) gaseous attenuation and its relation to water vapor density and air temperature. The methodology combines numerical experiments using virtual attenuation time series simulated from weather observations with analyses of CML observations obtained during the dedicated case study.



### 3.1 Experimental sites and instrumentation

Drop size distribution is obtained from a Parsivel disdrometer (1st generation, manufactured by OTT). Drop counts and fall
velocities are recorded over 30 s intervals. The data was collected during the CoMMon experiment in Duebendorf (CH) at site
2, which is described, *e.g.*, in (Wang et al., 2012). We further refer to this dataset as Duebendorf data.

The E-band evaluation case study is performed on six CMLs (Table 1) located in the south-east suburb of Prague (CZ). Five
shorter CMLs are located in a residential area with a housing estate. The path of the long CML goes over an area with mixed
land use, mostly agricultural. The main node from which all CML paths originate is located on the roof of a 65-m-tall building;
the end nodes are about 15 m to 30 m above ground (Fig. 3). All the CMLs operate at an Ericsson MINILINK platform. CMLs
have full-duplex configuration with two sub-links operating in one direction at 73–74 GHz and in the second direction at 83–
84 GHz with a duplex separation of 10 GHz. Transmitted signal power (*tx*) and received signal power (*rx*) are collected with
custom-made server-sided software which polls selected CMLs using SNMP protocol and stores records into a PostgreSQL
database. The sampling time step is approx. 10 s. The resolution of a *tx* and *rx* reading is 0.1 dB. All devices have automatic
transmitted power control (ATPC), *i.e.*, transmitted power is automatically controlled to minimize fluctuations in *rx*. CML
data acquisition is described in detail in the *Supplementary material*.

Tipping bucket rain gauges (MR3, METEOSERVIS v.o.s., catch area 500 cm$^2$, resolution 0.1 mm) have been deployed at four
measuring sites. Two rain gauges are located at the end nodes of the long CML, one at ground level close to the CML path
about 1.5 km from the main network node, and one about 2 km south-west from the main node. The rain gauges at sites 1 and 2
are equipped with temperature and air humidity sensors collecting observations in a 5-min time step. To minimize instrumental
errors, all four rain gauges have been regularly maintained (on a monthly basis) and are dynamically calibrated (Humphrey et
al., 1997). We further refer to the case study dataset as Prague data (Fencl et al., 2020).

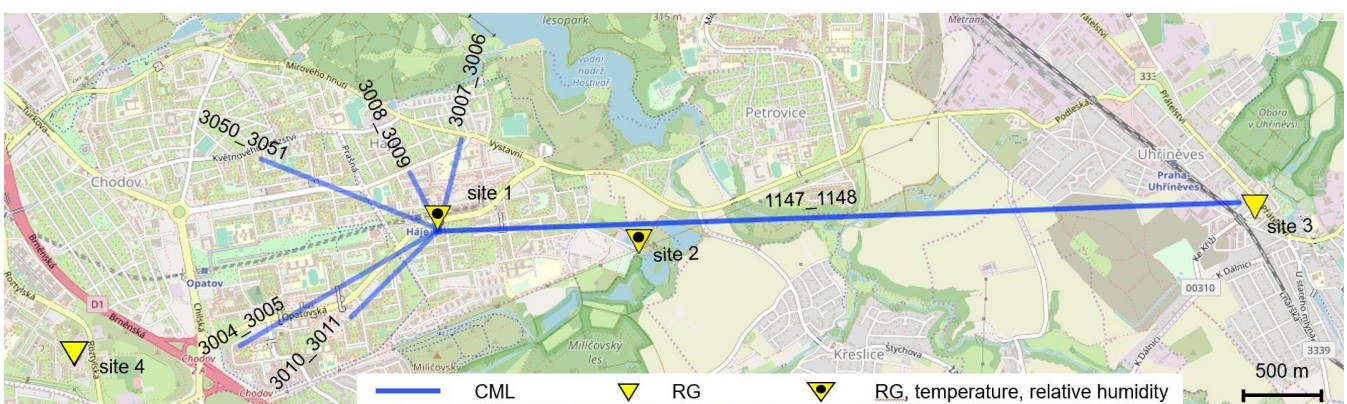

**Figure 3: Case study area Prague-Haje. Two of the rain gauges are equipped with air humidity and temperature probes.**
**© OpenStreetMap contributors 2020. Distributed under a Creative Commons BY-SA License.**



**Table 1: CML characteristics in Prague-Haje. The suffixes A and B denote first, resp. second end node, and suffixes AB and BA the direction from A to B resp. B to A.**

| ID | LonA (deg) | LatA (deg) | LonB (deg) | LatB (deg) | FreqAB (GHz) | FreqBA (GHz) | PolAB | PolBA | Length (m) |
|---|---|---|---|---|---|---|---|---|---|
| 1147_1148 | 14.529 | 50.0301 | 14.597 | 50.0317 | 73.5 | 83.5 | V | V | 4866 |
| 3007_3006 | 14.531 | 50.0352 | 14.5291 | 50.0302 | 73 | 83 | V | V | 573 |
| 3008_3009 | 14.5291 | 50.0302 | 14.5266 | 50.0333 | 73.25 | 83.25 | H | H | 389 |
| 3050_3051 | 14.5291 | 50.0302 | 14.514 | 50.0341 | 72.75 | 82.75 | V | V | 1164 |
| 3004_3005 | 14.5291 | 50.0302 | 14.5122 | 50.0237 | 73.75 | 83.75 | V | V | 1409 |
| 3010_3011 | 14.5291 | 50.0302 | 14.5216 | 50.0253 | 74.25 | 84.25 | H | H | 765 |


### 3.2 Experimental data

**Experimental periods:** Duebendorf data span from March 2011 to April 2012. Prague data span from 20[th] August to 16[th] December 2018. The rainfall observations are, due to technical problems available from 28[th] October to 16[th] December 2018. The periods for evaluating rainfall retrieval and for evaluating the effect of humidity and temperature fluctuations on gaseous

attenuation are, therefore, different.

**Duebendorf data:** The disdrometer data is quality-checked and suspicious records are excluded using filters described in (Jaffrain and Berne, 2010). Moreover, only records classified by the disdrometer as rainfall (at least from 90 %) are used for further analysis; hail events are excluded. The data which pass the quality check are aggregated to a 1-min temporal resolution.

**Prague data:** Total loss is calculated for each CML as the difference between transmitted and received signal powers. The

total loss data is aggregated to a 1-min temporal resolution.

Rain-gauge data are separated into rainfall events. An event is defined as a period with intervals between consecutive rain gauge tips shorter than one hour. The rainfall events with rainfall height lower than 1 mm are excluded from the evaluation. Furthermore, events during which the temperature dropped below 2° C were also excluded from the evaluation to limit the performance assessment to liquid precipitation only. This results in a set of five events (see Table 2) representing, in terms of

total depth, 81 % of all the precipitation during the experimental period. Rainfall data are aggregated to a 15-min temporal resolution to limit uncertainties due to rain gauge quantization and uncertainties related to uncaptured rainfall spatial variability. The 15-minute rainfall intensities are, for all four rain gauges, highly correlated (r = 0.88–0.96). The cumulative rainfall observed by the rain gauges is also in very good agreement and differs from the mean rainfall only by 1–3 %.

The air temperature and relative humidity data (5-min temporal resolution) is not further processed. The correlation between

temperature observations is 0.95 and between humidity observations 0.86. In general, observations on the roof of the 70-m-tall building (site 1) have slightly lower variability than close-to-ground observations (site 2). The discrepancies are especially pronounced in the morning hours.



**Table 2: Rainfall events used for the evaluation of CML rainfall retrieval**

| Event start | Duration | Height | $R_{max}$ |
|---|---|---|---|
| | (min) | (mm) | (mm h$^{-1}$) |
| 2018-10-28 01:10 | 1218 | 21.0 | 4.4 |
| 2018-11-02 19:14 | 500 | 5.1 | 2.5 |
| 2018-11-24 09:46 | 176 | 1.9 | 1.7 |
| 2018-12-03 05:00 | 158 | 1.8 | 2.6 |
| 2018-12-03 22:03 | 210 | 4.9 | 3.0 |

**Table 3: Empirical parameters of convective and stratiform rainfall for DSD reconstruction as observed by Fujiwara (1965) and re-parameterized by Ulbrich, (1983).**

| Type | $N_0$ (m$^{-3}$ cm$^{-1-\mu}$) | $\mu$ (-) | $\varepsilon$ (h$^{-\delta}$) | $\delta$ (-) |
|---|---|---|---|---|
| Convective (thunderstorm) | 7.05 10$^4$ | 0.4 | 0.118 | 0.20 |
| Widespread or stratiform | 1.96 10$^5$ | 0.18 | 0.082 | 0.21 |

**3.3 Sensitivity of the k-R model to drop size distribution**

The analysis of the k-R model (5) with respect to DSD is based on fitting Eq. 3 on attenuation and rainfall intensities obtained from Eq. 1 and 2. First, the sensitivity of the k-R model parameters to the type of rainfall (stratiform vs. convective) is investigated on theoretical DSD and on DSD from the Duebendorf dataset.

**Investigation of theoretical DSD:** The number of drops $N$ with diameter $D$ is modeled using the gamma distribution function

scaled to rainfall intensity (Ulbrich, 1983). The parameters of gamma distribution for stratiform and convective rainfall are taken from (Ulbrich, 1983). The empirical parameters needed for the reconstruction of $N(D)$ for stratiform and convective rainfall are in Table 3. The distribution functions for two different rainfall intensities are shown in Fig. 4.

$N(D)$ is calculated for a sequence of reference rainfall intensities from 0 to 50 mm h$^{-1}$ with an increment of 0.1 mm h$^{-1}$ for both types of rainfalls. Specific attenuations corresponding to a given intensity and rainfall type are then calculated according to

Eq. (2). Specific attenuations are calculated for 73.5 and 83.5 GHz, vertical polarization, *i.e.*, frequencies of the sub-links belonging to the long CML in the Prague data. These frequencies are approximately in the middle of the frequency bands of 71–76 GHz and 81–86 GHz allocated for E-band fixed wireless services and, thus, representative for all E-band CMLs.

The k-R model (5) is fitted separately for each frequency and rainfall type by minimizing the sum of squared residuals between reference rainfall intensities and rainfall intensities estimated by the model using a specific attenuation obtained by Eq. (2).

**Investigation of Duebendorf data:** The effect of DSD on the k-R power-law approximation is further tested on DSD data (Deubendorf) and its influence on rainfall estimation accuracy is quantified. The procedure is analogous to the analysis of the theoretical DSD. However, rainfall intensities and specific attenuations are calculated from the observed DSD. The rainfall





records are classified into two types according to the mass-weighted drop diameter $D_m$ (m), which is the ratio between the fourth and third DSD moments:

$$D_m = \frac{\int_{D_{min}}^{D_{max}} N(D)D^4 dD}{\int_{D_{min}}^{D_{max}} N(D)D^3 dD} \quad , \quad (9)$$

The mass-weighted diameter $D_m$ is a common descriptor of the center of a probability density function $f(D)$ characterizing DSD, thus the $D_m$-based classification resembles the classification on convective and stratiform rainfalls (Jaffrain and Berne, 2012). The mass-weighted diameter can be approximately related to the rainfall intensity by a power-law function:

$$\widehat{D_m} = c\, R^d, \quad (10)$$

where $R$ (mm h$^{-1}$) is rainfall intensity and $c$ (h$^{-d}$ mm$^{1/d}$) and $d$ (-) are empirical parameters. Such an approximation results in perfect fits for stratiform and convective rainfall types when applied to theoretical DSD (Fig. 4b). The approximation (10) is, therefore, used to calculate rainfall intensity dependent on the threshold for classifying rainfalls as convective or stratiform. Parameters $c$ and $d$ are estimated by fitting Eq. (10) to $D_m$ as derived from real disdrometer data usage by Eq. (9).

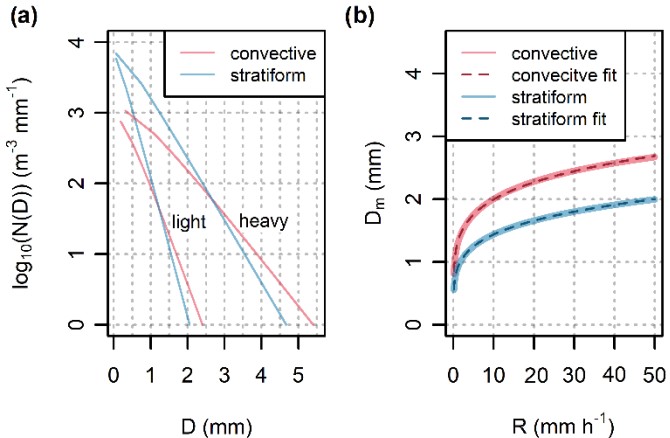

**Figure 4: (a) Theoretical DSD for light (1 mm h$^{-1}$) and heavy (50 mm h$^{-1}$) convective and stratiform rainfall, and (b) the power-law relation between mass weighted diameter $D_m$ and rainfall intensity for the same rainfall types. Gamma distribution functions with parameters corresponding to storms reported by (Fujiwara, 1965) are used.**

### 3.4 CML rainfall retrieval
Rainfall retrieval is performed for each sub-link separately. First, total observed loss aggregated to a 1-min resolution is
checked for hardware related artifacts. Second, total observed loss is aggregated to a 15-min temporal resolution and the baseline is identified and separated. Third, WAA is estimated and, finally, attenuation corrected for WAA is converted to rainfall intensity. Although dry-wet weather classification is not used for rainfall retrieval in this study, it is included in the performance assessment as it might be needed for future studies and applications (see Discussion section).



**Quality check:** All the time series of total losses are visually inspected to identify obvious hardware related artifacts. In one
case (CML 3004_3005), the sudden change in the baseline is manually corrected, as automated procedures used for attenuation
processing are not designed to cope with this artifact.

**Dry-wet weather classification:** The classification is performed on quality-checked total observed losses using the algorithm
of (Schleiss and Berne, 2010) which is based on a moving window standard deviation. The window size is set to 15 minutes
and the threshold for classifying the record as wet ($\sigma_0$) is set to the 94% quantile of all standard deviations resulting from the
moving window filter. The 94 % probability corresponds approximately to the wet weather ratio in the Prague data as classified
by the rain gauges.

**Baseline identification:** Background attenuation, the so-called baseline, is needed to identify rainfall induced attenuation and
is estimated as a moving median with a centered window having a size of one week applied on time series of total losses
averaged over hourly intervals. A one-week window size seems to be appropriate for the climate of the Czech Republic as it
covers a period with more than half of the records belonging to dry weather. On the other hand, it is sufficiently short to capture
reliably long-term baseline drifts related to the instability of the CML hardware, or gaseous attenuation.

**Wet antenna attenuation:** WAA during rainfall is estimated by comparing attenuations as observed by sub-links of different
path lengths. WAA quantification assumes spatially uniform rainfall under which specific attenuations $k_1, k_2, ..., k_n$ of the sub-
links $1, 2, ..., n$ operating at the same frequency band in the same area should be identical:

$$k_1 = \frac{A_1 - Aw_1}{l_1} \approx k_2 = \frac{A_2 - Aw_2}{l_2} \approx \cdots \approx k_n = \frac{A_n - Aw_n}{l_n}, \tag{11}$$

where $A$ is rainfall-induced attenuations, *i.e.*, the difference between total observed loss ($L_t$) and the baseline, $Aw$ is wet antenna
attenuation and $l$ is CML (sub-link) length. Assuming correct baseline identification and the same $Aw$ for all CMLs, $Aw$ can
be directly quantified from any pair of sub-links of different lengths operating at the same frequency. The accuracy of the
quantification relies on the fulfillment of the assumptions and the difference between the sub-link lengths. The larger is the
length difference between the CMLs, the smaller is the effect of an inaccurate baseline identification or dissimilar $Aw$ within
the CML pair. On the other hand, the assumption of spatially uniform rainfall is unlikely to be valid for CMLs covering a large
area, *i.e.*, with contrasting lengths.

WAA is quantified at each time step by comparing the attenuation of the short CMLs to the attenuation of the long CML.
WAA after rainfall and during dew events is assumed to be equal to the total attenuation.

**Rainfall estimation:** Rainfall is estimated for each sub-link using the k-R power-law model (5) with ITU parameters and
parameters derived from DSD classified as stratiform rains, alternatively. The parameters for stratiform rainfalls are used for
its dominance in light and moderate autumn rainfalls in the Czech Republic. The specific attenuation $k$ (dB km$^{-1}$) used as an
input to the k-R model (5) is calculated:

$$k = max \left( \frac{L_t - B - Aw_{const}}{l}, 0 \right), \tag{12}$$





where $L_t$ (dB) is the total observed loss, $B$ (dB) is the baseline, $Aw_{const}$ is constant WAA, and $l$ (km) is the CML, resp. sub-link path length. The constant WAA is estimated separately for 73–74 GHz and 83–84 GHz sub-links as the median of WAA quantified according to Eq. (11).

### 3.5 Gaseous attenuation

The effect of temperature and air humidity on total CML attenuation is estimated theoretically from observed air temperature and relative humidity (see section 2.4) and compared to the real CML data obtained during the case study. Atmospheric pressure was not measured and is assumed to be constant at a level of 1013.25 hPa. Variations in atmospheric pressure have, however, an almost negligible effect on theoretical attenuation (the results of the sensitivity analysis are not shown here). The temperature and air humidity used in the analyses are averages from the observations at two locations along the CML path. Gaseous attenuation is estimated for the period from 20th August to 16th December 2018 and only considers dry weather, *i.e.,* periods without rainfall and dew occurrences (events causing the tipping of at least one of the rain gauges). A safety window of 6 h was set before and after each event with an event considered to start with the first tip of any rain gauge and ending with the last tip.

The theoretical attenuation derived from air temperature and relative humidity observations is then compared to the observed attenuation of the long CML. To enable a comparison, the observed attenuation is also aggregated to a 5-min time step corresponding to the time step of temperature and humidity observations, resp. theoretical attenuation. Furthermore, the constant baseline is subtracted from the observed attenuation time series. The constant baseline is set separately for each sub-link (73.5 GHz and 83.5 GHz), such as the median attenuation obtained after the baseline separation corresponds to the median theoretical attenuation. The median attenuation is calculated considering dry-weather periods only.

### 3.6 Performance evaluation

**Hardware related artifacts:** Hardware related artifacts are identified (visually) in the time series of total observed losses and described.

**Dry-wet weather classification:** Dry-wet weather classifiers obtained from CML sub-links are compared with each other and with classifiers obtained from rain gauge observations. Correlation is used here as a measure of similarity. The evaluation of dry-wet weather is performed on one-minute data because precise identification of the onset and ending of rainfall significantly affects baseline identification methods based on dry wet classification, as well as the quantification of wet antenna attenuation.

**Wet antenna attenuation:** Wet antenna attenuation during rainy periods is evaluated for shorter CMLs and related to rainfall intensity in terms of correlation. Wet antenna analysis is performed on attenuation data aggregated to 15 min.

**Rainfall retrieval:** CML rainfall retrieval performance is evaluated for two sets of k-R model parameters: parameters derived from ITU recommendations and parameters obtained from DSD observations (Duebendorf) classified as stratiform. The CML quantitative precipitation estimates (QPEs) of the long CML are compared to average 15-min rainfall from rain gauges rg_1, rg_2, and rg_3. The QPEs of the short CMLs are compared to average 15-min rainfall from rain gauges rg_1, rg_2, and rg_4. The quantitative evaluation focuses on the long CML, which is sufficiently long enough to capture even the light rainfalls dominating the Prague data. The performance of the short CMLs is shown to demonstrate limitations related to the improper





baseline and WAA identification which are more pronounced by shorter CMLs. The CML QPEs are evaluated over selected
rainfall events (Table 2) in terms of correlation, relative error in cumulative rainfall, and root mean square error (RMSE).

**Accuracy of the k-R power-law approximation**: The effect of DSD on the k-R power-law approximation is evaluated by comparing rainfall intensities obtained directly from DSD (3) to rainfall intensities estimated from the k-R model (5). Virtual specific attenuations derived from DSD (2) for two different frequencies are used as inputs to the k-R model.

The performance is evaluated for three model settings: i) the k-R model with a single set of parameters obtained by fitting the model to the whole dataset, ii) the k-R model with two sets of parameters for periods with stratiform resp. convective rainfall obtained by fitting the model separately for these two rainfall types, iii) the k-R model with parameters from ITU recommendations (ITU-R P.838-3, 2005).

The analysis is first performed for the theoretical DSD (theoretical pdfs describing drop size spectra) and secondly, in more detail, for DSD measured by a disdrometer. In the second analysis the k-R model is evaluated in terms of RMSE (root mean square error criterion) for the whole dataset and then separately for light ($R \leq 4$ mm h$^{-1}$), moderate ($4$ mm h$^{-1} < R \leq 12$ mm h$^{-1}$) and heavy ($R > 12$ mm h$^{-1}$) rainfalls. The parameters of the k-R model obtained for Duebendorf data are verified CML attenuation observations (CML 1147_1448) in Prague data.

**Gaseous attenuation:** The effect of air humidity and temperature on attenuation patterns is demonstrated during dry-weather periods on the sub-links belonging to the long CML. The observed attenuation patterns are compared to the theoretical patterns calculated from temperature and air humidity observations assuming constant atmospheric pressure 1013.25 hPa. The agreement between theoretical and observed attenuation is quantified in terms of correlations and the magnitude of their amplitudes. In addition, seasonal drift is demonstrated on time series smoothed by a moving average with a window size of one week.

## 4 Results

### 4.1 Identifying hardware related artifacts

There have been three types of hardware-related artifacts identified in the data (Fig. 5): a) a sudden change in $L_t$, b) long-term gradual $L_t$ drift, and c) 'degraded resolution'. 'Degraded resolution' is defined as behavior where $tx$ and $rx$ change with considerably lower frequency than is common with other CMLs. The degraded resolution can be easily recognized visually as a time series with no signal fluctuation within intervals of several hours.



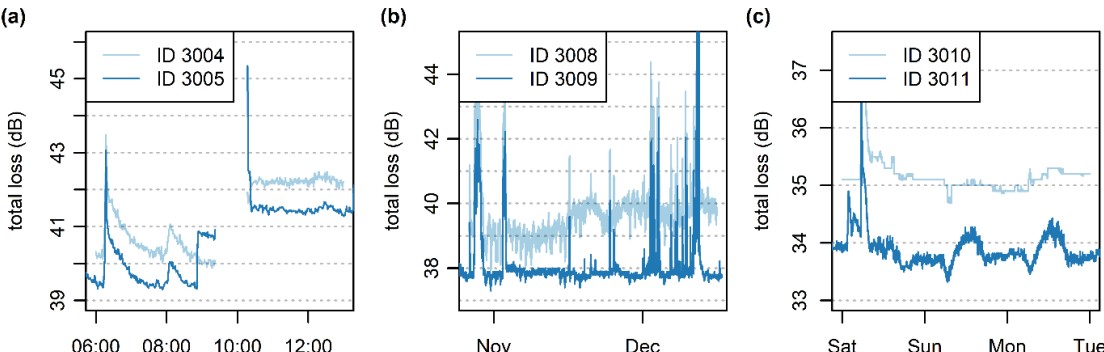


**Figure 5: Demonstration of hardware related artefacts (a) sudden change in the baseline of CML 3004_3005, (b) baseline drift of sub-link 3008, and (c) degraded resolution of sub-link 3010.**

The sudden change in $L_t$ of about 2 dB occurred on CML 3004_3005 on both sub-links (Fig. 5a). The change in baseline level was proceeded by approx. two hours of an outage and is probably related to the slight displacement of the CML unit during

maintenance work. The long-term gradual drift of $L_t$ occurred on sub-link 3008. $L_t$ levels observed during dry weather increased gradually, on average, to about 1.5 dB during the experimental period. Finally, sub-link 3010 was affected during the whole experimental period by a degraded resolution. Interestingly, the degraded resolution was more pronounced during dry weather periods than rainy ones.

In general, attenuation levels during dry weather are relatively stable with respect to long-term drift (in the order of weeks). It

holds for all CMLs except the 73 GHz sub-link of CML 3008_3009 which has dry weather attenuation levels of about 1.5 dB higher at the end of the period compared to the beginning (Fig. 5b). The most significant fluctuations in the baseline occur during dew events when water film forming on the CML antennas causes wet antenna attenuation (see subsection 4.4). The baseline fluctuations related to water vapor are presented in subsection 4.6.

### 4.2 Dry-wet classification

Dry-wet weather classifiers of single sub-links belonging to one CML are strongly correlated (Fig. 6). An exception is sub-link 3010 which is affected by a hardware malfunction (degraded resolution). The correlation between the classifiers of sub-links belonging to different CMLs is lower but still reaches high values ranging between r = 0.57 and r = 0.84. The correlation of CML classifiers to the classifiers based on rain gauges is, on average, slightly lower (r = 0.57–0.67). In general, the CMLs tend to detect rainfall earlier than rain gauges. It can be attributed to the delay of rain gauge rain detection due to the filling of

the bucket.



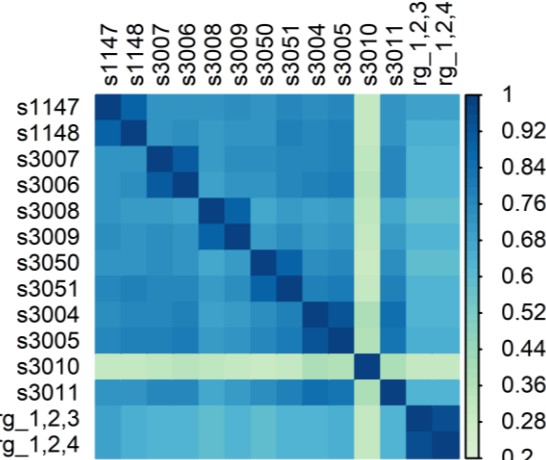

**Figure 6: Statistical relationship between dry-wet classifiers based on single CML sub-links and rain gauges along the path of the long CML (wet_rg1,2,3) and three rain gauges near five shorter CMLs (wet_rg1,2,4) expressed by correlation coefficient.**

### 4.3 Wet antenna attenuation

Figure 8 presents CML data at a 1-min temporal resolution featuring: i) attenuation during peak rainfall; ii) attenuation during dry spells at night on 3[rd] Nov and after a rainfall; and iii) attenuations during dew occurrence on the morning of 4[th] Nov. Attenuation during peak rainfall is dominated by raindrop path attenuation and is proportional to path length. In contrast, attenuation both during dry spells and after a rainfall, as well as attenuation during dew occurrences (with the exception of sub-link 3009) is dominated by WAA and, thus, independent of path length. Therefore, the WAA quantification method

utilizing different CML path lengths seems to be conceptually justified.

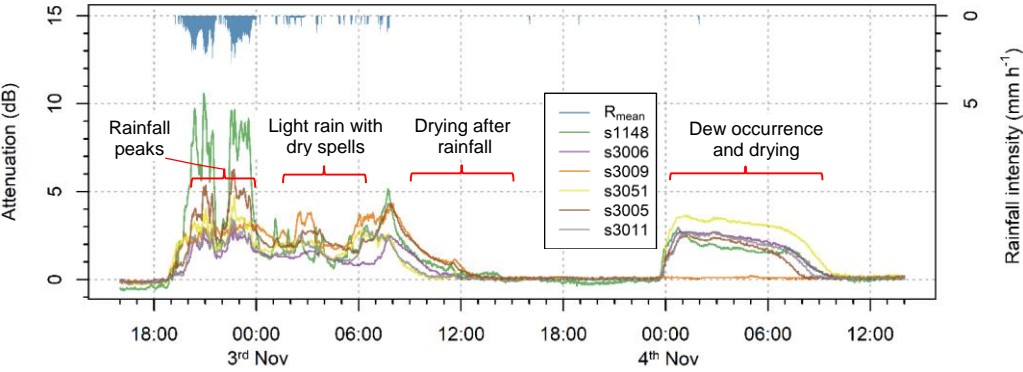

**Figure 7: Total rainfall-induced attenuation of 83–84 GHz sub-links and mean rainfall intensity from all four rain gauges. Period with peak rainfalls on 2[nd] Nov form approx. 19:00 to 00:00, period with light rainfall and dry spells on 3[rd] Nov form approx. 00:00**

**to 05:00, antenna drying period on 3[rd] Nov from approx. 08:00 to 14:00, and dew occurrence and subsequent antenna drying on 4[th] Nov form approx. 00:00 to 10:00.**





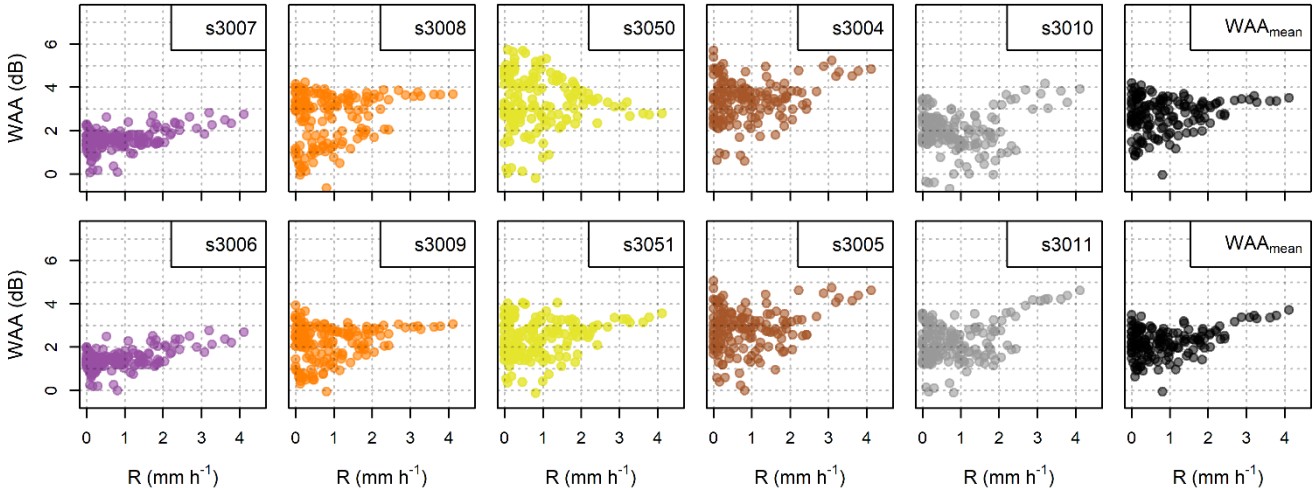

**Figure 8: Wet antenna attenuation during rainfall estimated from the differential attenuation of short and long CMLs. WAAs for both sub-links (73–74 GHz and 83–84 GHz) are shown for CML 1–5. The panels on the right side show mean WAA for all CMLs.**

WAA quantified for each shorter CML and their average is shown in Fig. 8 at a 15-min temporal resolution. The correlation to rainfall intensity is weak except for sub-links 3008 and 3006 with correlation coefficients r = 0.46 and r = 0.53, respectively. Higher rainfall intensities are, in general, associated with high WAA, whereas WAA reaches a wide range of values during lower intensities. WAA averaged over the whole evaluation period is between 1.60–3.47 dB for the 73–74 GHz sub-links and 1.41–2.48 for the 83–84 GHz sub-links. Further, inspection of CML time series reveals that attenuation after rainfall decreases

exponentially which is probably due to the drying of the antennas (Fig. 7, 3$^{rd}$ Nov).

WAA also contributes to total attenuation during the occurrence of dew when water condensates on the antenna radomes. Attenuation associated with dew deposition is similar for both frequency bands and reaches up to 4 dB (Fig. 7, 4$^{th}$ Nov morning). These values are higher than attenuation caused by rainfall.

### 4.4  Accuracy of the k-R power-law approximation

**Evaluation of theoretical DSD:** The relationship between attenuation and rainfall can be, for both frequencies, extremely well approximated by the power-law model, however, the parameters heavily depend on DSD (Fig. 9). For example, the specific attenuation 16 dB km$^{-1}$ corresponds to a rainfall intensity of about 30 mm h$^{-1}$ for rainfall with DSD typical for stratiform rainfalls. However, throughout convective rainfall the same specific attenuation would occur for rainfall intensities of about 50 mm h$^{-1}$. For both frequencies the model uses ITU parameter results between curves fitted to the rainfalls with

stratiform resp. convective DSD. However, it is closer to the 'stratiform' curve for lower rainfall intensities and approximates a better 'convective' curve for intensities higher than approx. 10–15 mm h$^{-1}$. The ITU parameters, therefore, provide a good approximation when no information on precipitation type is available.

**Evaluation on DSD data Duebendorf:** Similar power-law fits are obtained when modelling attenuation and rainfall from real DSD observations. Here, two types of rainfall are classified based on mass-weighted drop diameter $D_m$ (9). The fitting of the

classification threshold $\widehat{D_m}$ (10) results in parameters $c = 1.29$ h$^{-d}$ mm$^{1/d}$ and $d = 0.16$. The relation between rainfall intensity





 Atmospheric
and theoretical attenuation obtained is shown, together with fitted k-R power-law curves in Fig. 10. The relation between rainfall intensity and attenuation is clearly heteroscedastic. The k-R model deficiencies, therefore, increase with increasing rainfall intensity, as can be seen from the RMSE values (Table 4).

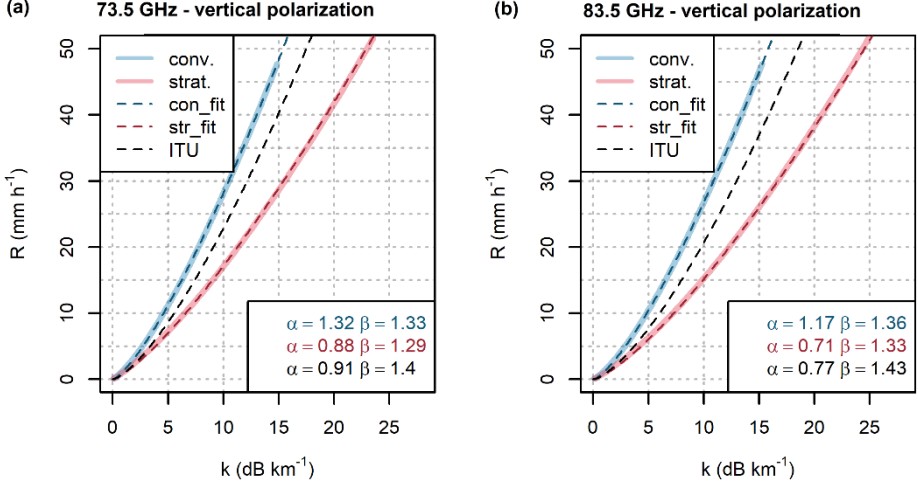

**Figure 9: Attenuation-rainfall relation for vertically polarized radio waves at (a) frequency 73.5 GHz and (b) 83.5 GHz derived from theoretical DSD corresponding to stratiform and convective rainfall. A k-R model (5) with parameters according to ITU (ITU-R P.838-3, 2005) lies between the curves corresponding to virtual convective and stratiform rainfalls. Parameters of the models are shown in the legends in the lower right-hand corners.**

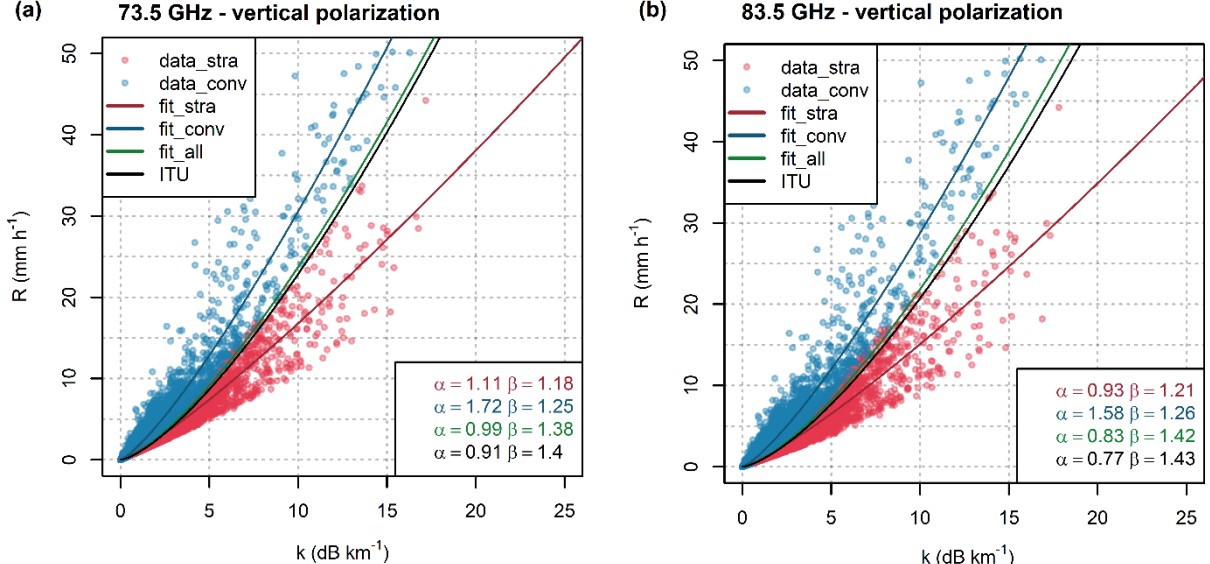

**Figure 10: Relation between specific attenuation and rainfall derived from one year of DSD data for vertically polarized radio waves at (a) frequency 73.5 GHz, and (b) 83.5 GHz. The k-R model (5) with parameters according to ITU (ITU-R P.838-3, 2005) resembles the model optimized for all the records. The curves optimized for convective and stratiform rainfalls differ significantly. Parameters of the models are shown in the legends in the lower right-hand corner.**



**Table 4: RMSE values (mm h$^{-1}$) calculated for observed and simulated rainfall using the k-R model with different parameter sets.**
**The evaluation is provided separately for light, moderate, and heavy rainfall as well as for the whole dataset.**

| Parameter set | Freq. (GHz) | RMSE (mm h$^{-1}$) | | | |
|---|---|---|---|---|---|
| | | all | $R \leq 4$ | $R = 4\text{–}12$ | $R > 12$ |
| Separate fit | 73.5 | 0.67 | 0.20 | 1.34 | 4.75 |
| | 83.5 | 0.73 | 0.24 | 1.48 | 5.08 |
| Joined fit | 73.5 | 1.17 | 0.43 | 2.46 | 8.03 |
| | 83.5 | 1.26 | 0.41 | 2.73 | 8.43 |
| ITU | 73.5 | 1.18 | 0.41 | 2.39 | 8.17 |
| | 83.5 | 1.27 | 0.45 | 2.63 | 8.74 |

## 4.5 Rainfall estimation

Figure 11 shows QPEs obtained from CMLs using the k-R model with ITU parameters and parameters derived from DSD during rainfalls classified as stratiform. Note that DSD is obtained from the independent Duebendorf dataset. The long CML,
in particular the 83.5 GHz sub-link, is capable of capturing even light rainfall intensities reliably. The correlation to rain-gauge observations is excellent (r ≈ 0.96). However, QPEs derived with ITU parameters tend to underestimate light rainfalls, which also leads to increased RMSE (Table 5). The model with DSD-derived parameters improves performance with respect to all metrics. Sub-link 1147 also remains significantly underestimated with DSD-derived parameters. This is due to deficits in the baseline and WAA identification. Shorter CMLs are less sensitive to rainfall along their shorter path and are more affected by
deficiencies in the estimated baseline and WAA. CMLs shorter than 1 km overestimate rainfall intensities more than longer CMLs.

In general, rainfall retrieval by E-band CMLs is affected during light rainfalls not only by deficiencies related to WAA and the baseline, but also to DSD. WAA and baseline-related errors clearly dominate by shorter CMLs, whereas regarding the 4.86-km-long CML, DSD seems to have a similar or, in the case of the 83.5 GHz sub-link, even larger effect. The effect of DSD is
likely to increase with high rainfall intensities, which were, however, not encountered during the case study period.





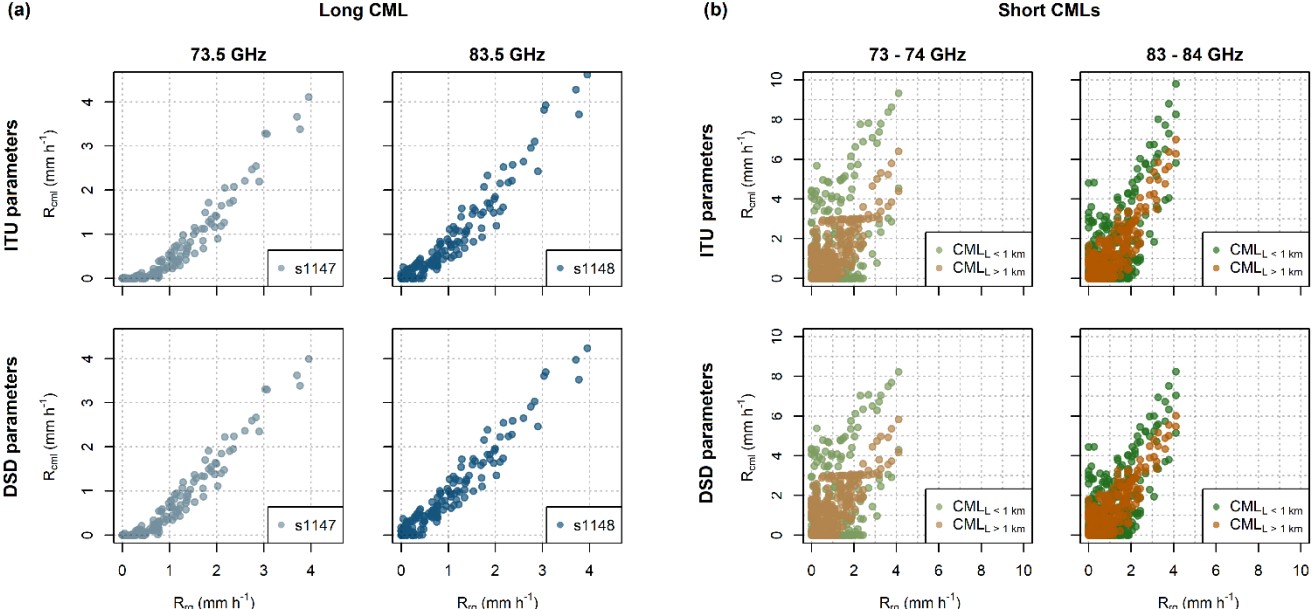

**Figure 11: CML QPEs for the long CML (a) and short (b) CMLs when using k-R model with ITU (top) and DSD-derived (bottom) parameters. Results are shown for both frequency ranges. QPEs for short CMLs are differentiated by color into two groups to depict CMLs with path lengths shorter and longer than 1 km separately.**

**Table 5: Performance of the CML QPEs obtained with the k-R model using ITU and DSD-derived parameters**

| Sub-link | ITU parameters | | | DSD parameters | | |
|---|---|---|---|---|---|---|
| | r | rel. | RMSE | r | rel. | RMSE |
| id | (-) | error (-) | (mm h$^{-1}$) | (-) | error (-) | (mm h$^{-1}$) |
| 1147 | 0.95 | -0.44 | 0.49 | 0.96 | -0.35 | 0.39 |
| 1148 | 0.96 | -0.17 | 0.31 | 0.97 | -0.08 | 0.24 |
| 3007 | 0.73 | -0.78 | 0.92 | 0.74 | -0.77 | 0.90 |
| 3006 | 0.80 | -0.61 | 0.79 | 0.81 | -0.59 | 0.76 |
| 3008 | 0.57 | 1.26 | 2.29 | 0.53 | 1.25 | 2.18 |
| 3009 | 0.63 | 0.61 | 1.56 | 0.61 | 0.59 | 1.44 |
| 3050 | 0.67 | 0.34 | 0.92 | 0.66 | 0.43 | 0.97 |
| 3051 | 0.84 | 0.15 | 0.71 | 0.83 | 0.21 | 0.69 |
| 3004 | 0.86 | 0.10 | 0.64 | 0.86 | 0.23 | 0.64 |
| 3005 | 0.87 | 0.26 | 0.75 | 0.87 | 0.33 | 0.69 |
| 3010 | 0.74 | -0.48 | 0.95 | 0.74 | -0.44 | 0.92 |
| 3011 | 0.78 | 0.24 | 1.31 | 0.77 | 0.26 | 1.14 |





### 4.6 Gaseous attenuation – effect of air humidity and temperature

Theoretical gaseous attenuation calculated from observed temperatures and relative humidity is highly correlated to water vapor density (r = 0.94–0.97) at both frequencies studied. The fluctuations in temperature affect this relation negligibly. The

further evaluation, therefore, concentrates on the comparison of theoretical attenuation to attenuation observed by two sub-links of the long CML 1147_1148. To separate gaseous attenuation from other possible attenuations, only periods with no rainfall are evaluated.

Time series of theoretical and observed attenuation are compared in Fig. 12 which shows time series of attenuations smoothed by a moving average (one-week window size). The correlation between theoretical and observed attenuation is high for both

sub-links (r = 0.82–0.83) and the long-term patterns of observed and theoretical attenuations correspond to each other quite well. Both theoretical and observed attenuations are higher during the summer period (August-September) and gradually decrease during the autumn period (October–December). The difference between mean attenuation levels in August and December is about 1 dB for the 83.5 GHz sub-link compared to only 0.7 dB for the 73.5 GHz sub-link. The theoretical and observed attenuations have similar median values for both frequencies during summer (2.11 resp. 2.12 dB for 73.5 GHz and

2.05 resp. 2.09 dB for 83.5 GHz). The theoretical and observed attenuations during autumn are about 0.3 dB higher for 73.5 GHz, compared to the 83.5 GHz sub-link (1.81 resp. 1.93 dB for 73.5 GHz compared to 1.58 resp. 1.65 dB for 83.5 GHz).

The higher attenuations of the 73.5 GHz sub-link during the autumn period, in comparison to the 83.5 GHz sub-link, can be explained by dry air attenuation. Dry air attenuation of 73.5 GHz is about 0.2 - 0.3 dB higher (depending on temperature) than that of 83.5 GHz. On the other hand, higher frequency bands are more sensitive to water vapor attenuation, which is higher

during summer. Different sensitivity to water vapor attenuation also causes more significant seasonal drift in the attenuation of the 83.5 GHz sub-link compared to the 73.5 GHz one.

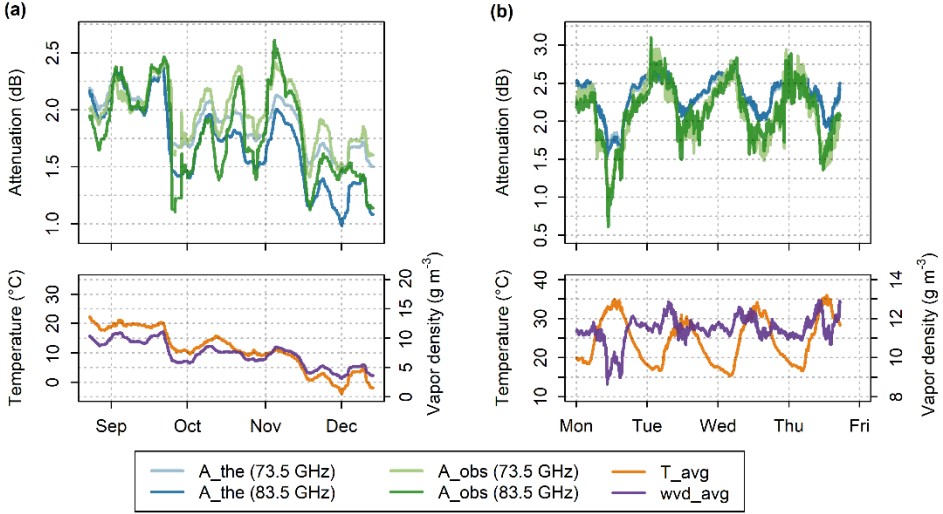

**Figure 12: Theoretical and observed attenuation from the 73.5 and 83.5 GHz sub-links of CML 1147_1148 – (a) data over the whole observation period smoothed by a moving average with a window size of one week; (b) 5-min data during four summer days.**


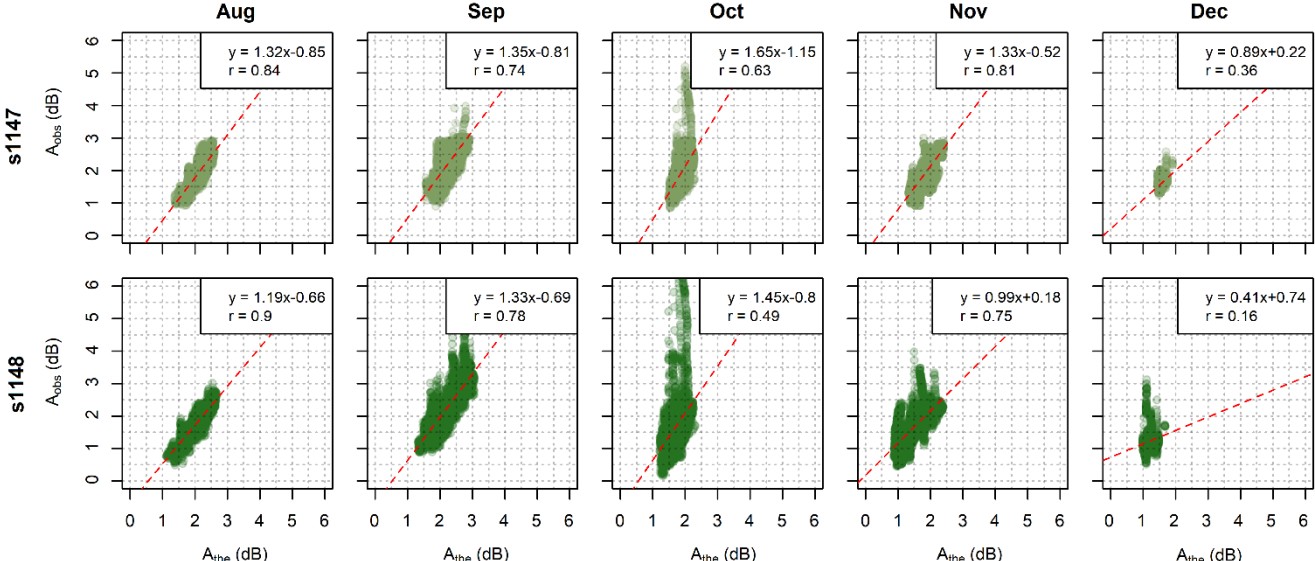

**Figure. 13: Comparison of theoretical (x-axis) and observed (y-axis) gas attenuations at the 73.5 and 83.5 GHz sub-links of the CML 1147_1148 for 5-min data. Data are shown separately for each month.**

The discrepancies between theoretical and observed attenuations are more pronounced when analyzing data at a 5-min

resolution, as demonstrated on the time series of four summer days shown in Fig. 12b. This is because the separation of gaseous attenuation from the other sources of attenuation or hardware related artifacts is challenging in real conditions. Despite these discrepancies, the correlation between theoretical and observed attenuations remains relatively high (r = 0.70–0.72). The theoretical and observed attenuations highly correlated during the summer period (Fig. 13) with the correlation coefficients reaching 0.84 and 0.90 for the 73.5 GHz resp. 83.5 GHz sub-link. The correlation is lowest during December (r = 0.36 resp.

480 0.16).

## 5    Discussion

**Hardware related artifacts:** The hardware related artifacts identified in the E-band attenuation time series are similar to those occurring on older 15–40 GHz CMLs. The 'degraded resolution' can be identified easily by analyzing attenuation variability within (sub)hourly subsets. Detecting (and correcting for) the sudden change in attenuation level will be especially challenging

in operation mode when attenuation needs to be processed in real-time. Long-term drift can be captured very well and corrected using a median moving window with a size of one week. Such a size of window is sufficiently long to not include more than 50 % of wet weather records into the window at any time step in the temperate climate.

The median moving window baseline is not suitable for distinguishing between long-term drift related to hardware malfunction (*e.g.*, sub-link 3008) and drift related to seasonal changes in air humidity and temperature (sub-links 1147 and 1148). Possible

water vapor monitoring thus poses higher requirements on the hardware with respect to the stability of the attenuation baseline. Note, our results are hardware specific and based on four months of CML records only.



**Dry-wet weather classification and baseline separation:** The evaluation of dry-wet weather classification is only approximate because tipping bucket rain gauges are unable to detect the exact beginning or end of a rain event. Visual inspection of time series reveals that disagreement between rain gauges and CMLs occur most commonly during dew events
and during periods of low temperature where mixed or snow events probably occur.

The dry-wet classification has been reported as an important step in CML pre-processing as it minimizes unwanted changes in attenuation level by setting the baseline separately for each event from the relatively short period before the event (Chwala and Kunstmann, 2019; Overeem et al., 2011). Here, dry-wet weather classification was not used for baseline identification. It was a pragmatic choice enabling better descriptions of the WAA effect. Dry-wet classification is needed for filtering out
periods with increased attenuation due to WAA (after rainfall and during dew events), nevertheless, these periods are in the event-based evaluation not included. It should be noted that, with the exception of one case, the observed attenuation levels were stable and the baseline identification method with a moving median (without dry-wet weather classification) performed well for rainfall retrieval purposes.

The separation of wet weather (including dew occurrence) was identified as a crucial step when analyzing attenuation due to
water vapor. When rain gauges are not available a CML-based classification needs to be performed. The dry-wet weather classification used (Schleiss and Berne, 2010) is designed to identify rainy periods and consider dew occurrences as dry weather. Although sensitivity to dew events can be increased by optimizing the parameters of the algorithm, dew events have similar dynamics as changes in air humidity as they are both dependent on temperature. Thus, other methods also considering observations of neighboring CMLs (Overeem et al., 2011) might be more appropriate for the dry-wet weather classification
used for the separation of attenuation caused by water vapor.

**Wet antenna attenuation:** Quantification of WAA during rainfall is based on the assumption that rainfall has a uniform distribution over the study area, and that water formation on the surface of antenna radomes is the same for both the short CMLs and the long one. In our case, the first assumption holds well as all four rain gauges observe similar rainfall intensities during the evaluated events. The correlation coefficient between rain gauges rg_1, rg_2 and rg_4, which are closer to each
other, is 0.94–0.96 and 0.88–0.93 for the more distant rg_3. The similarity in antenna characteristics was not inspected directly. That said, the estimation procedure is relatively insensitive to WAA occurring on the long CML as attenuation along its path dominates over WAA, even during relatively light rainfalls and, thus, WAA does not significantly influence the estimated specific attenuation (12).

WAA during rainfall is weakly correlated to rainfall intensity (*e.g.*, Schleiss et al., 2013). Our results are limited to light and
moderate rainfall only. Schleiss et al. (2013) reported drying of up to six hours with an exponential decrease of WAA, which also corresponds to our observations (Fig. 7). However, quantification of exact durations of drying require additional instrumentation to enable us to determine ends of rainfalls directly. The exponential WAA decrease during drying was also reported by Leth et al. (2018) who suggested that this drying pattern occurs on antennas with non-degraded coating, which is also the case of the CML antennas anlayzed here. On the other hand, WAA attenuation patterns on antennas with degraded
coating might be markedly different.





WAA due to water vapor condensation reaches higher values than during light rainfall. This might be caused by an absence of water rivulets (Leth et al., 2018). The higher values of attenuation caused by water droplets, in comparison to attenuation caused by rivulets, was also reported by Mancini et al. (2019). Comparable attenuation patterns of light rainfall and water vapor condensation may cause the misclassification of dew as rainfall.

In general, WAA quantified in this study is slightly higher than WAA reported for lower frequencies (Leth et al., 2018). However, the relative contribution of WAA to the total attenuation is less significant (given the high sensitivity of CMLs to raindrop path attenuation). WAA is, thus, a smaller source of possible bias than on 15–40 GHz frequencies, nevertheless, its accurate quantification is still important, especially for shorter CMLs. In addition, WAA during heavy rainfalls was not investigated in this study and might be higher, as was shown for lower frequencies by (Fencl et al., 2019).

**Rainfall estimation and effect of drop size distribution:** The relation between rainfall and raindrop attenuation (4, 5) on E-band frequencies is substantially more dependent on DSD than on 15–40 GHz CMLs (Chwala, 2017). The parameters of the power-law model (5), when optimized for all the DSD data, corresponds extremely well to the ITU parameters (ITU-R P.838-3, 2005). However, high RMSE results from the variability in DSD when using one fit for all the records. Separate fits for convective and stratiform rainfalls reduce the RMSE to half. Moreover, the separate power-law fits are closer to linear (beta

between 1.18 and 1.26, compared to beta between 1.38 and 1.4) and are less prone to errors related to non-uniform rainfall distribution along the CML path. Errors due to non-linearity of relation (5) might be reduced by reconstructing rainfall spatial variability along the CML path from the neighboring CMLs, or by introducing a climate-based relation between the non-uniformity of rainfall distribution and rainfall intensity. Such methods will, however, require further research. In general, unknown DSD will probably be one of the major uncertainties in quantitative estimates of heavy rainfall. On the other hand,

high sensitivity to DSD creates the opportunity to infer information on DSD from the attenuation of E-band CMLs, *e.g.*, in a condensed form of DSD moments. This is, in theory, also possible at 15–40 GHz, though difficult to accomplish in practice (Leth et al., 2019). Additional information on rainfall intensity or a combination with attenuation data of CMLs operating at lower frequencies will be required for DSD retrieval.

The E-band CMLs proved to be markedly more sensitive to raindrop path attenuation than 15–40 GHz devices. The long CML

provided surprisingly accurate rainfall estimates, even for light rainfalls lower than 1 mm h$^{-1}$ in intensity (Fig. 11). Assuming a detection threshold of 1 dB (typical *tx* power quantization of older devices), a 1-km-long 83 GHz CML can already detect rainfall intensity of 0.6 - 1 mm h$^{-1}$ depending on rainfall type, whereas, *e.g.*, a 23 GHz or 38 GHz CML (typical for older networks) only detects rainfalls heavier than 8.4 resp. 3.6 mm h$^{-1}$, *i.e.,* the sensitivity to light rainfalls is almost an order of magnitude higher for E-band CMLs. Moreover, the quantization of *rx* and *tx* records has improved to 0.1 dB with E-band

CMLs. On the other hand, long E-band CMLs are prone to outages (*rx* drops under detection level) during heavy rainfall. This high sensitivity to rainfall, together with improved quantization, opens the opportunity for monitoring rainfall with CMLs having a sub-kilometer path length, which was practically not possible before without adjusting CML QPEs to the rain gauges (Fencl et al., 2017). Short CMLs are more affected by errors related to WAA, yet the influence of WAA is relatively smaller during heavier rainfall, which, unfortunately, did not occur during the evaluation period. The use of short CMLs may be



convenient, especially during heavy rainfalls associated with high spatial variability by which an assumption about uniform rainfall distribution along a CML path is more likely valid than for long CMLs. Reliable rainfall estimation from short CMLs, however, requires further research on WAA modeling at E-band frequencies.

**Gaseous attenuation:** The theoretical gaseous attenuation for 73.5 GHz sub-link ranges between 1.33 dB and 2.90 dB (amplitude 0.33 dB km$^{-1}$) and the 83.5 GHz sub-link between 0.95 dB and 3.06 dB (amplitude 0.45 dB km$^{-1}$). These

fluctuations have a minor effect on rainfall retrieval (with respect to uncaptured baseline variability), as 0.33, resp. 0.45 dB km$^{-1}$ corresponds to a rainfall intensity of about 0.19 mm h$^{-1}$ resp. 0.25 mm h$^{-1}$ for 73.5 GHz and 83.5 GHz sub-links. On the other hand, this signal is sufficiently strong to enable the detection of water vapor at long CMLs, as 0.33 dB km$^{-1}$ resp. 0.45 dB km$^{-1}$ corresponds to the long CML (4.86 km) of 1.60 dB resp. 2.19 dB. The major challenge lies in the separation of gaseous attenuation from losses caused by other phenomena. This is easier during periods without rainfall, nevertheless, the following

causes of losses need to be identified and separated.

First, WAA occurring after rainfall and during dew events can reach 4 dB, *i.e.*, substantially exceeds the gaseous attention. Here, a safety window of 6 h size was used before and after each rain gauge tipping to exclude periods with WAA contribution. This mostly eliminated WAA before and after rainfall events and WAA during strong dew events causing rain-gauge tippings. However, such eliminations considerably reduce ratio of time intervals with observations. Moreover, periods before and after

rainfalls might have higher relative humidity than average and discarding those from the evaluation leads to potentially biased long-term estimates of water vapor density.

Second, signal fluctuations due to multipath propagation or other sources of uncertainty might affect the observed attenuation level. So-called multipath interference occurs due to the constructive or destructive phase summation of the signal at the receiving antenna during the atmospheric multipath propagation conditions (Valtr et al., 2011). Such interferences often lead

to a decreased signal power level of one sub-link while keeping the signal power level of the other sub-link.

Finally, hardware related artifacts might destroy a gaseous attenuation signal. For example, sub-link 3008 drifts about 1.5 dB during the period from the end of October to mid-December. This drift is clearly related to the hardware as the total loss due to gaseous attenuation along the path length of 0.39 km can reach only about 0.13 dB. Such a drift would, however, make the quantification of gaseous attenuation impossible even at long CMLs.

The separation of gaseous attenuation from other sources of signal loss is challenging. Further research could take advantage of the 10 GHz duplex separation between the sub-links of E-band CMLs. Combining attenuation information from CMLs of different lengths might also be promising.

**Limitations of this study:** The study investigates the weather monitoring capabilities of E-band CMLs on a dataset comprised of four months of attenuation data from six Ericsson MINILINK CMLs operated within cellular backhaul. The number of

CMLs and length of the period demonstrate the challenges and opportunities related to rainfall and water vapor monitoring at an E-band. The limited size of the dataset does not enable us to draw strong conclusions on the overall reliability of weather monitoring with an E-band, nor to investigate in detail new opportunities related to CML sensitivity to water vapor and DSD.





Specifically, the dataset does not include heavy rainfalls. The reliability of E-band CML rainfall estimation for heavy rainfalls
is based only on the evaluation of theoretical attenuations obtained from DSD observations (Duebendorf). The DSD effect on
the attenuation-rainfall relation could not be, therefore, studied in detail on the observed CML data. Finally, air temperature
and humidity are measured at two locations close to one node of the CML path. Despite these limitations, we believe that the
presented results reliably demonstrate new challenges and opportunities of E-band CML weather monitoring.

## 6  Conclusions

E-band microwave links are increasingly updating and frequently replacing the older hardware of backhaul networks operating
mostly at 15–40 GHz. This investigation demonstrates new challenges and opportunities related to CML weather monitoring.
The principles behind weather retrieval is the same as for lower frequency bands, nevertheless the influence of atmospheric
phenomena such as drop size distribution, or changes in air temperature and humidity affect radiowave propagation in a
significantly different manner. Furthermore, hardware used by E-bands is different (quantization, accuracy, antenna wetting,
etc.). The results, obtained from simulations and the case study with attenuation data from real-world CMLs, are encouraging.
The main conclusions are listed below:

- E-band CMLs are by about one order of magnitude more attenuated by raindrops along their path than older 15–40 GHz devices. This significantly improves the ability of E-band CMLs to quantify rainfall intensity accurately during light rainfalls.

- The rainfall retrieval at E-band frequencies is less influenced by wet antenna attenuation than by lower frequencies.
WAA observed in this study has a similar pattern as that described by (Schleiss et al., 2013), *i.e.*, it is almost uncorrelated with rainfall intensity and exhibits an exponential decrease after rainfall lasting up to several hours. WAA during dew occurrences reaches up to 4 dB.

- The power-law approximation of the attenuation-rainfall relation depends substantially more on DSD than on 15–40 GHz frequencies. The variability in DSD represents significant uncertainties in E-band CML rainfall retrieval. Use
of different parameter sets for different types of rainfall, as done with weather radars, reduce DSD-related errors, nevertheless this requires additional information on rainfall type.

- The k-R relation at E-band frequencies is less linear than at lower frequencies. This might cause errors, especially by longer CMLs, for which a uniform distribution of rainfall intensity along their path cannot always be assumed. On the other hand, even short (sub-kilometer) E-band CMLs are sufficiently sensitive to raindrop path attenuation to be
used for rainfall retrieval.

- Gaseous attenuation at E-band CMLs is detectable, however, it is two orders of magnitude smaller than attenuation due to rainfall. Gaseous attenuation is driven mainly by water vapor density and is, thus, in theory, an accurate predictor of this atmospheric variable. This, however, requires the efficient separation of attenuation from other signal losses which is, in practice, challenging. Our first results show that this separation is, to some extent, possible during
dry weather periods, if a sufficiently long CML (several km) is available.

In general, the ongoing shift of CML networks towards higher frequencies creates opportunities for the monitoring of rainfall on a qualitatively different level. New E-band CMLs are able to observe even light rainfalls and, in combination with lower frequency CMLs, potentially serve as DSD predictors. The rainfall retrieval methods developed for CMLs operating at 15–40 GHz frequencies proved to be useful for E-band CMLs as well. Water vapor retrieval from E-band CMLs having a path length of several kilometers might be possible, although the efficient separation of gaseous attenuation from other signal losses will be challenging in practice. This first experience with E-band weather retrieval, as presented in this study, will hopefully contribute to more robust designs of future experimental studies and case studies investigating this new technology with respect to weather monitoring.

*Supplement link (will be included by Copernicus).*

*Code and data availability.* The code for Prague data analysis and Prague data are publically available at Zenodo repository DOI 10.5281/zenodo.3632095. Duebendorf data (disdrometer observations) and the code are available upon request from the corresponding author.

*Author contribution.* MF and VB. designed the study layout. Data was collected by MF, VB and MD. Analysis was performed by MF with contribution of MD, MG, and PV. MF prepared the manuscript with contribution of all co-authors.

*Competing interests.* The authors declare that they have no conflict of interest.

*Acknowledgements.* This work was supported by the project of Czech Science Foundation (GACR) No. 17-16389S. We would like to thank T-Mobile Czech Republic a.s. for kindly providing us CML data and specifically to Pavel Kubík, for assisting with our numerous requests. Special thanks are extended to Prazska vodohospodarska spolecnost a.s. for providing rainfall data from their rain gauge network and Prazske vodovody a kanalizace, a.s. who carefully maintained the rain gauges. Last, but not least, we would like to thank Eawag for supporting COMMON project and Dr. Christian Chwala from Karlsruhe Institute of Technology (KIT) for supporting our analysis by calculating extinction cross-sections.

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
