# Peer review of "Atmospheric observations with E-band microwave links – challenges and opportunities"

_Atmospheric Measurement Techniques, 2020_

## Short Comment (SC1) · 20 Feb 2020

This paper by Fencl et al. presents for the first time an experimental investigation into the effect of rainfall and water vapour on the attenuation of links at E-Band, with the perspective of measuring rainfall in particular from these attenuations. The paper is well written, easy to follow and well organised. The topic is of high interest and these new experimental data and findings would be important for the growing scientific community working with CMLs as weather sensors. Here are some non-exhaustive general and technical comments on that paper, which I read with interest:

General comments

1. The introduction covers most important aspects but it omits to mention that attenuation by rainfall at these frequencies (E-Band) has already been investigated experimentally, e.g. by Shresta and Choi (2017) or Norouzian et al. (2020) for instance (and maybe other papers). This literature is often going under the radar of the atmospheric community maybe because it is published in specific engineering and electronics journals (often IEEE). Yes, the objective of these studies was the optimal design of backhaul networks, as to minimise the occurrence of signal fading, instead of opportunistic measurement of rainfall. But nethertheless, experimentally this is the same setup and collected data, and often even formulations and theoretical approaches. I would include a mention of their existence in the introduction, and eventually in the discussion if appropriate.

Sujan Shrestha, Dong-You Choi, Rain attenuation statistics over millimeter wave bands in South Korea, Journal of Atmospheric and Solar-Terrestrial Physics, Volumes 152–153, 2017, Pages 1-10, ISSN 1364-6826, https://doi.org/10.1016/j.jastp.2016.11.004.

F. Norouzian et al., "Rain Attenuation at Millimeter Wave and Low-THz Frequencies," in IEEE Transactions on Antennas and Propagation, vol. 68, no. 1, pp. 421-431, Jan. 2020, doi: 10.1109/TAP.2019.2938735.

2. L105: Which parameters values for the canting angle and temperature, and which model did you use for the T-Matrix calculations? You have to specify it here.

3. Effect of the k-R relation on the retrievals. Those are interesting results but I think the paper abstract does not reflect the actual results of this section. Typically, in Table 5: the performance criteria are actually quite similar between the use of ITU parameters and DSD parameters. Often only 0.01 differs for R2, and RMSE rarely exceeds 0.05 mm h-1. So only based on this, one might think the parameters used are not of real matter. But when looking at Table 4 and the parameters per rain type (stratiform and convective, low and moderate rain rates) one sees the discrepancies. So it is the k-R relations for local DSD and differentiation for specific rain classes/types being used

that has an impact on the retrievals at these frequencies. When lumped all together, ITU or specific DSD actually lead to similar retrievals outcomes. As the authors noted, including larger rain rates in the dataset would increase the differences in retrievals and the importance of the DSD on the retrievals at these rain rates.

4. At these frequencies, longer CML path lengths translates in higher sensitivities to light rainfall – but it also means a coarser spatial resolution for CML rainfall maps – this can be a drawback and can be highlighted.

Specific technical comments

L52 Comma missing before "which"

L115 units of v(D)?

L130 units of these variables?

L145 why not calling it the "Liebe model" as you did for the "Leijnse model"?

Table 5: Instead of "Performance", use "performance metrics", or "evaluation criteria"?

There are a couple of instances in the paper when the citation format is not suitable, in particular l137 "…in (Liebe et al., 1993)…" should be "in Liebe et al. (1993)". L166 has the same issue. L226 has the same issue. L263 has the same problem. L197 has the same issue.

"Parsivel" is an acronym and therefore should be written in Upper case letters, e.g. PARSIVEL

L238 Dm is in mm

Figure 4 (b) there is a typo for "convective". The legend has the same citation format issue (brackets should only be around the date).

L265 and 265, keep a consistent spacing between value and units (%). I think there should be a space.

L277 units of Aw? (dB km-1)

L290 units of Awconst?

Figure 7, the word "dry spells" seems cut at the bottom. Abbreviation for the month of November should be "Nov." if following the correct abbreviation, otherwise writing it in full could be actually better.

L497 chronological order for citations should be preferred

L571 "attention" should be "attenuation"

L 609 "by lower frequencies" "by" should be removed.
* * *

---

## Author Comment (AC1) · 25 Feb 2020

First of all, we would like to thank Dr. Guyot for his valuable comments and suggestions. Below are our reactions:

**General comments**

1. *The introduction covers most important aspects but it omits to mention that attenuation by rainfall at these frequencies (E-Band) has already been investigated experimentally, e.g. by Shresta and Choi (2017) or Norouzian et al. (2020) for*

[Figure]

*instance (and maybe other papers). This literature is often going under the radar of the atmospheric community maybe because it is published in specific engineering and electronics journals (often IEEE). Yes, the objective of these studies was the optimal design of back-haul networks, as to minimise the occurrence of signal fading, instead of opportunistic measurement of rainfall. But nethertheless, experimentally this is the same setup and collected data, and often even formulations and theoretical approaches. I would in-clude a mention of their existence in the introduction, and eventually in the discussion if appropriate.*

This is an interesting point. We are aware of E-band investigation being reported in the electrical engineering journals (often IEEE) and refer to two examples of this work in the Introduction at L51 (Hansryd et al., 2010; Luini et al., 2018) and two other examples in the section about wet antenna attenuation at L94-101 (Hong et al., 2017; Ostrometzky et al., 2018). The investigations of Hansryd et al. (2010) and Luini et al. (2018) have similar scope as the papers suggested by Dr. Guyot. We have been considering also referring to other works, nevertheless, the propagation studies aiming at community designing microwave link networks are mostly focused on long-term availability and lack evaluation in shorter time scales, which is critical for evaluating retrieval of atmospheric variables. We thus believe, that our relatively concise list of references on radio engineering investigations suffices as it, first, covers most important topics being previously investigated and, secondly, can provide further reading through referenced or citing studies. On the other hand, we might have missed some important point in the suggested references or some topic in the radio engineering literature and will be thus happy for a notice.

2. *L105: Which parameters values for the canting angle and temperature, and which model did you use for the T-Matrix calculations? You have to specify it here.*

Thank you for notifying us of missing piece of information. Python implementation of the T-Matrix model suggested by Mishchenko and Travis (1998) was used. We

will add a reference to this implementation (Leinonen, 2014) in the revised version of the paper. The calculation assumes temperature 10° C, canting angle 0°, drop shape being oblate spheroid, drop axial ratio according to Pruppacher and Beard (1970), and for drop smaller than 0.5 mm a heuristic approximation of Pruppacher and Beard's formula is used. We will provide this information in the revised paper.

3. *Effect of the k-R relation on the retrievals. Those are interesting results but I think the paper abstract does not reflect the actual results of this section. Typically, in Table 5: the performance criteria are actually quite similar between the use of ITU parameters and DSD parameters. Often only 0.01 differs for R2, and RMSE rarely exceeds 0.05mm h-1. So only based on this, one might think the parameters used are not of real matter. But when looking at Table 4 and the parameters per rain type (stratiform and convective, low and moderate rain rates) one sees the discrepancies. So it is the k-R relations for local DSD and differentiation for specific rain classes/types being used that has an impact on the retrievals at these frequencies. When lumped all together, ITU or specific DSD actually lead to similar retrievals outcomes. As the authors noted, including larger rain rates in the dataset would increase the differences in retrievals and the importance of the DSD on the retrievals at these rain rates.*

The conclusions about sensitivity to drop size distribution are based on theoretical evaluation using attenuation and rainfall calculated from drop size distribution data. The effect of DSD is also demonstrated on real commercial microwave links (CMLs), nevertheless, other sources of errors, especially wet antenna attenuation (WAA) affect these results significantly. Moreover, our results are affected by the absence of heavy rainfalls during the experimental period (see Discussion section L589 – 595). This is the reason why only the longest CML is suitable for demonstrating DSD related errors, as stated in the Performance evaluation subsection (L319 – 324). The shorter CMLs are overly affected by WAA during light rainfalls to enable interpretations on DSD errors. We might stress this also

when referring to Table 5 or at another place in the manuscript to avoid confusion. We are, however, convinced that results from the long CML are capable of demonstrating benefit of using parameters specifically derived for stratiform rainfalls. Although the improvement in the metrics is not large in absolute values it is significant in relative values. E.g. decrease in RMSE from 0.49 to 0.39 mm h$^{-1}$ by 73.5 GHz sub-link and from 0.31 to 0.24 mm h$^{-1}$ by 83.5 GHz sub-link is decreased by 20 % resp. 23 %. This is even more pronounced in relative error where improvement from -0.44 to -0.35 by 73.5 GHz sub-link and -0.17 to -0.08 by 83.5 GHz sub-link is an improvement by 20 % resp. 53 %. Later, we will consider modification of the manuscript in this direction based on reviewers' comments.

4. *At these frequencies, longer CML path lengths translates in higher sensitivities to light rainfall – but it also means a coarser spatial resolution for CML rainfall maps – this can be a drawback and can be highlighted.*

   The higher sensitivity of longer CMLs to total rainfall attention is intrinsic to all frequency regions and the investigation of rainfall reconstruction goes beyond the scope of this study. Investigation addressing specifically CML rainfall reconstruction errors is presented for example in Rios Gaona et al. (2015).

**Specific comments**

We would like to thank Dr. Guyot for tracking our manuscript for typos and shortcomings. We will address all of them in the revised manuscript as suggested.

**Additional references**

Sujan Shrestha, Dong-You Choi, Rain attenuation statistics over millimeter wave bands in South Korea, Journal of Atmospheric and Solar-Terrestrial Physics, Volumes 152–153, 2017, Pages 1-10, ISSN 1364-6826, https://doi.org/10.1016/j.jastp.2016.11.004.

F. Norouzian et al., "Rain Attenuation at Millimeter Wave and Low-THz Frequencies,"in IEEE Transactions on Antennas and Propagation, vol. 68, no. 1, pp. 421-431, Jan.2020, doi: 10.1109/TAP.2019.2938735.

Hansryd, J., Li, Y., Chen, J. and Ligander, P.: Long term path attenuation measurement of the 71–76 GHz band in a 70/80 GHz microwave link, in Proceedings of the Fourth European Conference on Antennas and Propagation, pp. 1–4., 2010.

Hong, E. S., Lane, S., Murrell, D., Tarasenko, N. and Christodoulou, C.: Mitigation of Reflector Dish Wet Antenna Effect at 72 and 84 GHz, IEEE Antennas and Wireless Propagation Letters, 16, 3100–3103, doi:10.1109/LAWP.2017.2762519, 2017.

Leinonen, J.: High-level interface to T-matrix scattering calculations: architecture, capabilities and limitations, Opt. Express, OE, 22(2), 1655–1660, doi:10.1364/OE.22.001655, 2014.

Luini, L., Roveda, G., Zaffaroni, M., Costa, M. and Riva, C.: EM wave propagation experiment at E band and D band for 5G wireless systems: Preliminary results, in 12th European Conference on Antennas and Propagation (EuCAP 2018), pp. 1–5., 2018.

Mishchenko, M. I. and Travis, L. D.: Capabilities and limitations of a current FORTRAN implementation of the T-matrix method for randomly oriented, rotationally symmetric scatterers, Journal of Quantitative Spectroscopy and Radiative Transfer, 60(3), 309–324, 1998.
Ostrometzky, J., Raich, R., Bao, L., Hansryd, J. and Messer, H.: The Wet-Antenna Effect—A Factor to be Considered in Future Communication Networks, IEEE Transactions on Antennas and Propagation, 66(1), 315–322, doi:10.1109/TAP.2017.2767620, 2018.

Pruppacher, H. R. and Beard, K. V.: A wind tunnel investigation of the internal circulation and shape of water drops falling at terminal velocity in air, Quarterly Journal of the Royal Meteorological Society, 96(408), 247–256, doi:10.1002/qj.49709640807, 1970.

Rios Gaona, M. F., Overeem, A., Leijnse, H. and Uijlenhoet, R.: Measurement and interpolation uncertainties in rainfall maps from cellular communication networks, Hydrol. Earth Syst. Sci., 19(8), 3571–3584, doi:10.5194/hess-19-3571-2015, 2015.

---

## Referee Comment (RC1) · Anonymous Referee #1 · 16 Mar 2020

The paper investigates challenges and opportunities of the E-band microwav links to derive rainfall. The study mirros previous studies conducted at lower frequency, it is potentially of great interest but at the moment the paper presents some major issues.

1) Fig.1: in the caption "scattering efficiency" is mentioned. Clearly this is not the same as extinction efficieny. Please clarify.

2) Fig.2: I supsect there is something wrong here. I do not see why the water vapor attenuation should have a drop at 60 GHz. Is this affecting results later on???

3) I find the narrative from 3.3 onwards (including Sect4) very difficult to follow. I would recommend to reshu so that for instance when you talk about Sensitivity of the k-

R model to drop size distribution you cover the whole thing (including line 400-425). Same for the other bits (e.g. Quality check, Dry-wet weather classification, Baseline identification,Wet antenna attenuation). At the moment the reader need to jump back and forward because the logical thread is erratic. Also some topic (e.g. gas attenuation should come before rainfall retrieval because of course the effect of gas must be subtracted first!).

4) Eq.10 and Fig.4b. Where is this coming from? I have never seen such a relationship between an intensive quantity like RR and D_M!!!!!

5) Table 3: you are introducing parameters (epsilon, delta,) that are not defined anywhere.

6) I do not understand the rationale of doing the investigation with the "theoretical DSD" (not clear where they come from). On the other hand I see the point of using disdrometer data but I would recommend to use extensive datasets like available at https://ghrc.nsstc.nasa.gov/home/field-campaigns (and plot density functions instead of plotting scatterplots as in Fig.10). This should also allow to assess uncertainty errors due to DSD variability like done in Tab.4 in a more robust way.

7) The authors mention stratiform vs convective precipitation coefficients. How do they practically envisage to separate stratiform vs convective precipitation?

8) Assuming that "rainfall has a uniform distribution over the study area, and that water formation on the surface of antenna radomes is the same for both the short CMLs and the long one" is quite an assumption! This approach is very provisional.

9) "E-band CMLs are by about one order of magnitude more attenuated by raindrops along their path than older 15–40 GHz devices": this is a very vague (imprecise) statement also given the fact that attenuation at 40 GHz is already 6 times attenuation at 15 GHz!!!! Same for the sentence "Gaseous attenuation at E-band CMLs is detectable, however, it is two orders of magnitude smaller than attenuation due to rainfall" (again

quite vague and approximate!)

---

## Author Comment (AC2) · 31 Mar 2020

First of all, we would like to thank the reviewer for his valuable comments and suggestions. Below are our reactions:

**Specific comments**

1. *Fig.1: in the caption "scattering efficiency" is mentioned. Clearly this is not the same as extinction efficieny. Please clarify.*

   Thank you for spotting this inconsistency. The figure as well as equation 6 describe extinction efficiency ($Q_{ext}$). We will correct this.

2. *Fig.2: I suspect there is something wrong here. I do not see why the water vapor attenuation should have a drop at 60 GHz. Is this affecting results later on???*

We have checked our results and there is nothing wrong with it. Results are fully consistent with Liebe et al. (1993) and could be cross-checked by independent computer codes available on GitHub, e.g. https://github.com/cchwala/pyMPM.

The attenuation due to water vapor is in here (as well as later on) defined as the difference between wet-air and dry-air attenuation under the same moist air pressure and temperature. Thus, also effect of water vapor on attenuation due to oxygen is considered as described in ITU-R, (2019), and formerly by Liebe et al. (1993): First, moist air pressure is the sum of dry-air pressure and partial water pressure, thus dry-air pressure decreases during humid conditions (under the assumption of the same moist air pressure). This leads to a decrease in attenuation by molecules of oxygen. Second, partial water vapor pressure influences the width of the oxygen spectral lines (pressure broadening) as it affects the rate of collisions between the molecules (eq. 5 and 6a in ITU-R (2019)). These two effects lead to a decrease in attenuation due to oxygen when water vapor content increases. This decrease can reach about 4 % for conditions with moist-air pressure 1015 hPa, temperature 30° C, and 100 % air humidity. Such decrease has relatively negligible effect for frequencies used in this study (73-74 and 83-84 GHz). Nevertheless, it is not negligible around 60 GHz (as can be seen on Figure 2), where the attenuation due to oxygen dominates over attenuation directly caused by water vapor molecules.

We will clarify the definition of attenuation due to water vapor in the paragraph describing the Figure 2 (L150-153 in the original manuscript). We will also add an explanation how water vapor influences attenuation due to oxygen. Finally, the reference to ITU-R recommendations (ITU-R, 2016) will be updated to ITU-R (2019).

3. *I find the narrative from 3.3 onwards (including Sect4) very difficult to follow. I would recommend to reshu so that for instance when you talk about Sensitivity of the k-R model to drop size distribution you cover the whole thing (including line 400-425). Same for the other bits (e.g. Quality check, Dry-wet weather classification, Baseline identification, Wet antenna attenuation). At the moment the reader need to jump back and forward because the logical thread is erratic. Also some topic (e.g. gas attenuation should come before rainfall retrieval because of course the effect of gas must be subtracted first!).*

We thank for the suggestions on modifying the structure of the manuscript. First part of the comment suggests presenting jointly methods and results belonging to the one subtopic. However, at this moment, we prefer to avoid the inclusion of results in the Materials and Methods section. Moreover, the research performed is multifocal, the use of subheadings within the Materials and Methods section, that are mirror-imaged in the Results section, is intended to make it easier for readers of the manuscript to associate which Materials and Methods are used to obtain which results. Furthermore, all the subtopics (e.g. Quality check, Dry-wet weather classification, Baseline identification, Wet antenna attenuation) are part of the same processing chain and influence each other and thus it is in our view reasonable to present them together.

Regarding the second suggestion (re-order rainfall retrieval and gas attenuation), we agree that effect of gaseous attenuation influences baseline identification and can be presented earlier. We will modify the structure of the Methods, Results, and Discussion section accordingly.

4. *Eq.10 and Fig.4b. Where is this coming from? I have never seen such a relationship between an intensive quantity like RR and $D_M$!!!!!*

The parameters of theoretical functions describing drop size distribution (DSD) are scaled to rainfall intensity, as can be seen in Figure 4a. In this particular case, gamma distribution as proposed by Ulbrich (1983) is used. Thus, mass-weighted

diameters (ratio between 4th and 3th DSD moments) need to be also scaled to rainfall intensity, as shown in Figure 4b. The scaling procedure is explained in the paragraph above the eq. 10 (lines 224-226 of the original manuscript) and adequately referenced to the benchmark paper of Ulbrich (1983).

5. *Table 3: you are introducing parameters (epsilon, delta,) that are not defined anywhere.*

   The parameters are taken from Table 2 in Ulbrich (1983). The paper was cited in the Table caption of the original manuscript. We will add details on each parameter to the Table:

   $N_0(m^{-3}cm^{-1-\mu})$ and $\mu$ (-) are parameters of semi-empirical gamma distribution function from Ulbrich (1983). $\epsilon(h^{-\delta})$ and $\delta$ (-) are scaling parameters of this function as proposed by Ulbrich (1983).

6. *I do not understand the rationale of doing the investigation with the "theoretical DSD" (not clear where they come from). On the other hand I see the point of using disdrometer data but I would recommend to use extensive datasets like available at https://ghrc.nsstc.nasa.gov/home/field-campaigns (and plot density functions instead of plotting scatterplots as in Fig.10). This should also allow to assess uncertainty errors due to DSD variability like done in Tab.4 in a more robust way.*

   Theoretical DSD enables us to show, that results based on observed DSD are consistent with results using widely acknowledged theoretical DSD functions, which in our opinion supports the conclusions and shows that the results are not only site-specific. When describing theoretical DSD, we use gamma distribution function scaled to rainfall intensity as suggested by Ulbrich (1983).

   In our opinion, one year of DSD data from the well-controlled experiment is sufficient to demonstrate the effect of DSD on attenuation-rainfall power-law relation. The dataset was previously used in several papers in high-quality journals (e.g.,

Schleiss et al., 2013; Wang et al., 2012) and we had the dataset handy. Moreover, the dataset is from temperate climate and thus applicable also for our case study.

7. *The authors mention stratiform vs convective precipitation coefficients. How do they practically envisage to separate stratiform vs convective precipitation?*

The separation to convective and stratiform rainfalls is performed only in the investigation with theoretical and observed DSD (sections 3.3 and 4.4 of the original manuscript) to demonstrate that attenuation-rainfall relation is sensitive to rainfall type. The classification procedure is described in section 3.3 of the original manuscript.

The separation of convective and stratiform rainfalls in practical applications is discussed in Discussion section of the original manuscript (L545-548). This section refers to the work of Leth et al. (2019). Unfortunately, detailed elaboration on possible methods for separation of convective and stratiform rainfalls is out of the scope of this manuscript.

8. *Assuming that "rainfall has a uniform distribution over the study area, and that water formation on the surface of antenna radomes is the same for both the short CMLs and the long one" is quite an assumption! This approach is very provisional.*

We agree that assumptions seem to be strong. Nevertheless, we believe that its appropriateness is, in our case, justified and was carefully discussed in Discussion section of the original manuscript (L511-518). Line of reasoning is briefly repeated below.

The rainfall spatial variability (in the studied period) is low: The correlation coefficient between 15-min rainfall observations of nearby rain gauges (rg_1 and rg_2) is 0.94–0.96. The correlation between these rain gauges and the more distant rain gauge rg_3 is still over 0.88.

Wet antenna attenuation (WAA): All the CML units are of the same type and have similar ages (deployed during 2016 and 2017). Furthermore, as discussed on L516-518 of the original manuscript, the procedure is in our case relatively insensitive to differences in antenna properties, because WAA is quantified by comparing short (0.4 – 1.4 km) CMLs by which WAA dominates over path attenuation, to the 4.86 km long CML, which is relatively insensitive to WAA even during light rainfalls. Satisfactory performance of the approach is also demonstrated in the Results section in Figure 7.

To further strengthen the arguments about the legitimacy of those assumptions, we will add information on the age of the units into section 3.1 Experimental sites and instrumentation.

9. *E-band CMLs are by about one order of magnitude more attenuated by raindrops along their path than older 15–40 GHz devices": this is a very vague (imprecise) statement also given the fact that attenuation at 40 GHz is already 6 times attenuation at 15 GHz!!!! Same for the sentence "Gaseous attenuation at E-band CMLs is detectable, however, it is two orders of magnitude smaller than attenuation due to rainfall" (again quite vague and approximate!)*

Agreed. We will be more precise in our statements:

- "E-band CMLs are markedly more attenuated by raindrops along their path than older 15–40 GHz devices, during lighter rainfalls by about 20 times more than 15 GHz and 2 - 3 times more than 40 GHz devices."

- "Gaseous attenuation at E-band CMLs is detectable, however, it is substantially smaller than attenuation due to rainfall. Fluctuations in specific attenuation caused by water vapor typically not exceed 1 dB km$^{-1}$ in the region of temperate climate. This magnitude is reached by rainfall with intensity around 1 mm h$^{-1}$."

**Additional references**

ITU-R: ITU-R P.676-11, [online] Available from: https://www.itu.int/dms_pubrec/itu-r/rec/p/R-REC-P.676-11-201609-I!!PDF-E.pdf, 2016.

ITU-R: RECOMMENDATION ITU-R P.676-12 - Attenuation by atmospheric gases and related effects, (online) Available from: https://www.itu.int/dms_pubrec/itu-r/rec/p/R-REC-P.676-12-201908-I!!PDF-E.pdf, 2019.

Leth, T. C. van, Leijnse, H., Overeem, A. and Uijlenhoet, R.: Estimating raindrop size distributions using microwave link measurements, Atmospheric Measurement Techniques Discussions, 1–27, doi:https://doi.org/10.5194/amt-2019-51, 2019.

Liebe, H. J., Hufford, G. A. and Cotton, M. G.: Propagation modeling of moist air and suspended water/ice particles at frequencies below 1000 GHz. [online] Available from: http://adsabs.harvard.edu/abs/1993apet.agar.....L (Accessed 17 October 2019), 1993.

Schleiss, M., Rieckermann, J. and Berne, A.: Quantification and Modeling of Wet-Antenna Attenuation for Commercial Microwave Links, IEEE Geoscience and Remote Sensing Letters, 10(5), 1195–1199, doi:10.1109/LGRS.2012.2236074,2013.

Ulbrich, C. W.: Natural Variations in the Analytical Form of the Raindrop Size Distribution, J. Climate Appl. Meteor., 22(10), 1764–1775, doi:10.1175/1520-0450(1983)022<1764:NVITAF>2.0.CO;2, 1983.

Wang, Z., Schleiss, M., Jaffrain, J., Berne, A. and Rieckermann, J.: Using Markov switching models to infer dry and rainy periods from telecommunication microwave link

signals, Atmospheric Measurement Techniques, 5(7), 1847–1859, doi:10.5194/amt-5-1847-2012, 2012.

---

## Referee Comment (RC2) · Giacomo Roversi (Referee) · 17 Apr 2020

The paper by Fencl et al. addresses a topical and interesting matter, as extends known opportunistic precipitation sensing techniques to the more recent E band links. It highlights the new possibilities uncovered by the different frequencies and hardware and focuses on the consequent challenges. The authors give a complete picture of the subject from theory to application, preparing the ground for future studies. The article is therefore certainly valuable and of primary interest to the CML scientific community and AMT readers.

The work is well written and the goals defined in the abstract and introduction are all

met. The discussion of the main issues is complete and rich, while some redundancy and repetitiveness is found in introductory and methodological sections, combined in certain cases with lack of the detailed quantitative information needed to contextualize some statements. Accordingly, a minor revision is suggested in order to provide the reader with more concise and relevant informations in the cases treated in the comments below.

The author's answers to previous comments (AC1 and AC2 to SC1 and RC1 resp.) have been taken into consideration.

General comments:

1. The fragmentation of the presentation as reported in comment 3. of RC1 is recognized: most of the topics are introduced in Sections 2 and 3 and then corroborated with quantitative data only in section 4 or even 5. Given the different data sets and methods utilized for the various steps of the investigation, the reading results sometimes erratic indeed. However the intentions declared by the authors (AC2) are also well understood. I will then strongly encourage a more widespread use of subsection cross referencing, to help the reader understand without changing the logical structure of the paper. An example of convenient referencing is found e.g. in L362 and 363. This should be replicated diffusely to connect introductory and discussion Sections. It seems to me that multipath disturbance instead is not introduced at all before L577 and should be added to Section 2 with some estimate of its magnitude.

2. Another downside of the chosen presentation layout is the need of re-introducing some aspects generally many times throughout the paper, without going quickly into the necessary detail. A more concise and unitary approach to the problems encountered and the solutions adopted would facilitate a global understanding of the work. I suggest therefore to support the introductory informations, in the first sections already, with quantitative informations and stating author's intentions regarding approximations and further discussions. In that way the reader could expect what to find in the next sections

and repetitive recalls to the qualitative introduction would not be needed. Some non-exhaustive examples are reported below and most of the specific comments deal with this same issue. L74 to 78 - though the paragraph's introductory intent is clear, it lacks the detail and clearness about which assumptions are kept and which are discussed, with respect to previous 15-40 GHz approaches. L94 to 101 - It is not clear at this point how the authors will deal with the reported considerations further in the paper.

3. An additional figure showing WAA against link length could be used to illustrate the linear regressions proposed in Eq. 11 and the constant behaviour in dew cases. A sample of how the figure could look is attached.

Specific comments (in order of appearance):

L68 - Free space loss (Lbf) is said to be uniquely defined by distance and wavelength. Reporting the formula could be appropriate and helpful for further understanding of the discussion, as the frequency is a key variable for this study (E band).

L74-78 - The phrase "Attenuation during dry weather is assumed to be a baseline" is apparently in direct contrast with the following "Fluctuations in the baseline during dry weather can be attributed..." if the reader does not know already the different magnitudes involved. Early introduction of orders of magnitude and average behaviours is therefore encouraged.

L101 - "More extensive investigations..." I think this sentence will state the motivation of the author's work, but it could be also interpreted as what still remains unknown after the work's results instead. Please clarify to avoid this ambivalence.

L131 - Fig. 4 is useful to the contextualisation of this sentence and should be referenced. "Contribute relatively less" is not gaugeable, some more detail may be added.

L145 and following - The study on the components of N is not justified by following discussion or results and could be omitted as it lacks quantitative information. I think that the qualitative concept of the dependency of k to the various components is already

well stressed.

L194 - "The periods for evaluating rainfall retrieval and for evaluating the effect of humidity and temperature fluctuations on gaseous attenuation are, therefore, different." The phrase itself is a quite obvious consequence of the previous sentence, while its implications are not. It should either be omitted or some expected implications should also be discussed (or at least some reference to the respective discussion should be made) in terms e.g. of which investigations are precluded by using different time windows.

L200 and other appearances of "aggregate" - it should be pointed out how the aggregation to different time scales is performed (mean, median, sum, max, other...)

L246 and 247 - The sentence is not clear and should be rephrased and expanded. "dependent" should perhaps be substituted with "depending", commas before and after "therefore" are not necessary and slow the reading. The threshold for Dm is not indicated.

L259 - Visual inspection does not seem like a robust approach to filter the outliers. Some technique should be at least suggested to cope with this kind of artefacts, as the visual approach is clearly not feasible at larger and near real-time scales.

L269 - One-week sized moving window "is sufficiently short": are baseline drifts proven to happen only at longer time scales? Is the same for gaseous attenuation? Could it be that some higher frequency signal is masked by this approach resulting in the weakening of the water vapour detection capabilities?

L283 - A reference to Fig. 8 or to the suggested new figure could be added here.

L296 and 340 - Since Prague is located at an altitude around 200m (990 hPa), to assume the atmospheric pressure of 1013.25 hPa seems either systematically wrong or reported with too high precision (if differences between 990 and 1013 are negligible for the author purposes, then decimals of hPa are even more so). It is therefore suggested to utilize 990 hPa as reference pressure or at least replace the number with a more generic "at sea-level pressure".

L323 - Short CMLs are highlighted in some following sections as valuable tools for intense rainfall detection. Here instead the sentence "The performance ..." says that they will be presented only as examples of bad performance. Please clarify.

L347 to 358 - Is there any indication of what could cause the "degraded resolution" on the hardware side? If yes, it would be an interesting topic to read here.

L365 and Fig. 6 - It should be reminded to the reader that sub-links belonging to one CML are presented in pairs in consecutive order. It should be consequently pointed out that intra-CML correlation creates 2x2 darker squares along the diagonal in the correlation matrix plot.

L369 - It is stated that the delay of the raingauges in detecting rainfall with respect to CMLs "can be attributed to the delay of rain gauge rain detection due to the filling of the bucket." Please discuss whether delays and volume losses are compatible to the bucket size.

L377 to 380 - Same as L283, the dependency (and independence) of WAA to path length should be presented for rain (and no-rain) occurrences with a specific scatterplot and a linear fit (suggested figure attached).

Fig.7 - When comparing signals from CMLs of different path length, specific attenuation (dB/km) should be preferred to pure attenuation (dB). If the aim is to show the different regimes (dependency and independence to path length), then two plots should be shown (dB and dB/km time series), in order to appreciate inter-CML concordance on specific attenuation during rainfall and on pure attenuation during dry periods.

L405 - "However, it is closer ..." the reported considerations is interesting for an operational use and therefore valuable, but it is poorly proven (only visually). Without a gauge of the goodness of the approximation (or some reference to following consistent

results), the ITU fit may as well not be good for either case (convective and stratiform).

L433 - To my understanding, it is the first time here that some specific deficits in baseline and WAA identification for sub-link 1147 are asserted. It seems quite in contradiction with other parts of the text were the long CML has the best results.

L448, 449 and Fig. 12 - The anti-correlation of the attenuation with temperature is evident from figure 12b and should be highlighted here, as temperature seems to be the dominant component of the signal. Moreover, this appears in direct contradiction with what stated in the first paragraph of Section 4.6, so that may be reformulated differently.

L515 - "The similarity in antenna characteristics was not inspected directly." Are the antenna factory features known to the authors? Is this sentence referring to technical specifications of the antennas or to the actual status of the radomes?

Supplementary material - The ATPC (5th paragraph) is said to be "switched off" but, to mine understanding of Fig. S1, the concept of "saturated" may be more adherent to the case. It seems to me that ATPC can deal only with maximum 7 dB gains on tx, but it keeps working even there, in the sense that the gain remains 7 dB, while "ATPC switched off" is more likely a zero-gain scenario.

Technical observations:

Figure 2 - It is not clear what the coloured bands represent (standard deviation or total spread) and neither is the direction from low to high pressure.

L202 and Table 2 - "Height" is used, but maybe "depth" is a more common choice to indicate precipitation amount.

Fig. 6 - Raingauge labels differ between image and caption ("wet_" prefix)

Fig. 8 - Since the two plot rows represent different frequency ranges, some labels indicating the two ranges are fostered to be shown to the left of the plot. Otherwise

this information should at least appear in the caption with "upper row" and "lower row" indications.

L413 - I suggest the replacement of "heteroscedastic" with a more generic formulation, e.g. "the spread clearly grows with R and k". Although the adjective is certainly correct for a distribution like the one shown in Fig. 10, its use seems not proper for this context: given its precise statistical meaning and implications, I think it is preferable to run some specific tests of heteroscedasticity before asserting this property.

Fig. 12 - The colours for theoretical and observed attenuations are poorly chosen as they appear very similar (especially light green against light blue), both on paper and on screen.

[Figure]

[Figure]

**Fig. 1.** Scatterplot of attenuation against pathlength with separated linear fits for rain and no-rain intervals.

---

## Author Comment (AC3) · 21 May 2020

First of all, we would like to thank the reviewer for a constructive and encouraging review. Bellow are our reactions:

*The paper by Fencl et al. addresses a topical and interesting matter, as extends known opportunistic precipitation sensing techniques to the more recent E band links. It highlights the new possibilities uncovered by the different frequencies and hardware and focuses on the consequent challenges. The authors give a complete picture of the subject from theory to application, preparing the ground for future studies. The article*

[Figure]

*is therefore certainly valuable and of primary interest to the CML scientific community and AMT readers. The work is well written and the goals defined in the abstract and introduction are all met. The discussion of the main issues is complete and rich, while some redundancy and repetitiveness is found in introductory and methodological sections, combined in certain cases with lack of the detailed quantitative information needed to contextualize some statements. Accordingly, a minor revision is suggested in order to provide the reader with more concise and relevant information in the cases treated in the comments below. The author's answers to previous comments (AC1 and AC2 to SC1 and RC1 resp.) have been taken into consideration.*

We will follow specific suggestions of the reviewer to remove identified redundancies and repetitiveness and will provide additional quantitative information where required.

**General comments**

1. *The fragmentation of the presentation as reported in comment 3. of RC1 is recognized: most of the topics are introduced in Sections 2 and 3 and then corroborated with quantitative data only in section 4 or even 5. Given the different data sets and methods utilized for the various steps of the investigation, the reading results some-times erratic indeed. However, the intentions declared by the authors (AC2) are also well understood. I will then strongly encourage a more widespread use of subsection cross referencing, to help the reader understand without changing the logical structure of the paper. An example of convenient referencing is found e.g. in L362 and 363. This should be replicated diffusely to connect introductory and discussion Sections. It seems to me that multipath disturbance instead is not introduced at all before L577 and should be added to Section 2 with some estimate of its magnitude.*

   Thank you for understanding to our intention to avoid inclusion of Results into Method and Material section. However, to make our presentation clearer we

carefully identified redundancies and use more widespread cross-referencing as suggested. In the revised version of the manuscript, we also introduced multipath disturbance (already in the Section 2.1, where different components of total observed loss are introduced).

2. *Another downside of the chosen presentation layout is the need of re-introducing some aspects generally many times throughout the paper, without going quickly into the necessary detail. A more concise and unitary approach to the problems encountered and the solutions adopted would facilitate a global understanding of the work. I suggest therefore to support the introductory informations, in the first sections already, with quantitative informations and stating author's intentions regarding approximations and further discussions. In that way the reader could expect what to find in the next sections and repetitive recalls to the qualitative introduction would not be needed. Some non-exhaustive examples are reported below and most of the specific comments deal with this same issue. L74 to 78 - though the paragraph's introductory intent is clear, it lacks the detail and clearness about which assumptions are kept and which are discussed, with respect to previous 15-40 GHz approaches. L94 to 101 - It is not clear at this point how the authors will deal with the reported considerations further in the paper.*

Our intention is to avoid inclusion of our original findings in section 1 and section 2. Section 1 provide general introduction with state-of-the art in microwave link rainfall estimation based upon which the goals of this manuscript are defined. Section 2 provides theoretical background enabling reader to follow our original methodology and results. We would like to keep our original methodology and findings clearly separated and thus we want to avoid summary of our original findings already in the introductory sections. Similarly, we would like to keep our original methodology separated from theoretical background provided by previous works (section 2). Thus, although concept of baseline separation is introduced already in the Section 2 (L74-78) we prefer to explain how we approach this challenge in Section 3 - Material and Methods, specifically on lines L266 – L271. Following the same intention, we prefer not to explain assumptions behind quantifying wet antenna attenuation in this work already at L94 – 101, but again in the Section 3, specifically on lines L273-284. To clarify this intention, we will modify the paragraph describing structure of the manuscript, specifically description of section 2 and 3: "Section 2 of the manuscript summarizes based upon previous works the principles behind retrieving atmospheric variables from CML observations, Section 3 describes the methodology and datasets used in this manuscript for the E-band CML assessment, ..."

3. *An additional figure showing WAA against link length could be used to illustrate the linear regressions proposed in Eq. 11 and the constant behaviour in dew cases. A sample of how the figure could look is attached.:*

   Agreed. We will show such figure. Details are provided in the specific comment no. 11.

**Specific comments (in order of appearance)**

1. *L68 - Free space loss (Lbf) is said to be uniquely defined by distance and wavelength. Reporting the formula could be appropriate and helpful for further understanding of the discussion, as the frequency is a key variable for this study (E band)*

   Yes. We will report the formula of free space loss on lines L68-69 of the original manuscript. "Free space loss ($L_{bf}$) is uniquely defined by the distance ($d$) between the transmitter and receiver, and by wavelength ($\lambda$):

   $$L_{bf} = 20log\left(\frac{4\pi d}{\lambda}\right)$$

where $L_b f$ is expressed in dB and distance $d$ and wavelength $\lambda$ are expressed in the same unit."

2. *L74-78 - The phrase "Attenuation during dry weather is assumed to be a baseline" is apparently in direct contrast with the following "Fluctuations in the baseline during dry weather can be attributed..." if the reader does not know already the different magnitudes involved. Early introduction of orders of magnitude and average behaviours is therefore encouraged.*

   Agreed. We will report typical magnitudes of rainfall and gaseous attenuation in the section 2.1, after description of different components of total observed loss (L74 in the original manuscript).

3. *L101 - "More extensive investigations..." I think this sentence will state the motivation of the author's work, but it could be also interpreted as what still remains unknown after the work's results instead. Please clarify to avoid this ambivalence.*

   The sentence indeed state our motivation. We will try to make it clearer by expressing at the end of the Introduction section, where structure of the manuscript is described, that section 2 provides review of previous work (see response 2 in the general comments).

4. *L131 - Fig. 4 is useful to the contextualisation of this sentence and should be referenced. "Contribute relatively less" is not gaugeable, some more detail may be added.*

   Agreed. We will reference Figure 4 at line 132 of the original manuscript. Nonetheless, we kindly disagree with the second suggestion. We would like to avoid detailed quantitative description in here. Reader can easily read quantitative information from the figure 1 referenced in this sentence.

5. *L145 and following - The study on the components of N is not justified by following discussion or results and could be omitted as it lacks quantitative information.*

*I think that the qualitative concept of the dependency of k to the various components is already well stressed.*

Agreed. As suggested, study on the components of N will be omitted. Interested reader can find these details in the cited literature. We will thus remove lines 145-150 of the original manuscript.

6. *L194 - "The periods for evaluating rainfall retrieval and for evaluating the effect of humidity and temperature fluctuations on gaseous attenuation are, therefore, different." The phrase itself is a quite obvious consequence of the previous sentence, while its implications are not. It should either be omitted or some expected implications should also be discussed (or at least some reference to the respective discussion should be made) in terms e.g. of which investigations are precluded by using different time windows.*

Agreed. The phrase is obvious and we will thus delete it.

7. *L200 and other appearances of "aggregate" - it should be pointed out how the aggregation to different time scales is performed (mean, median, sum, max, other...)*

We aggregate to different time scales using mean. We will add this information to the corresponding places (L198, L200, and 205 in the original manuscript).

8. *L246 and 247 - The sentence is not clear and should be rephrased and expanded. "dependent" should perhaps be substituted with "depending", commas before and after "therefore" are not necessary and slow the reading. The threshold for Dm is not indicated.*

Thank you, for spotting this typo. This typo apparently led to misunderstanding. $D_m$ as estimated by Eq. 10 is actually the threshold which is used for classifying rainfalls. This threshold ($D_m$) is dependent on rainfall intensity. We will change the sentence to: 'The approximation (10) is used to calculate threshold for classifying disdrometer records as convective or stratiform. The threshold is dependent

on rainfall intensity.

9. *L259 - Visual inspection does not seem like a robust approach to filter the outliers. Some technique should be at least suggested to cope with this kind of artefacts, as the visual approach is clearly not feasible at larger and near real-time scales.*

Visual check is indeed not a robust approach which could be used in future applications. The automation of quality check is, however, out the scope of this manuscript. Visual identification of artifacts is, in our view, first step towards future automation of this process. Moreover, the correction for artifacts is performed only in a single case. This correction is transparently reported (L260), to ensure reproducibility of the results.

10. *L269 - One-week sized moving window "is sufficiently short": are baseline drifts proven to happen only at longer time scales? Is the same for gaseous attenuation? Could it be that some higher frequency signal is masked by this approach resulting in the weakening of the water vapour detection capabilities?*

No, as reported on L260-261, also sudden change in baseline occur in the case of CML 3004_3005 and this change was manually corrected. The baseline identification using one-week sized moving window is used only for rainfall retrieval. As reported on L305-306 of the original manuscript, constant baseline is used when analyzing effect of gaseous attenuation and potential for water vapor retrieval. Hardware related artifacts causing slow baseline drift have probably potential to destroy gaseous attenuation signal as discussed in the Discussion section (L581-584 in the original manuscript).

11. *L283 - A reference to Fig. 8 or to the suggested new figure could be added here.*

OK. We will add a reference to the Figure 8 (created in the revised manuscript according to reviewer's suggestion). The figure is shown at the end of this response (Fig. 2). It depicts the period from 19:00 on 2$^{nd}$ Nov to 14:00 on 3$^{rd}$ Nov which is indicated as rainy on Figure 7 of the original manuscript.

12. *Fig.7 - When comparing signals from CMLs of different path length, specific attenuation (dB/km) should be preferred to pure attenuation (dB). If the aim is to show the different regimes (dependency and independence to path length), then two plots should be shown (dB and dB/km time series), in order to appreciate inter-CML concordance on specific attenuation during rainfall and on pure attenuation during dry periods.*

    The signal shown in the figure 7 is predominantly caused by wet antenna attenuation, which is independent of path length. We therefore prefer to show exclusively total attenuation in this figure.

13. *L405 - "However, it is closer ..." the reported considerations is interesting for an operational use and therefore valuable, but it is poorly proven (only visually). Without a gauge of the goodness of the approximation (or some reference to following consistent results), the ITU fit may as well not be good for either case (convective and stratiform).*

    The attenuation-rainfall relation is for theoretical drop size distribution almost perfectly approximated by power-law fits (as reported on L400-401). Thus, distances between presented power-law curves (absolute errors) provide meaningful gauge of goodness. The term "it is closer" on L405 describe distances between the curves. Thus, it is, in our opinion, appropriate. Moreover, reader can easily get information on approximate distances for any rainfall intensity between 0-50 mm/h from the figure 9. In addition, parameters of power-law fits as well as parameters obtained from ITU (ITU-R, 2005) are provided as a part of figure 9. Interested reader can thus easily express exact value of absolute errors for any rainfall intensity, resp. specific attenuation.

14. *L433 - To my understanding, it is the first time here that some speciific deficits in baseline and WAA identification for sub-link 1147 are asserted. It seems quite in contradiction with other parts of the text were the long CML has the best results.*

The longest CML clearly outperform shorter CMLs in terms of correlation (r = 0.96 resp. 0.97 compared to 0.53 – 0.86 resp. 0.61 – 0.87) to and RMSE (0.39 resp. 0.24 mm h-1compared to 0.64 – 2.18 resp. 0.69 – 1.44 mm h$^{-1}$), which can be seen in table 5 of the original manuscript. Its markedly better performance is also clearly visible from scatter plots in figure 11 of the original manuscript. The long CML cannot, however, accurately capture very light rainfalls under 1 mm h$^{-1}$, which represent about 25 % of the total rainfall depth in our case. We will add an information about underestimation of very light rainfalls to the paragraph (L428-L436 of the original manuscript) describing performance of the long CML.

15. *L448, 449 and Fig. 12 - The anti-correlation of the attenuation with temperature is evident from figure 12b and should be highlighted here, as temperature seems to be the dominant component of the signal. Moreover, this appears in direct contradiction with what stated in the first paragraph of Section 4.6, so that may be reformulated differently.*

The negative correlation between attenuation and temperature appears in the figure, because water vapor density is strongly correlated with temperature. As gaseous attenuation is highly correlated to water vapor density, there is also strong (negative) correlation link between gaseous attenuation and temperature. It is, however, not caused by direct dependence, which is almost negligible: See ITU-R, (2019) and Figure 2 of the original manuscript.

16. *L515 - "The similarity in antenna characteristics was not inspected directly." Are the antenna factory features known to the authors? Is this sentence referring to technical specifications of the antennas or to the actual status of the radomes?*

It will be specified. The sentence refers to hydrophobic properties of antenna radomes as well as actual status of the radomes.

17. *Supplementary material - The ATPC (5th paragraph) is said to be "switched off" but, to mine understanding of Fig. S1, the concept of "saturated" may be more*

*adherent to the case. It seems to me that ATPC can deal only with maximum 7 dB gains on tx, but it keeps working even there, in the sense that the gain remains 7 dB, while "ATPC switched off" is more likely a zero-gain scenario.*

Yes, the ATPC keeps working in the sense it maintains tx power on the maximal (allowed) level. In the revised version of the supplementary material, we will use the term 'saturated' instead 'switched off'.

**Technical observations:**

1. *Figure 2 - It is not clear what the coloured bands represent (standard deviation or total spread) and neither is the direction from low to high pressure.*

   The color bands represent total spread. We will clarify this in the figure caption.

2. *L202 and Table 2 - "Height" is used, but maybe "depth" is a more common choice to indicate precipitation amount.*

   Agreed. We will use the term 'depth'.

3. *Fig. 6 - Raingauge labels differ between image and caption ("wet_" prefix)*

   Thank you for spotting this inconsistency. We will correct it.

4. *Fig. 8 - Since the two plot rows represent different frequency ranges, some labels indicating the two ranges are fostered to be shown to the left of the plot. Otherwise this information should at least appear in the caption with "upper row" and "lower row" indications.*

   We will add to the left two labels indicating frequency ranges.

5. *L413 - I suggest the replacement of "heteroscedastic" with a more generic formulation, e.g. "the spread clearly grows with R and k". Although the adjective is*

*certainly correct for a distribution like the one shown in Fig. 10, its use seems not proper for this context: given its precise statistical meaning and implications, I think it is preferable to run some specific tests of heteroscedasticity before asserting this property.*

Done.

6. *Fig. 12 - The colours for theoretical and observed attenuations are poorly chosen as they appear very similar (especially light green against light blue), both on paper and on screen.*

   OK, we will adjust the colors in Figure 12 to differentiate better the time series.

**Additional references**

ITU-R: ITU-R P.676-11, [online] Available from: https://www.itu.int/dms_pubrec/itu-r/rec/p/R-REC-P.676-11-201609-I!!PDF-E.pdf, 2016.

ITU-R: RECOMMENDATION ITU-R P.838-3 - Specific attenuation model for rain for use in prediction methods, (online) Available from: https://www.itu.int/dms_pubrec/itu-r/rec/p/R-REC-P.838-3-200503-I!!PDF-E.pdf, 2005.

———————————————

[Figure]

**Fig. 1.** Scatterplot of attenuation against path length with separated linear fits for rain and no-rain intervals (reviewer's suggestion).

Fig. 2. Scatterplot of attenuation against path length for 83-84 GHz sub-links with separated
linear fits for intervals with moderate rainfall and intervals with very-light rainfall with dry spells.

---

## Author Comment (AC4) · 24 May 2020

This is a revised version of our response. Our reactions to six specific comments has not been by accident included into the previous version. We would like to apologize for this mistake and provide here an updated version with complete list of responses.

First of all, we would like to thank the reviewer for a constructive and encouraging review. Bellow are our reactions:

*The paper by Fencl et al. addresses a topical and interesting matter, as extends known*

[Figure]

*opportunistic precipitation sensing techniques to the more recent E band links. It highlights the new possibilities uncovered by the different frequencies and hardware and focuses on the consequent challenges. The authors give a complete picture of the subject from theory to application, preparing the ground for future studies. The article is therefore certainly valuable and of primary interest to the CML scientific community and AMT readers. The work is well written and the goals defined in the abstract and introduction are all met. The discussion of the main issues is complete and rich, while some redundancy and repetitiveness is found in introductory and methodological sections, combined in certain cases with lack of the detailed quantitative information needed to contextualize some statements. Accordingly, a minor revision is suggested in order to provide the reader with more concise and relevant information in the cases treated in the comments below. The author's answers to previous comments (AC1 and AC2 to SC1 and RC1 resp.) have been taken into consideration.*

We will follow specific suggestions of the reviewer to remove identified redundancies and repetitiveness and will provide additional quantitative information where required.

**General comments**

1. *The fragmentation of the presentation as reported in comment 3. of RC1 is recognized: most of the topics are introduced in Sections 2 and 3 and then corroborated with quantitative data only in section 4 or even 5. Given the different data sets and methods utilized for the various steps of the investigation, the reading results some-times erratic indeed. However, the intentions declared by the authors (AC2) are also well understood. I will then strongly encourage a more widespread use of subsection cross referencing, to help the reader understand without changing the logical structure of the paper. An example of convenient referencing is found e.g. in L362 and 363. This should be replicated diffusely to connect introductory and discussion Sections. It seems to me that multipath*

*disturbance instead is not introduced at all before L577 and should be added to Section 2 with some estimate of its magnitude.*

Thank you for understanding to our intention to avoid inclusion of Results into Method and Material section. However, to make our presentation clearer we carefully identified redundancies and use more widespread cross-referencing as suggested. In the revised version of the manuscript, we also introduced multipath disturbance (already in the Section 2.1, where different components of total observed loss are introduced).

2. *Another downside of the chosen presentation layout is the need of re-introducing some aspects generally many times throughout the paper, without going quickly into the necessary detail. A more concise and unitary approach to the problems encountered and the solutions adopted would facilitate a global understanding of the work. I suggest therefore to support the introductory informations, in the first sections already, with quantitative informations and stating author's intentions regarding approximations and further discussions. In that way the reader could expect what to find in the next sections and repetitive recalls to the qualitative introduction would not be needed. Some non-exhaustive examples are reported below and most of the specific comments deal with this same issue. L74 to 78 - though the paragraph's introductory intent is clear, it lacks the detail and clearness about which assumptions are kept and which are discussed, with respect to previous 15-40 GHz approaches. L94 to 101 - It is not clear at this point how the authors will deal with the reported considerations further in the paper.*

Our intention is to avoid inclusion of our original findings in section 1 and section 2. Section 1 provide general introduction with state-of-the art in microwave link rainfall estimation based upon which the goals of this manuscript are defined. Section 2 provides theoretical background enabling reader to follow our original methodology and results. We would like to keep our original methodology and findings clearly separated and thus we want to avoid summary of our

original findings already in the introductory sections. Similarly, we would like to keep our original methodology separated from theoretical background provided by previous works (section 2). Thus, although concept of baseline separation is introduced already in the Section 2 (L74-78) we prefer to explain how we approach this challenge in Section 3 - Material and Methods, specifically on lines L266 – L271. Following the same intention, we prefer not to explain assumptions behind quantifying wet antenna attenuation in this work already at L94 – 101, but again in the Section 3, specifically on lines L273-284. To clarify this intention, we will modify the paragraph describing structure of the manuscript, specifically description of section 2 and 3: "Section 2 of the manuscript summarizes based upon previous works the principles behind retrieving atmospheric variables from CML observations, Section 3 describes the methodology and datasets used in this manuscript for the E-band CML assessment, . . ."

3. *An additional figure showing WAA against link length could be used to illustrate the linear regressions proposed in Eq. 11 and the constant behaviour in dew cases. A sample of how the figure could look is attached.:*

Agreed. We will show such figure. Details are provided in the specific comment no. 11.

**Specific comments (in order of appearance)**

1. *L68 - Free space loss (Lbf) is said to be uniquely defined by distance and wavelength. Reporting the formula could be appropriate and helpful for further understanding of the discussion, as the frequency is a key variable for this study (E band)*

Yes. We will report the formula of free space loss on lines L68-69 of the original manuscript. "Free space loss $(L_{bf})$ is uniquely defined by the distance $(d)$ between the transmitter and receiver, and by wavelength $(\lambda)$:

$$L_{bf} = 20log\left(\frac{4\pi d}{\lambda}\right)$$

where $L_{bf}$ is expressed in decibels and distance $d$ and wavelength $\lambda$ are expressed in the same unit."

2. *L74-78 - The phrase "Attenuation during dry weather is assumed to be a baseline" is apparently in direct contrast with the following "Fluctuations in the baseline during dry weather can be attributed..." if the reader does not know already the different magnitudes involved. Early introduction of orders of magnitude and average behaviours is therefore encouraged.*

Agreed. We will report typical magnitudes of rainfall and gaseous attenuation in the section 2.1, after description of different components of total observed loss (L74 in the original manuscript).

3. *L101 - "More extensive investigations..." I think this sentence will state the motivation of the author's work, but it could be also interpreted as what still remains unknown after the work's results instead. Please clarify to avoid this ambivalence.*

The sentence indeed state our motivation. We will try to make it clearer by expressing at the end of the Introduction section, where structure of the manuscript is described, that section 2 provides review of previous work (see response 2 in the general comments).

4. *L131 - Fig. 4 is useful to the contextualisation of this sentence and should be referenced. "Contribute relatively less" is not gaugeable, some more detail may be added.*

Agreed. We will reference Figure 4 at line 132 of the original manuscript. Nonetheless, we kindly disagree with the second suggestion. We would like to

avoid detailed quantitative description in here. Reader can easily read quantitative information from the figure 1 referenced in this sentence.

5. *L145 and following - The study on the components of N is not justified by following discussion or results and could be omitted as it lacks quantitative information. I think that the qualitative concept of the dependency of k to the various components is already well stressed.*

Agreed. As suggested, study on the components of N will be omitted. Interested reader can find these details in the cited literature. We will thus remove lines 145-150 of the original manuscript.

6. *L194 - "The periods for evaluating rainfall retrieval and for evaluating the effect of humidity and temperature fluctuations on gaseous attenuation are, therefore, different." The phrase itself is a quite obvious consequence of the previous sentence, while its implications are not. It should either be omitted or some expected implications should also be discussed (or at least some reference to the respective discussion should be made) in terms e.g. of which investigations are precluded by using different time windows.*

Agreed. The phrase is obvious and we will thus delete it.

7. *L200 and other appearances of "aggregate" - it should be pointed out how the aggregation to different time scales is performed (mean, median, sum, max, other...)*

We aggregate to different time scales using mean. We will add this information to the corresponding places (L198, L200, and 205 in the original manuscript).

8. *L246 and 247 - The sentence is not clear and should be rephrased and expanded. "dependent" should perhaps be substituted with "depending", commas before and after "therefore" are not necessary and slow the reading. The threshold for Dm is not indicated.*

Thank you, for spotting this typo. This typo apparently led to misunderstanding. $D_m$ as estimated by Eq. 10 is actually the threshold which is used for classifying rainfalls. This threshold ($D_m$) is dependent on rainfall intensity. We will change the sentence to: 'The approximation (10) is used to calculate threshold for classifying disdrometer records as convective or stratiform. The threshold is dependent on rainfall intensity.

9. *L259 - Visual inspection does not seem like a robust approach to filter the outliers. Some technique should be at least suggested to cope with this kind of artefacts, as the visual approach is clearly not feasible at larger and near real-time scales.*

   Visual check is indeed not a robust approach which could be used in future applications. The automation of quality check is, however, out the scope of this manuscript. Visual identification of artifacts is, in our view, first step towards future automation of this process. Moreover, the correction for artifacts is performed only in a single case. This correction is transparently reported (L260), to ensure reproducibility of the results.

10. *L269 - One-week sized moving window "is sufficiently short": are baseline drifts proven to happen only at longer time scales? Is the same for gaseous attenuation? Could it be that some higher frequency signal is masked by this approach resulting in the weakening of the water vapour detection capabilities?*

   No, as reported on L260-261, also sudden change in baseline occur in the case of CML 3004_3005 and this change was manually corrected. The baseline identification using one-week sized moving window is used only for rainfall retrieval. As reported on L305-306 of the original manuscript, constant baseline is used when analyzing effect of gaseous attenuation and potential for water vapor retrieval. Hardware related artifacts causing slow baseline drift have probably potential to destroy gaseous attenuation signal as discussed in the Discussion section (L581-584 in the original manuscript).

11. *L283 - A reference to Fig. 8 or to the suggested new figure could be added here.*

    Agreed. We will add a reference to the new Figure (created in the revised manuscript according to reviewer's suggestion). The figure is shown at the end of this response (Fig. 2). It depicts the period from 19:00 on 2nd Nov. to 14:00 on 3rd Nov. which is on Figure 7 of the original manuscript indicated as rainy.

12. *L296 and 340 - Since Prague is located at an altitude around 200m (990 hPa), to as-sume the atmospheric pressure of 1013.25 hPa seems either systematically wrong or reported with too high precision (if differences between 990 and 1013 are negligible for the author purposes, then decimals of hPa are even more so). It is therefore suggested to utilize 990 hPa as reference pressure or at least replace the number with a more generic "at sea-level pressure".*

    Agreed. We will replace the number with more generic "sea-level pressure" at L296 and 340.

13. *L323 - Short CMLs are highlighted in some following sections as valuable tools for intense rainfall detection. Here instead the sentence "The performance ..." says that they will be presented only as examples of bad performance. Please clarify.*

    Potential of short CMLs is highlighted in the Discussion section (L556 -562 of the original manuscript) in the context of observing heavy rainfalls associated with high spatial variability by which an assumption about uniform rainfall distribution along a CML path is more likely valid for short CMLs than for long ones. Nevertheless, only light and moderate rainfalls occurred during observation period. Short CMLs are, during these rainfalls relatively more affected by wet antenna attenuation than longer CMLs, as demonstrated e.g. on figure 7 in the original manuscript. The sentence, in the original manuscript actually states that: 'The performance of the short CMLs is shown to demonstrate limitations related to the improper baseline and WAA identification which are more pronounced by shorter

CMLs.' This is in our view, not in contradiction with Discussion section, where shorter CMLs are suggested as valuable tool for detecting heavy rainfalls.

To avoid misunderstanding, we will modify the sentence to: 'The performance of the short CMLs is shown to demonstrate limitations related to the improper baseline and WAA identification which are, especially during light rainfalls, more pronounced by shorter CMLs.'

14. *L347 to 358 - Is there any indication of what could cause the "degraded resolution" on the hardware side? If yes, it would be an interesting topic to read here.*

In our opinion, the degraded resolution might be related to automatic power control. Nevertheless, this was not tested. We thus prefer to not speculate in this direction.

15. *L365 and Fig. 6 - It should be reminded to the reader that sub-links belonging to one CML are presented in pairs in consecutive order. It should be consequently pointed out that intra-CML correlation creates 2x2 darker squares along the diagonal in the correlation matrix plot.*

We have decided to change IDs of CMLs in the whole manuscript to better indicate sub-links belonging to the same CML and in general improve clarity of the whole manuscript: each CML will have unique ID (numbers from 1 to 6). IDs of sub-links operating at 73 – 74 GHz and 83 – 84 GHz frequency will then consist of CML ID and suffix "a" resp. "b".

Regarding intra-CML correlation, we believe, that the first sentence referring to the Figure 6 (L365 of the original manuscript) will then provide sufficient guidance in this respect: "Dry-wet weather classifiers of single sub-links belonging to one CML are strongly correlated (Fig. 6)."

16. *L369 - It is stated that the delay of the rain gauges in detecting rainfall with respect to CMLs "can be attributed to the delay of rain gauge rain detection due to*

*the filling of the bucket." Please discuss whether delays and volume losses are compatible to the bucket size.*

Thank you for this comment. We will remove last two sentences (L366-367) as they were based on analysis not shown in the manuscript and might have been considered speculative. The section will thus contain only results shown on figure 6 of the original manuscript.

17. *L377 to 380 - Same as L283, the dependency (and independence) of WAA to path length should be presented for rain (and no-rain) occurrences with a specific scatterplot and a linear fit (suggested figure attached).*

Agreed. Figure 7 will be extended by a scatter plot showing relation between path length and total attenuation together with linear fits indicating effect of WAA. (The figure is shown at the end of this document).

18. *Fig.7 - When comparing signals from CMLs of different path length, specific attenuation (dB/km) should be preferred to pure attenuation (dB). If the aim is to show the different regimes (dependency and independence to path length), then two plots should be shown (dB and dB/km time series), in order to appreciate inter-CML concordance on specific attenuation during rainfall and on pure attenuation during dry periods.*

The signal shown in the figure 7 is predominantly caused by wet antenna attenuation, which is independent of path length. We therefore prefer to show exclusively total attenuation in this figure.

19. *L405 - "However, it is closer ..." the reported considerations is interesting for an operational use and therefore valuable, but it is poorly proven (only visually). Without a gauge of the goodness of the approximation (or some reference to following consistent results), the ITU fit may as well not be good for either case (convective and stratiform).*

The attenuation-rainfall relation is for theoretical drop size distribution almost perfectly approximated by power-law fits (as reported on L400-401). Thus, distances between presented power-law curves (absolute errors) provide meaningful gauge of goodness. The term "it is closer" on L405 describe distances between the curves. Thus, it is, in our opinion, appropriate. Moreover, reader can easily get information on approximate distances for any rainfall intensity between 0-50 mm/h from the figure 9. In addition, parameters of power-law fits as well as parameters obtained from ITU (ITU-R, 2005) are provided as a part of figure 9. Interested reader can thus easily express exact value of absolute errors for any rainfall intensity, resp. specific attenuation.

20. *L433 - To my understanding, it is the first time here that some speciific deficits in baseline and WAA identification for sub-link 1147 are asserted. It seems quite in contradiction with other parts of the text were the long CML has the best results.*

    The longest CML clearly outperform shorter CMLs in terms of correlation (r = 0.96 resp. 0.97 compared to 0.53 – 0.86 resp. 0.61 – 0.87) to and RMSE (0.39 resp. 0.24 mm h-1compared to 0.64 – 2.18 resp. 0.69 – 1.44 mm h$^{-1}$), which can be seen in table 5 of the original manuscript. Its markedly better performance is also clearly visible from scatter plots in figure 11 of the original manuscript. The long CML cannot, however, accurately capture very light rainfalls under 1 mm h$^{-1}$, which represent about 25 % of the total rainfall depth in our case. We will add an information about underestimation of very light rainfalls to the paragraph (L428-L436 of the original manuscript) describing performance of the long CML.

21. *L448, 449 and Fig. 12 - The anti-correlation of the attenuation with temperature is evident from figure 12b and should be highlighted here, as temperature seems to be the dominant component of the signal. Moreover, this appears in direct contradiction with what stated in the first paragraph of Section 4.6, so that may be reformulated differently.*

The negative correlation between attenuation and temperature appears in the figure, because water vapor density is strongly correlated with temperature. As gaseous attenuation is highly correlated to water vapor density, there is also strong (negative) correlation link between gaseous attenuation and temperature. It is, however, not caused by direct dependence, which is almost negligible: See ITU-R, (2019) and Figure 2 of the original manuscript.

22. *L515 - "The similarity in antenna characteristics was not inspected directly." Are the antenna factory features known to the authors? Is this sentence referring to technical specifications of the antennas or to the actual status of the radomes?*

   It will be specified. The sentence refers to hydrophobic properties of antenna radomes as well as actual status of the radomes.

23. *Supplementary material - The ATPC (5th paragraph) is said to be "switched off" but, to mine understanding of Fig. S1, the concept of "saturated" may be more adherent to the case. It seems to me that ATPC can deal only with maximum 7 dB gains on tx, but it keeps working even there, in the sense that the gain remains 7 dB, while "ATPC switched off" is more likely a zero-gain scenario.*

   Yes, the ATPC keeps working in the sense it maintains tx power on the maximal (allowed) level. In the revised version of the supplementary material, we will use the term 'saturated' instead 'switched off'.

**Technical observations:**

1. *Figure 2 - It is not clear what the coloured bands represent (standard deviation or total spread) and neither is the direction from low to high pressure.*

   The color bands represent total spread. We will clarify this in the figure caption.

2. *L202 and Table 2 - "Height" is used, but maybe "depth" is a more common choice to indicate precipitation amount.*

Agreed. We will use the term 'depth'.

3. *Fig. 6 - Raingauge labels differ between image and caption ("wet_" prefix)*

Thank you for spotting this inconsistency. We will correct it.

4. *Fig. 8 - Since the two plot rows represent different frequency ranges, some labels indicating the two ranges are fostered to be shown to the left of the plot. Otherwise this information should at least appear in the caption with "upper row" and "lower row" indications.*

We will add to the left two labels indicating frequency ranges.

5. *L413 - I suggest the replacement of "heteroscedastic" with a more generic formulation, e.g. "the spread clearly grows with R and k". Although the adjective is certainly correct for a distribution like the one shown in Fig. 10, its use seems not proper for this context: given its precise statistical meaning and implications, I think it is preferable to run some specific tests of heteroscedasticity before asserting this property.*

Done.

6. *Fig. 12 - The colours for theoretical and observed attenuations are poorly chosen as they appear very similar (especially light green against light blue), both on paper and on screen.*

OK, we will adjust the colors in Figure 12 to differentiate better the time series.

**Additional references**

ITU-R: RECOMMENDATION ITU-R P.676-12 - Attenuation by atmospheric gases and related effects, (online) Available from: https://www.itu.int/dms_pubrec/itu-r/rec/p/R-REC-P.676-12-201908-I!!PDF-E.pdf, 2019.

ITU-R: RECOMMENDATION ITU-R P.838-3 - Specific attenuation model for rain for use in prediction methods, (online) Available from: https://www.itu.int/dms_pubrec/itu-r/rec/p/R-REC-P.838-3-200503-I!!PDF-E.pdf, 2005.
* * *

**Fig. 1.** Scatterplot of attenuation against path length with separated linear for rain and no-rain intervals (reviewer's suggestion).

**Fig. 2.** Scatterplot of attenuation against path length for 83-84 GHz sub-links with separated linear for intervals with moderate rainfall and intervals with very-light rainfall with dry spells.

---

## Author Response (AR1)

We would like to thank again both referees and Dr. Guyot for their valuable comments and suggestions. Bellow, are our responses to the comments together with changes in the revised manuscript. The marked-up version of the manuscript, attached at the end of this final response, does not include changes in the structure (order of subsections). The revisions in the structure are reported separately instead in Section 4 of this response.

**1. Response to dr. Guyot**

**General comments**

*1) The introduction covers most important aspects but it omits to mention that attenuation by rainfall at these frequencies (E-Band) has already been investigated experimentally, e.g. by Shresta and Choi (2017) or Norouzian et al. (2020) for instance (and maybe other papers). This literature is often going under the radar of the atmospheric community maybe because it is published in specific engineering and electronics journals (often IEEE). Yes, the objective of these studies was the optimal design of back-haul networks, as to minimise the occurrence of signal fading, instead of opportunistic measurement of rainfall. But nethertheless, experimentally this is the same setup and collected data, and often even formulations and theoretical approaches. I would include a mention of their existence in the introduction, and eventually in the discussion if appropriate.*

This is an interesting point. We are aware of E-band investigation being reported in the electrical engineering journals (often IEEE) and refer to two examples of this work in the Introduction (Hansryd et al., 2010; Luini et al., 2018) and two other examples in the section about wet antenna attenuation (section 2.4) (Hong et al., 2017; Ostrometzky et al., 2018). The investigations of Hansryd et al. (2010) and Luini et al. (2018) have similar scope as the papers suggested by Dr. Guyot. We have been considering also referring to other works, nevertheless, the propagation studies aiming at community designing microwave link networks are mostly focused on long-term availability and lack evaluation in shorter time scales, which is critical for evaluating retrieval of atmospheric variables. We thus believe, that our relatively concise list of references on radio engineering investigations suffices as it, first, covers most important topics being previously investigated and, secondly, can provide further reading through referenced or citing studies. On the other hand, we might have missed some important point in the suggested references or some topic in the radio engineering literature and will be thus happy for a notice.

*2) L105: Which parameters values for the canting angle and temperature, and which model did you use for the T-Matrix calculations? You have to specify it here.*

Thank you for notifying us of missing piece of information. We have provided this information in the revised manuscript in section 3.4:

"The extinction cross-sections used in Eq. (5) are calculated using Python implementation of T-matrix model (Leinonen, 2014). The calculation assumes temperature 10° C, canting angle 0°, drop shape being oblate spheroid, drop-axis ratio according to Pruppacher and Beard, (1970), and for drops smaller than 0.5 mm heuristic approximation of Pruppbacher and Beard formula is used."

*3) Effect of the k-R relation on the retrievals. Those are interesting results but I think the paper abstract does not reflect the actual results of this section. Typically, in Table 5: the performance criteria are actually quite similar between the use of ITU parameters and DSD parameters. Often only 0.01 differs for R2, and RMSE rarely exceeds 0.05mm h-1. So only based on this, one might think the parameters used are not of real matter. But when looking at Table 4 and the parameters per rain type (stratiform and*

*convective, low and moderate rain rates) one sees the discrepancies.  So it is the k-R relations for local DSD and differentiation for specific rain classes/types being used that has an impact on the retrievals at these frequencies.  When lumped all together, ITU or specific DSD actually lead to similar retrievals outcomes. As the authors noted, including larger rain rates in the dataset would increase the differences in retrievals and the importance of the DSD on the retrievals at these rain rates.*

The conclusions about sensitivity to drop size distribution are based on theoretical evaluation using attenuation and rainfall calculated from drop size distribution data. The effect of DSD is also demonstrated on real commercial microwave links (CMLs), nevertheless, other sources of errors, especially wet antenna attenuation (WAA) affect these results significantly. Moreover, our results are affected by the absence of heavy rainfalls during the experimental period (see Discussion section L589 – 595 of the original manuscript). This is the reason why only the longest CML is suitable for demonstrating DSD related errors. The shorter CMLs are overly affected by WAA during light rainfalls to enable interpretations on DSD errors. We might stress this also when referring to Table 5 or at another place in the manuscript to avoid confusion.
We are, however, convinced that results from the long CML are capable of demonstrating benefit of using parameters specifically derived for stratiform rainfalls. Although the improvement in the metrics is not large in absolute values it is significant in relative values. E.g. decrease in RMSE from 0.49 to 0.39 mm h$^{-1}$ by 73.5 GHz sub-link and from 0.31 to 0.24 mm h$^{-1}$ by 83.5 GHz sub-link is decreased by 20 % resp. 23 %. This is even more pronounced in relative error where improvement from -0.44 to -0.35 by 73.5 GHz sub-link and -0.17 to -0.08 by 83.5 GHz sub-link is an improvement by 20 % resp. 53 %.

Also based on comments of referee 2, we have revised Rainfall estimation subsection (now 4.4) and extended description of the results obtained from CMLs:

"The model with DSD-derived parameters improves performance with respect to all metrics. Sub-link 1147 1a also remains significantly underestimated with DSD-derived parameters. This is due to deficits in the baseline and WAA identification. The underestimation is pronounced especially during very light rainfalls with rainfall intensities under 1 mm h$^{-1}$, which represent 25% of total rainfall dept. Shorter CMLs are less sensitive to rainfall along their shorter path and are more affected by deficiencies in the estimated baseline and WAA. Thus, use of DSD parameters does not significantly improve performance of short CMLs. CMLs shorter than 1 km overestimate rainfall intensities more than longer CMLs. "

*4) At these frequencies, longer CML path lengths translates in higher sensitivities to light rainfall – but it also means a coarser spatial resolution for CML rainfall maps – this can be a drawback and can be highlighted.*

The higher sensitivity of longer CMLs to total rainfall attention is intrinsic to all frequency regions and the investigation of rainfall reconstruction goes beyond the scope of this study. Investigation addressing specifically CML rainfall reconstruction errors is presented for example in Rios Gaona et al. (2015).

**Specific technical comments**

We would like to thank Dr. Guyot for tracking our manuscript for typos and technical shortcomings as missing units or in several cases improper format of references. We have addressed all of them as suggested.

*1) L52 Comma missing before "which"*
Added.

*2) L115 units of v(D)?*
*v(D)* (m). Added.

*3) L130 units of these variables?*
Both are in cm$^2$.

*4) L145 why not calling it the "Liebe model" as you did for the "Leijnse model"?*
*OK*. Corrected.

*5) Table 5: Instead of "Performance", use "performance metrics", or "evaluation criteria"?*

*6)There are a couple of instances in the paper when the citation format is not suitable, in particular l137 "...in (Liebe et al., 1993)..." should be "in Liebe et al. (1993)". L166 has the same issue. L226 has the same issue. L263 has the same problem. L197 has the same issue.*
Corrected.

*7) "Parsivel" is an acronym and therefore should be written in Upper case letters, e.g. PARSIVEL*
Corrected.

*8) L238 Dm is in mm*
Yes. Corrected.

*9) Figure 4 (b) there is a typo for "convective". The legend has the same citation format issue (brackets should only be around the date).*
Corrected.

*10) L265 and 265, keep a consistent spacing between value and units (\%). I think there should be a space.*
Corrected.

*11) L277 units of Aw? (dB km-1)*
No. $A_w$ is in dB. Added.

*12) L290 units of Awconst?*
$A_{wconst}$ is in dB. Added.

*13) Figure 7, the word "dry spells" seems cut at the bottom. Abbreviation for the month of November should be "Nov." if following the correct abbreviation, otherwise writing it in full could be actually better.*
OK.

*14) L497 chronological order for citations should be preferred*
We use alphabetical order.

*15) L571 "attention" should be "attenuation"*
Corrected.

*16) L 609 "by lower frequencies" "by" should be removed.*
"by" replaced by "at".

**2. Response to anonymous referee 1**

*1) Fig.1: in the caption "scattering efficiency" is mentioned. Clearly this is not the same as extinction efficieny. Please clarify.*

Thank you for spotting this inconsistency. The figure as well as equation 6 describe extinction efficiency ($Q_{ext}$).

We have corrected the Figure labels, description of the equation, and also one occurrence in the introduction section.

*2) Fig.2: I suspect there is something wrong here. I do not see why the water vapor attenuation should have a drop at 60 GHz. Is this affecting results later on???*

We have checked our results and there is nothing wrong with it. Results are fully consistent with Liebe et al. (1993) and could be cross-checked by independent computer codes available on GitHub, e.g. https://github.com/cchwala/pyMPM. The attenuation due to water vapor is in here (as well as later on) defined as the difference between wet-air and dry-air attenuation under the same moist air pressure and temperature. Thus, also effect of water vapor on attenuation due to oxygen is considered as described in ITU-R, (2019), and formerly by Liebe et al. (1993): First, moist air pressure is the sum of dry-air pressure and partial water pressure, thus dry-air pressure decreases during humid conditions (under the assumption of the same moist air pressure). This leads to a decrease in attenuation by molecules of oxygen. Second, partial water vapor pressure influences the width of the oxygen spectral lines (pressure broadening) as it affects the rate of collisions between the molecules (eq. 5 and 6a in ITU-R (2019)). These two effects lead to a decrease in attenuation due to oxygen when water vapor content increases. This decrease can reach about 4 % for conditions with moist air pressure 1015 hPa, temperature 30° C, and 100 % air humidity. Such decrease has relatively negligible effect for frequencies used in this study (73-74 and 83-84 GHz). Nevertheless, it is not negligible around 60 GHz (as can be seen on Figure 2), where the attenuation due to oxygen dominates over attenuation directly caused by water vapor molecules.

We have clarified the definition of attenuation due to water vapor in the paragraph describing the Figure. We have also added an explanation how water vapor influences attenuation due to oxygen (L97 – 101 in the revised manuscript). Finally, the reference to ITU-R recommendations (ITU-R, 2016) has been updated to ITU-R (2019):

*3) I find the narrative from 3.3 onwards (including Sect4) very difficult to follow. I would recommend to reshu so that for instance when you talk about Sensitivity of the k-R model to drop size distribution you cover the whole thing (including line 400-425). Same for the other bits (e.g. Quality check, Dry-wet weather classification, Baseline identification, Wet antenna attenuation). At the moment the reader need to jump back and forward because the logical thread is erratic. Also some topic (e.g. gas attenuation should come before rainfall retrieval because of course the effect of gas must be subtracted first!).*

We thank for the suggestions on modifying the structure of the manuscript. First part of the comment suggests presenting jointly methods and results belonging to the one subtopic. However, we prefer to avoid the inclusion of results in the Materials and Methods section. Moreover, the research performed is multifocal, the use of subheadings within the Materials and Methods section, that are mirror-imaged in the Results section, is intended to make it easier for readers of the manuscript to associate which Materials and Methods are used to obtain which results. Furthermore, all the subtopics *(e.g. Quality check, Baseline*

*identification, Wet antenna attenuation)*' are part of the same processing chain and influence each other and thus it is in our view reasonable to present them together.

Regarding the second suggestion (re-order rainfall retrieval and gas attenuation), we agree that effect of gaseous attenuation influences baseline identification and can be presented earlier.

In the revised manuscript, we, firstly, present subsections related to gaseous attenuation before subsections related to effect of drop size distribution on attenuation-rainfall model and subsection related to rainfall estimation procedure. The order of subsections has been, therefore, changed in Sections 2, 3, 4, and 5 to maintain mirror-imaging of the topics within the sections. Secondly, separate subsection 3.6 Performance evaluation was removed and its content is presented within the other subsections of Method section. E.g. Performance evaluation related to gaseous attenuation is presented in the subsection about gaseous attenuation, etc. Thirdly, subsections referring to dry-wet weather classification and hardware related artifacts were moved to the appendices. Both of the topics are not critical to the conclusions but are in our view useful for experts utilizing CML data for rainfall retrieval. Finally, to further improve flow of the presentation, IDs of CMLs were simplified to numbers from 1 to 6 with suffices "a" and "b" (e.g. 1a and 1b) when referring to 73-74 GHz resp. 83-84 GHz sub-links.

*4) Eq.10 and Fig.4b. Where is this coming from? I have never seen such a relationship between an intensive quantity like RR and D_M!!!!!*

The parameters of theoretical functions describing drop size distribution (DSD) are scaled to rainfall intensity, as can be seen in Figure 4a. In this particular case, gamma distribution as proposed by Ulbrich (1983) is used. Thus, mass-weighted diameters (ratio between $4^{th}$ and $3^{rd}$ DSD moments) need to be also scaled to rainfall intensity, as shown in Figure 4b. The scaling procedure is explained in the paragraph above the eq. 10 (lines 224-226 of the original manuscript) and adequately referenced to the benchmark paper of Ulbrich (1983).

*5) Table 3: you are introducing parameters (epsilon, delta,) that are not defined anywhere.*

The parameters are taken from Table 2 in Ulbrich (1983). The paper was cited in the Table caption of the original manuscript. We have added details on each parameter to the Table:

*"$N_0$ ($m^{-3}cm^{-1-\mu}$) and $\mu$ (-) are parameters of semi-empirical gamma distribution function, $\varepsilon$ ($h^{-\delta}$) and $\delta$ (-) are scaling parameters of this function."*

*6) I do not understand the rationale of doing the investigation with the "theoretical DSD" (not clear where they come from). On the other hand I see the point of using disdrometer data but I would recommend to use extensive datasets like available at https://ghrc.nsstc.nasa.gov/home/field-campaigns (and plot density functions instead of plotting scatterplots as in Fig.10). This should also allow to assess uncertainty errors due to DSD variability like done in Tab.4 in a more robust way.*

Theoretical DSD enables us to show, that results based on observed DSD are consistent with results using widely acknowledged theoretical DSD functions, which in our opinion supports the conclusions and shows that the results are not only site-specific. When describing theoretical DSD, we use gamma distribution function scaled to rainfall intensity as suggested by Ulbrich (1983).

In our opinion, one year of DSD data from the well-controlled experiment is sufficient to demonstrate the effect of DSD on attenuation-rainfall power-law relation. The dataset was previously used in several papers in high-quality journals (e.g., Schleiss et al., 2013; Wang et al., 2012) and we had the dataset handy. Moreover, the dataset is from temperate climate and thus applicable also for our case study.

*7) The authors mention stratiform vs convective precipitation coefficients. How do they practically envisage to separate stratiform vs convective precipitation?*

The separation to convective and stratiform rainfalls is performed only in the investigation with theoretical and observed DSD (sections 3.3 and 4.4 of the original manuscript) to demonstrate that attenuation-rainfall relation is sensitive to rainfall type. The classification procedure is described in the section 3.3 of the original manuscript.

The separation of convective and stratiform rainfalls in practical applications is discussed in Discussion section of the original manuscript (L545-548). This section refers to the work of Leth et al. (2019). Unfortunately, detailed elaboration on possible methods for separation of convective and stratiform rainfalls is out of the scope of this manuscript.

*8) Assuming that "rainfall has a uniform distribution over the study area, and that water formation on the surface of antenna radomes is the same for both the short CMLs and the long one" is quite an assumption! This approach is very provisional.*

We agree that assumptions seem to be strong. Nevertheless, we believe that its appropriateness is, in our case, justified and was carefully discussed in Discussion section of the original manuscript (L511-518). Line of reasoning is briefly repeated below.

The rainfall spatial variability (in the studied period) is low: The correlation coefficient between 15-min rainfall observations of nearby rain gauges (rg_1 and rg_2) is 0.94–0.96. The correlation between these rain gauges and the more distant rain gauge rg_3 is still over 0.88.

Wet antenna attenuation (WAA): All the CML units are of the same type and have similar ages (deployed during 2016 and 2017). Furthermore, as discussed on L516-518 of the original manuscript, the procedure is in our case relatively insensitive to differences in antenna properties, because WAA is quantified by comparing short (0.4 – 1.4 km) CMLs by which WAA dominates over path attenuation, to the 4.86 km long CML, which is relatively insensitive to WAA even during light rainfalls. Satisfactory performance of the approach is also demonstrated in the Results section in Figure 7.

To further strengthen the arguments about the legitimacy of those assumptions, we have added an information on the age of the units into section 3.1 Experimental sites and instrumentation. Furthermore, Figure demonstrating the concept of WAA estimation on time series (Fig. 9a in the revised manuscript) was extended by a scatter plot (Fig. 9b) with regression lines relating observed attenuation during light and moderate rainfall to the length of sub-links.

*9) "E-band CMLs are by about one order of magnitude more attenuated by raindrops along their path than older 15–40 GHz devices": this is a very vague (imprecise) statement also given the fact that attenuation at 40 GHz is already 6 times attenuation at 15 GHz!!!! Same for the sentence "Gaseous attenuation at E-band CMLs is detectable, however, it is two orders of magnitude smaller than attenuation due to rainfall" (again quite vague and approximate!)*

Agreed. We will be more precise in our statements. We have modified the conclusions as follows:

- "E-band CMLs are markedly more attenuated by raindrops along their path than older 15–40 GHz devices, during lighter rainfalls by about 20 times more than 15 GHz and 2 - 3 times more than 40 GHz devices."

- "Gaseous attenuation at E-band CMLs is detectable, however, it is substantially smaller than attenuation due to rainfall. Fluctuations in specific attenuation caused by water vapor typically not exceed 1 dB km$^{-1}$ in the region of temperate climate. This magnitude is reached by rainfall with intensity around 1 mm h$^{-1}$."

Our intention is to avoid inclusion of our original findings in section 1 and section 2. Section 1 provide general introduction with state-of-the art in microwave link rainfall estimation based upon which the goals of this manuscript are defined. Section 2 provides theoretical background enabling reader to follow our original methodology and results.

We would like to keep our original methodology and findings clearly separated and thus we want to avoid summary of our original findings already in the introductory sections. Similarly, we would like to keep our original methodology separated from theoretical background provided by previous works (section 2). Thus, although concept of baseline separation is introduced already in the Section 2 we prefer to explain how we approach this challenge in Section 3 - Material and Methods. Following the same intention, we prefer not to explain assumptions behind quantifying wet antenna attenuation in this work already at L94 – 101 of the original manuscript, but again in the Section 3.

To clarify this intention, we have modified the paragraph describing structure of the manuscript, specifically description of section 2 and 3: "Section 2 of the manuscript summarizes based upon previous

works the principles behind retrieving atmospheric variables from CML observations, Section 3 describes the methodology and datasets used in this manuscript for the assessment of E-band CMLs, …".

*3)        An additional figure showing WAA against link length could be used to illustrate the linear regressions proposed in Eq. 11 and the constant behaviour in dew cases. A sample of how the figure could look is attached.:*

Agreed. We have shown such figure (Fig. 9b in the revised manuscript). Details are provided in the specific comments no. 11 and 17.

**Specific comments (in order of appearance):**

1) *L68 - Free space loss (Lbf) is said to be uniquely defined by distance and wavelength. Reporting the formula could be appropriate and helpful for further understanding of the discussion, as the frequency is a key variable for this study (E band).*

   Yes. We now report the formula of free space loss in the revised manuscript on line L68-70:

   "Free space loss $L_{bf}$ is uniquely defined by the distance $d$ (m) between the transmitter and receiver, and by wavelength $\lambda$ (m):

   $$L_{bf} = 20\left(\frac{4\pi d}{\lambda}\right)"$$

2) *L74-78 - The phrase "Attenuation during dry weather is assumed to be a baseline" is apparently in direct contrast with the following "Fluctuations in the baseline during dry weather can be attributed..." if the reader does not know already the different magnitudes involved. Early introduction of orders of magnitude and average behaviours is therefore encouraged.*
   Agreed. We report typical magnitudes of rainfall and gaseous attenuation in the section 2.1, after description of different components of total observed loss (L72-75 in the revised manuscript).

3) *L101 - "More extensive investigations..." I think this sentence will state the motivation of the author's work, but it could be also interpreted as what still remains unknown after the work's results instead. Please clarify to avoid this ambivalence.*

   The sentence indeed stated our motivation. The motivation was, however, presented already at the end of Section 1. We have, therefore, deleted this sentence to avoid misunderstanding. Furthermore, we express at the end of the Introduction section, where structure of the manuscript is described, that section 2 provides review of previous work (see response 2 in the general comments).

4) *L131 - Fig. 4 is useful to the contextualisation of this sentence and should be referenced. "Contribute relatively less" is not gaugeable, some more detail may be added.*

   Agreed. We have referenced subsection 3.4, which contains Figure 4 (we would like to avoid referencing Figure 4 before Figure 3 is referenced). Nonetheless, we kindly disagree with the second suggestion. We would like to avoid detailed quantitative description in here. Reader can easily read quantitative information from the figure referenced in this sentence (Fig. 2 in the revised manuscript).

5) *L145 and following - The study on the components of N is not justified by following discussion or results and could be omitted as it lacks quantitative information. I think that the qualitative concept of the dependency of k to the various components is already* well *stressed.*

Agreed. As suggested, study on the components of N (lines 145-150 of the original manuscript) has been omitted in the revised version of the manuscript. Interested reader can find these details in the cited literature.

6) *L194 - "The periods for evaluating rainfall retrieval and for evaluating the effect of humidity and temperature fluctuations on gaseous attenuation are, therefore, different." The phrase itself is a quite obvious consequence of the previous sentence, while its implications are not. It should either be omitted or some expected implications should also be discussed (or at least some reference to the respective discussion should be made) in terms e.g. of which investigations are precluded by using different time windows.*

Agreed. The phrase is obvious and it has been deleted.

7) *L200 and other appearances of "aggregate" - it should be pointed out how the aggregation to different time scales is performed (mean, median, sum, max, other...)*

We aggregate to different time scales using mean. We have added this information into revised manuscript.

8) *L246 and 247 - The sentence is not clear and should be rephrased and expanded. "dependent" should perhaps be substituted with "depending", commas before and after "therefore" are not necessary and slow the reading. The threshold for Dm is not indicated.*

Thank you, for spotting this typo. This typo apparently led to misunderstanding. $D_m$ as estimated by Eq. (11, in the revised manuscript) is actually the threshold which is used for classifying rainfalls. This threshold ($D_m$) is dependent on rainfall intensity.
We have changed the sentence to: "The approximation (Eq. 11) is used to calculate threshold for classifying disdrometer records as convective or stratiform. The threshold is dependent on rainfall intensity. Parameters *c* and *d* are estimated by fitting Eq. (11) to $D_m$ as derived from real disdrometer data using Eq. (10)".

9) *L259 - Visual inspection does not seem like a robust approach to filter the outliers. Some technique should be at least suggested to cope with this kind of artefacts, as the visual approach is clearly not feasible at larger and near real-time scales.*

Visual check is indeed not a robust approach which could be used in future applications. The automation of quality check is, however, out the scope of this manuscript. Visual identification of artifacts is, in our view, first step towards future automation of this process. Moreover, the correction for artifacts is performed only in a single case. This correction is transparently reported (L260 of the original manuscript), to ensure reproducibility of the results.

10) *L269 - One-week sized moving window "is sufficiently short": are baseline drifts proven to happen only at longer time scales? Is the same for gaseous attenuation? Could it be that some higher frequency signal is masked by this approach resulting in the weakening of the water vapour detection capabilities?*

No, as reported on L260-261, also sudden change in baseline occur in the case of CML 3004_3005 (ID 2 in the revised manuscript) and this change was manually corrected. The baseline identification using one-week sized moving window is used only for rainfall retrieval. As reported on L305-306 of the original manuscript, constant baseline is used when analyzing effect of gaseous attenuation and potential for water vapor retrieval. Hardware related artifacts causing slow baseline drift have probably potential to destroy gaseous attenuation signal as discussed in the Discussion section (L581-584 in the original manuscript).

11) *L283 - A reference to Fig. 8 or to the suggested new figure could be added here.*

We have modified the figure in the revised manuscript according to reviewer's suggestion, however, we have at the end decided not to reference the figure at this place to keep the methods strictly separated from the results.

12) *L296 and 340 - Since Prague is located at an altitude around 200m (990 hPa), to as-sume the atmospheric pressure of 1013.25 hPa seems either systematically wrong or reported with too high precision (if differences between 990 and 1013 are negligible for the author purposes, then decimals of hPa are even more so). It is therefore suggested to utilize 990 hPa as reference pressure or at least replace the number with a more generic "at sea-level pressure".*

Agreed. We have replaced the number with more generic "sea-level pressure".

13) *L323 - Short CMLs are highlighted in some following sections as valuable tools for intense rainfall detection. Here instead the sentence "The performance ..." says that they will be presented only as examples of bad performance. Please clarify.*

Potential of short CMLs is highlighted in the Discussion section (L556 -562 of the original manuscript) in the context of observing heavy rainfalls associated with high spatial variability by which an assumption about uniform rainfall distribution along a CML path is more likely valid for short CMLs than for long ones. Nevertheless, only light and moderate rainfalls occurred during observation period. Short CMLs are, during these rainfalls relatively more affected by wet antenna attenuation than longer CMLs, as demonstrated e.g. on figure 7 in the original manuscript. The sentence, in the original manuscript actually states that: 'The performance of the short CMLs is shown to demonstrate limitations related to the improper baseline and WAA identification which are more pronounced by shorter CMLs.' This is in our view, not in contradiction with Discussion section, where shorter CMLs are suggested as valuable tool for detecting heavy rainfalls.

To avoid misunderstanding, we have modified the sentence to: 'The performance of the short CMLs is shown to demonstrate limitations related to the improper baseline and WAA identification which are, especially during light rainfalls, more pronounced by shorter CMLs.'

14) *L347 to 358 - Is there any indication of what could cause the "degraded resolution" on the hardware side? If yes, it would be an interesting topic to read here.*

In our opinion, the degraded resolution might be related to automatic power control. Nevertheless, this was not tested. We thus prefer to not speculate in this direction.

15) *L365 and Fig. 6 - It should be reminded to the reader that sub-links belonging to one CML are presented in pairs in consecutive order. It should be consequently pointed out that intra-CML correlation creates 2x2 darker squares along the diagonal in the correlation matrix plot.*

We have decided to change IDs of CMLs in the whole manuscript to better indicate sub-links belonging to the same CML and in general improve clarity of the whole manuscript: each CML have unique ID (numbers from 1 to 6). IDs of sub-links operating at 73 – 74 GHz and 83 – 84 GHz frequency then consist of CML ID and suffix "a" resp. "b".

Regarding intra-CML correlation, we believe, that the first sentence referring to the Figure 6 (Figure A1 in the revised manuscript) now provides sufficient guidance in this respect: "Dry-wet weather classifiers of single sub-links belonging to one CML are strongly correlated (Fig. A1)."

16) *L369 - It is stated that the delay of the rain gauges in detecting rainfall with respect to CMLs "can be attributed to the delay of rain gauge rain detection due to the filling of the bucket." Please discuss whether delays and volume losses are compatible to the bucket size.*

Thank you for this comment. We have removed this statement as it was based on analysis not shown in the manuscript and might have been considered speculative. The section will thus contain only results shown on figure A1 of the revised manuscript.

17) *L377 to 380 - Same as L283, the dependency (and independence) of WAA to path length should be presented for rain (and no-rain) occurrences with a specific scatterplot and a linear fit (suggested figure attached).*

Agreed. Figure 7 (9 in the revised manuscript) has been extended by a scatter plot showing relation between path length and total attenuation together with linear fits indicating effect of WAA.

18) *Fig.7 - When comparing signals from CMLs of different path length, specific attenuation (dB/km) should be preferred to pure attenuation (dB). If the aim is to show the different regimes (dependency and independence to path length), then two plots should be shown (dB and dB/km time series), in order to appreciate inter-CML concordance on specific attenuation during rainfall and on pure attenuation during dry periods.*

The signal shown in the figure 7 (Figure 9 in the revised manuscript) is predominantly caused by wet antenna attenuation, which is independent of path length. We therefore prefer to show exclusively total attenuation in this figure.

19) *L405 - "However, it is closer ..." the reported considerations is interesting for an operational use and therefore valuable, but it is poorly proven (only visually). Without a gauge of the goodness of the*

*approximation (or some reference to following consistent results), the ITU fit may as well not be good for either case (convective and stratiform).*

The attenuation-rainfall relation is for theoretical drop size distribution almost perfectly approximated by power-law fits (as reported on L400-401 of the original manuscript). Thus, distances between presented power-law curves (absolute errors) provide meaningful gauge of goodness. The term "it is closer" on L405 in the original manuscript describe distances between the curves. Thus, it is, in our opinion, appropriate. Moreover, reader can easily get information on approximate distances for any rainfall intensity between 0-50 mm/h from the figure 9 in the original manuscript. In addition, parameters of power-law fits as well as parameters obtained from ITU (ITU-R, 2005) are provided as a part of figure 9. Interested reader can thus easily express exact value of absolute errors for any rainfall intensity, resp. specific attenuation.

20) *L433 - To my understanding, it is the first time here that some speciific deficits in baseline and WAA identification for sub-link 1147 are asserted. It seems quite in contradiction with other parts of the text were the long CML has the best results.*

The sub-link 1147 (1b in the revised manuscript) clearly outperform shorter CMLs in terms of correlation (r = 0.96 and 0.97 compared to 0.53 – 0.86 resp. 0.61 – 0.87) and in terms of RMSE (0.39 and 0.24 mm h$^{-1}$ compared to 0.64 – 2.18 resp. 0.69 – 1.44 mm h$^{-1}$), which can be seen in Table 5. Its markedly better performance is also clearly visible from the scatter plots in figure 11. The long CML cannot, however, accurately capture very light rainfalls under 1 mm h$^{-1}$, which represent about 25 % of the total rainfall depth in our case. We have added an information about underestimation of very light rainfalls to the revised manuscript at the place describing performance of the long CML (L434 - L435).

21) *L448, 449 and Fig. 12 - The anti-correlation of the attenuation with temperature is evident from figure 12b and should be highlighted here, as temperature seems to be the dominant component of the signal. Moreover, this appears in direct contradiction with what stated in the first paragraph of Section 4.6, so that may be reformulated differently.*

The negative correlation between attenuation and temperature appears in the figure, because water vapor density is strongly correlated with temperature. As gaseous attenuation is highly correlated to water vapor density, there is also strong (negative) correlation link between gaseous attenuation and temperature. It is, however, not caused by direct dependence, which is almost negligible: See ITU-R, (2019) and Figure 2 of the original manuscript.

22) *L515 - "The similarity in antenna characteristics was not inspected directly." Are the antenna factory features known to the authors? Is this sentence referring to technical specifications of the antennas or to the actual status of the radomes?*

The sentence refers to hydrophobic properties of antenna radomes as well as actual status of the radomes. It has been specified in the revised manuscript.

23) *Supplementary material - The ATPC (5th paragraph) is said to be "switched off" but, to mine understanding of Fig. S1, the concept of "saturated" may be more adherent to the case. It seems to me*

*that ATPC can deal only with maximum 7 dB gains on tx, but it keeps working even there, in the sense that the gain remains 7 dB, while "ATPC switched off" is more likely a zero-gain scenario.*

Yes, the ATPC keeps working in the sense it maintains *tx* power on the maximal (allowed) level. In the revised version of the supplementary material, we use the term 'saturated' instead 'switched off'.

*Technical observations:*

*Figure 2 - It is not clear what the coloured bands represent (standard deviation or total spread) and neither is the direction from low to high pressure.*

The color bands represent total spread. It is now clarified in the figure caption.

*L202 and Table 2 - "Height" is used, but maybe "depth" is a more common choice to indicate precipitation amount.*

Agreed. We now use the term 'depth'.

*Fig. 6 - Raingauge labels differ between image and caption ("wet_" prefix)*

Thank you for spotting this inconsistency. We have corrected it.

*Fig. 8 - Since the two plot rows represent different frequency ranges, some labels indicating the two ranges are fostered to be shown to the left of the plot. Otherwise this information should at least appear in the caption with "upper row" and "lower row" indications.*

We have added to the left two labels indicating frequency ranges.

*L413 - I suggest the replacement of "heteroscedastic" with a more generic formulation, e.g. "the spread clearly grows with R and k". Although the adjective is certainly correct for a distribution like the one shown in Fig. 10, its use seems not proper for this context: given its precise statistical meaning and implications, I think it is preferable to run some specific tests of heteroscedasticity before asserting this property.*

Done.

*Fig. 12 - The colours for theoretical and observed attenuations are poorly chosen as they appear very similar (especially light green against light blue), both on paper and on screen.*

OK, we have adjusted the colors to differentiate better the time series.

[Figure]

*Fig. 1. Scatterplot of attenuation against pathlength with separated linear fits for rain and no-rain intervals (reviewer's suggestion).*

Note, that the changes in the manuscript structure indicated above are not marked in the marked-up version of the revised manuscript presented on following pages.

[revised manuscript text omitted]

---

## Author Response (AR2)

**Response to reviewer 3**

**Recommendation to the editor**
Thanks for the opportunity to review this interesting paper. I find the topic of the paper of high interest for AMT since it presents an innovative atmospheric measurement opportunity that involves new technology and that requires ad-hoc processing.
However, there are, in my opinion, major deficiencies in the presentation quality that need to be addressed in order to make the suggested measurement technique applicable by other researchers.
Moreover, there are some weaknesses in the scientific reasoning presented that I recommend addressing before publishing. Those weaknesses might not affect the results, but it would be unfortunate to leave incorrect statements on published science that can be misused in the future.

Given the fact that the paper is already under review and the main data is already presented I recommend a major revision and I am definitely willing to contribute again in the review process. However, given the extent of weaknesses of the paper that I will try to list I also encourage the authors to take the time to reformulate the study from scratch and submit a new paper. In either case, I hope my comments will help the authors improving their study.

*Authors:*
*We would like to thank the reviewer for a constructive criticism and suggestions for improving our manuscript. We have tried to respond to all the comments and revised the manuscript significantly. Specifically, we have for the second time substantially revised the manuscript structure, markedly revised the sections related to drop size distribution, and made an attempt to quantify uncertainties influencing our results, which had been previously described rather qualitatively. We have also corrected mistakes, inconsistencies and ambiguities spotted by the reviewer.*

GENERAL COMMENTS:

C1) The paper lacks a clear and concise description of the procedure adopted to measure the atmospheric quantities. The concept of baseline and the separation of dry and wet weather is briefly presented at lines 78-82 and does not catch the attention it needs. When I first read the paper I thought that the measuring principle was Eq. 1, but in reality, it is a different formula that involves the concept of baseline (which only partially resembles the components of Eq. 1). The authors have to make a clear definition of dry and wet weather condition (at the moment the reader have to extrapolate from the contest). For dry and wet weather it would be beneficial to have a formula like Eq. 1 that explains how weather conditions influence your measurements (signal loss).

*A1: Issue is related to the procedure adopted to measure the atmospheric quantities. Originally, it was described in the method section (lines 230-234 and 297-299). We, however, agree that baseline separation is an important concept which deserves attention. We have, therefore, introduced the concept of the baseline already in section 2.1 (lines 77 - 88) including the formula (originally Eq. 13) and added there also the definition of dry and wet weather:*

*'Precise separation and quantification of different components of total loss requires detailed description of atmospheric conditions along a CML path as well as conditions influencing hardware of transmitting and receiving stations. The specific path-attenuation due to raindrops or due to water vapor k (dB km⁻¹) is thus usually separated from other sources of attenuation using data-driven approach:*

$$k = max \ (\frac{L_t - B - Aw}{l}, 0)$$ *(3)*

*where l (km) is a CML path length, B (dB) is background attenuation, so called baseline, and Aw (dB) wet antenna attenuation (WAA) caused by antenna radome wetting occurring during rainfall or dew events. Baseline is most commonly estimated from attenuation levels during periods without rain and without dew occurrence on antennas (Overeem et al., 2011; Schleiss and Berne, 2010). ... We further refer to the periods with and without rain and dew occurrence as wet resp. dry weather.'*

*The influence of wet weather conditions as a complex issue is described in the following three subsections.*
*We would like to also note, that section 2 is not intended to describe our original methodology but to provide theoretical background reviewing studies related to CML retrieval of atmospheric variables. The title of section 2 was, therefore, changed to "Retrieving atmospheric variables from CMLs – theoretical background" to stress intended content.*

C2) The paper title and abstract suggest that the main results of the study are to use CML to measure rain-rate and humidity. However, in the results section, it is possible to find only an attempt to measure the rainfall rate. Regarding humidity, the paper presents an attempt to estimate microwave attenuation from RH measurements and not the opposite. The results section is filled with arguments about the data processing which do not belong to the results section (wet antenna attenuation, k-R modeling, and gaseous attenuation). It is preferable to put all the data processing in one section and leave the Results section for the atmospheric measurement results and the accompanied estimated uncertainties (that will be of course a consequence of the processing).

*A2: We agree that the manuscript (MS) predominantly focuses on the evaluation of rainfall retrieval form E-band CMLs. However, in our opinion, it also investigates the potential of E-band CMLs to observe water vapor. This relies strongly on the ability to separate attenuation due to water vapor from other losses and thus our evaluation focuses on this aspect. The results show that such separation is highly challenging, and, in our view, we state this clearly in the last sentence of the abstract. This is also reason, why we did not proceed further in the quantitative assessment. We would like to also note that we use in the abstract term 'water vapor detection' which, in our view, reflects that water vapor retrieval is not directly quantitatively evaluated. With respect to this issue, we have, however, revised introduction of the method section and section 3.5 describing methodology for evaluating to stress the focus the analysis.*

*Regarding second concern, variable DSD and wet antenna have been previously identified as one of the most limiting factors influencing CML rainfall retrieval. Quantitative evaluation of these effects was thus in many previous papers considered as a result (e.g. Berne and Uijlenhoet, 2007; Schleiss et al., 2013). We, nevertheless, agree that putting all the data and processing into the Methodology sections and leaving the Result section for the atmospheric*

*measurement results and uncertainties is reasonable. We have therefore revised the MS accordingly.*

C3) The information about methods, data, and results are scattered around and it is very hard to follow the logic of the paper. The authors made an interesting assessment of the ITU k-R relation using PARSIVEL synthetic measurements. I believe that this is an interesting analysis, but it is logically separated from the atmospheric measurement attempts. It might be useful to separate it from the rest making it an entirely separate section (between data and results) or even better would be to include it in what is now section 3.4 making it a self-contained development of the k-R based retrieval technique. Doing that the paper will emphasize the two main atmospheric measurements, namely the dry-weather estimation of water vapor and the wet-weather estimation of rainfall (which includes wet antenna attenuation estimation as a processing step). It would also help to merge section 4 and 5, putting the discussion of the results close to the presentation of them.

*A3: We agree, that evaluation of k-R relation using PARSIVEL data can be presented as a self-contained development of the k-R based retrieval technique in one subsection section of the Method section and revised the MS accordingly (section 3.3. in the revised MS). We would keep discussion section separated from the results.*

C4) The difficulties in understanding the procedure adopted come also from the fact that not all the variables introduced in the paper are properly defined. Some variable names are reused (c and d are power-law coefficients in section 3 and 4, but where first introduced in section 2 as the speed of light and distance between antennas). Frequency f changes measuring units from Hz at line 93 to GHz at line 96. The definition of dry and wet weather is not explicit, it comes just at the end of the paper from practical considerations. The capped Dm (Eq. 11) is not defined (I think it is the assumed threshold between convective and stratiform events). The separation between stratiform and convective is not described anywhere (it was already question 3 from referee 1); after reading the paper several times I am supposing the threshold Dm is given by an imaginary line in between the two of figure 4b, but this is not written in the paper. Finally, the parameters of the DSD are reported in Tab. 3, but they are not explained (already question 5 from reviewer 1).

*A4: The variable names in the revised MS have been unified and units properly defined. The definition of dry-wet weather is introduced earlier (see answer A1)*

*Capped Dm is estimated based on disdrometer data and its definition and estimation procedure was described at lines 265-270 of the original MS. Suggested revisions further clarifying the procedure are discussed in the answer A8.*

*The parameters introduced in table 3 were explained at bottom line of the table only. In the revised version of MS, reflecting criticism in C8, we have decided to remove analysis with theoretical DSD functions.*

C5) Some comments appear to come out more from wishful thinking than from a proper quantitative evaluation of the results. As an example in lines 442-446 the uncertainties related to WAA and DSD assumptions are discussed only in a qualitative way. The authors missed the opportunity to quantify the uncertainty related to WAA to RR estimation as a function of link length. Alternatively, by analyzing the k-R scatterplot of the Prague data one could potentially make some quantitative assessment of the RR retrieval uncertainties due to DSD

assumptions (the underestimation of RR below 2 mm/h is a very interesting aspect related to this).

*A5: We made an attempt to estimate quantitatively uncertainties of CML QPEs and uncertainties related to WAA and DSD. The methodology of uncertainty estimation is now described in the method section and estimated uncertainties are presented together with the results (section 4.2). The updated evaluation indicates that underestimation of RR below 2 mm h$^{-1}$ might be indeed related to DSD as it corresponds quite well to expected underestimation of the ITU-based k-R model during stratiform rainfalls (Figure 10 of the revised MS and also Figure R3 in answer A25).*

SOME CONCERNS ON THE FIRST REVIEW:

C6) During the review process, the CML sub-link naming scheme changed. This modification makes sense since it simplifies the naming scheme, but I do not see in the author's response a mention to this change. Also, the naming change seems not consistent: link 3008-3009 became link 6 in the map of figure 3, the same link in table 1 became link 3 (but I see that there is a reordering problem here), but again in figure 10 the cluster of data point that was previously from 3008-3009 are now belonging to the subplot of link 3. I suggest mentioning all the changes made to the manuscript in the ``answers to the reviewers" documents, also the ones that have not been suggested by the reviewers.

*A6: This change was reported in the response to the referee 1, specifically response to the comment no. 3. We have decided to reorder the IDs to reflect link length. Reordering, in our view, improves clarity of the section quantifying WAA, where effect of CML path length on total attenuation is demonstrated. However, we made mistake in Figure 10 (Figure 6 in the revised MS). The mistake is corrected and all the figures are now consistent with the new naming.*

C7) I wasn't able to find the details of the T-matrix simulations in section 3.5 as the authors answered comment 2 by Dr. Guyot. Actually, I wasn't able to find those details anywhere in the manuscript. I suggest to include the T-matrix parameters information and to move it to section 2 (not 3.5 as the authors mentioned), this is because the T-matrix parameters are essential to reproduce the results of figure 2.

*A7: Thank you for spotting this inconsistency. We have added the details on T-matrix simulations into the revised MS to section 3.3 at lines 245-248, as the parameters of the T-matrix simulations are integral part self-contained development of the k-R based retrieval technique. Moreover, the paragraph explaining extinction-efficiency as well as figure 2 were removed (see A18).*

A8) The answers of the authors to reviewer 1 (questions 5 and 6) are not addressing the reviewer's concerns. Probably the authors misunderstood the questions since they briefly refer to Ulbrich (1983) to cover the entire discussion, but the points remain unanswered. Moreover, the phrasing used in both the manuscript and the answer is imprecise and leads to a misunderstanding of Ulbrich (1983) findings.
In Ulbrich (1983) it is assumed that any DSD is well represented by a three-parameter modified-gamma distribution. This assumption leads to the conclusion that every couple of moments of the DSD can be related through a power-law. Because of that, if one can

characterize a couple of moments through a power-law it follows that the DSD becomes a 1-parameter only function (the free parameter is lambda) and any other couple of moments will be characterized by a corresponding power-law. For this reason, the reflectivity-rain rate fits of Fujiwara (1965) can be converted into fixed parameters N0, mu for the DSD and power-law coefficients epsilon-delta for the Dm-RR relation.

There are many problems with this approach:

- All the assumptions of Ulbrich (1983) have to be valid. The authors did not test, for example, how good a modified gamma with fixed N0 and mu parameter fit the DSDs measured by PARSIVEL

- The error in the Z-R fit are not evaluated and transferred to errors in the N0, mu, or Dm as computed by the implied Dm-RR relation

- The mathematical foundations of Ulbrich (1983) have been demonstrated to be flawed in logic (Illingworth and Blackman 2002), leading to artificial correlations among parameters. To the best of my understanding, the theoretical DSD is used only to make a rough evaluation of the stratiform or convective nature of the precipitation in the PARSIVEL dataset. I do not think it will affect the results, but the explanation of how to use the theoretical DSD has to be corrected anyway.

What it comes out from these considerations and might be harder to sustain is the following: ``The k-R function (stratiform DSD) applied to the Prague dataset has been estimated from a fit to synthetic data obtained from the Duebendorf dataset (13 months) whose stratiform-convective classification is based on the distance of the data from the Dm-R curves derived with a mathematically faulty logic (Ulbrich 1983) using reflectivity-rain rate fits obtained by observing 31 storms in Florida (Fujiwara 1965)."

This argumentation looks weak to me and I wonder if the authors excluded any other possible option they had to discriminate between stratiform and convective cases in the Duebendorf dataset.

*A8: This is really interesting point. We were not aware of the shortcomings of Ulbrich (1983) parametrization and indeed did not interpret concerns of the reviewer 1 in this respect. We, therefore, suggest removing analysis with theoretical pdfs of drop size spectra (it was already suggested by reviewer 1) and keep only analysis with DSD from the PARSIVEL dataset.*

*Regarding rainfall type classification, we would like to note, that we aim at distinguishing between rainfalls based on their drop size spectra, rather than classifying nature of precipitation. In the revised MS, we explicitly state this in the introductory of section 3.3 (lines 242-244): 'The investigation is performed on PARSIVEL observations of DSD from Duebendorf dataset. The classification of rainfalls based on their mass-weighted diameter is introduced to enable optimization of k-R model separately for rainfalls with different drop sizes.'*

*We have also changed explanation of classification procedure on lines 261-263 and hopefully, more clearly state in the revised MS that the classification evaluates nature of the precipitation only roughly: '$D_m$-based classification separate rainfalls by size of their raindrops to two classes and thus also roughly evaluates convective and stratiform nature of the precipitation in the PARSIVEL dataset (Jaffrain and Berne, 2012). We further refer to those two classes as 'stratiform' and 'convective'.'*

*The mass-weighted diameter (or median volume diameter or other DSD moment ratio) is in our view appropriate descriptor enabling such classification. Relating the capped $D_m$*

threshold to rainfall intensity is consistent with our data and also with some other studies, *e.g.* Meshesha et al. (2014).

*The power-law relation between DSD descriptor and rainfall intensity was optimized using the whole Duebendorf (PARSIVEL) dataset (described originally at L270-271). We have revised this description (lines 259-265 in the revised MS) as follows: 'Parameters $\gamma$ and $\delta$ are estimated by fitting Eq. (11) to $D_m$ as derived from PARSIVEL records using Eq. (10). Specifically, sum of squared residuals between $D_m$ obtained from Eq. (10) and Eq. (11) is minimized. This results in parameters $\gamma = 1.29$ mm mm$^{-\delta}$ h$^{\delta}$ and $\delta = 0.16$.'*

*We have tested this approach for different moment ratios and also for median volume diameter (D0) (fig. R1). All the power-law fits result in similar number of rainfall records classified as stratiform (62 -65 %) and convective (35-38 %) and except the lowest moment ratio, most of the high intense records are classified as convective.*

*We have also tested how the classification, based on different DSD moments + median volume diameter, influence k-R power-law model when optimized separately for records stratiform and convective rainfalls (fig. R2). The power-law fits for convective rainfalls are similar for all the classifications except the one using the lowest order moments. For stratiform rainfalls, differences appear especially for rainfall intensities higher than approx. 7 mm / h. Classification using higher moment orders result in steeper k-R curves (higher R for given k). An exception is the lowest order moment ratio (M1/M0), which classifies records with high intensities into the stratiform group and thus seems to be inappropriate for such classification.*

[Figure]

*Figure R1: Power-law fits between rainfall intensity and different moment ratios quantified from DSD observed by PARSIVEL disdrometer.*

[Figure]

[Figure]

*Figure R2: k-R power law relation for vertically polarized 73.5 and 83.5 GHz plane wave fitted separately to rainfalls classified as stratiform (top) and convective (bottom). The classification uses different DSD moment ratios and median-volume diameter.*

*With respect to the last concern, we have to note that disdrometer observations which would enable classifying rainfalls based on their drop size spectra are not available for Prague dataset. We are, however, convinced that assuming drop size spectra with smaller mass-weighted diameter during Autumn period is reasonable in the climate of the Czech Republic. In addition, the uncertainty analysis, which is presented in the revised MS, supports our conclusion that the effects of DSD is significant for the longest CML and that reported improvement in QPEs is likely due to k-R model optimized for PARSIVEL records classified as stratiform.*

C9) A very minor point on comment 2 from reviewer 1. By looking at the figure and its caption I also get the wrong message that there is a drop in the water vapor attenuation around 60 GHz. The detail of k being defined differently for Oxygen and Water is not clear from the figure. If I just look at it, I see that a certain concentration of water vapor absorbs the plotted amount of energy which depicts a dip around 60GHz that shouldn't be there. One way to make the figure less prone to misinterpretation is to define the a) subplot y axes as k\_moist - k\_oxygen, this reflects the description added to the text and conveys a clear message.
Another option is to plot the attenuation only for the frequencies of interest for the paper 70-90 GHz. The rest of the spectrum is not needed since it is not utilized or even discussed as a comparison to lower frequencies CMLs. In this case, I would also avoid plotting the b) panel altogether since it is not used in the paper.

*A9: Agreed. We have modified the figure as suggested in the first paragraph. We only prefer to use k\_dryair instead of k\_oxygen, because the dry-air attenuation includes also attenuation by nitrogen.*

*We would like to keep the panel b in this figure, as dry-air attenuation influences also losses on 73.5 and 83.5 GHz, i.e. frequencies evaluated in our results.*

MORE SPECIFIC COMMENTS:

C10) Equation 1 -It may sound trivial, but I suggest to introduce the definition of Lt as tx -rx, so that the equation becomes Lt = tx - rx = Lbf + Lm + ...

*A10: Agreed (see line 66).*

C11) Line 48 - I do not know if the term resonance peak in parentheses can be considered correct. I would avoid it.

*A11: Agreed.*

C12) Line 73 and following - There is some confusion among the use of the terms loss, attenuation and specific attenuation. Perhaps it is better to clear in the introduction that in general, in the text the term attenuation refers to specific attenuation (dB/km) apart from WAA where it actually means loss (dB).

*A12: Agreed. We have clear at the end of the subsection the usage of these terms: "We further use the term loss when referring to reduction in power density of EM wave in dB, whereas the term attenuation mostly refers to specific attenuation (dB km$^{-1}$) apart from WAA where it actually means loss (dB). We, nevertheless, stick to the term WAA as it is already established in literature."*

*We also present in the revised MS results of gaseous attenuation analysis in dB km$^{-1}$. In addition, when quantifying WAA we use in the revised MS the term total rainfall-induced loss, instead of to total attenuation.*

C13) Line 112 - also polarization and orientation of the drop is relevant (if the drop is not considered spherical)

*A13: Agreed. We have modified the sentence as follows: "Attenuation caused by a single raindrop is determined by the wavelength, polarization, refractive index of water, shape parameters of the raindrop and its orientation."*

C14) Line 114 - How is D defined? From the typical usage of the pytmatrix package and Eq. 6 it only makes sense that this is the equivalent-volume diameter. Does this definition match the size measured by the PARSIVEL disdrometer?

*A14: Yes, it is equivalent volume diameter. We have specified this after the equation in the revised MS (L129). Although PARSIVEL disdrometer measures particles up to 25 mm we consider only particles with diameter from 0 to 5.5 mm to be raindrops. The definition (and calculation of extinction cross-section) is appropriate for this range of diameters.*

*We have added information on drop size range into section 3.1 of the revised manuscript where PARSIVEL (Duebendorf) dataset is presented (lines 180-181): "Particles larger than 5.5 mm are not considered to be raindrops and thus excluded from the analysis."*

C15) Line 115 - The contribution of secondary waves is commonly referred to as "multiple scattering" effects; I think the authors can use this term to simplify the discussion. Anyway, the argumentation on why multiple scattering is negligible is wrong.
Usually, the evaluation if multiple scattering has to be taken into account, is done in terms of optical depth (Battaglia 2006). Optical depth takes into account the scattering intensity through Cext, and particle number concentration Nt. Even assuming Cext to be not relevant what becomes important is not N(D) which is the drop density per size bin, but the total drop density Nt=int N(D)dD.

*A15: Thank you. We have reformulated the text to: "The total drop density in the unit volume $N_t$ ($m^{-3}$) is relatively small for natural rainfalls. Therefore, the multiple scattering effects can be neglected."*

C16) Line 119 (Eq. 5) - I believe there is an error in the formula. If Cext is cm**2 I think the coefficient at the beginning of the formula should be 0.4343 and not 4343.0 (Berne and Uijlenhoet 2007)

*A16: Indeed, it should be either 0.4343 or Cext should be in m**2. Thank you for spotting this mistake, we have corrected it.*

C17) Line 122 - Saying that R and k are equal to moments of the DSD implies that v(D) and Cext(D) are power-laws. This is a reasonable assumption for small drops for which the Stokes approximation of drag force can be assumed and the Rayleigh approximation for scattering applies, but it is not true in general (as it can be seen from Fig. 2 for drops larger than 1 mm). Also, the authors should change the term "equal" with the term "proportional to". As a matter of fact, if the two quantities would be always proportional to a moment of the DSD the relation between the two would be linear and not a power-law (that is what happens at lower frequencies).

*A17: Thank you for this explanation. We have decided to simplify the sentence and remove statement about moments, instead we refer to Olsen et al., (1978): "The relation between attenuation and rainfall intensity can be approximated by a power-law (Olsen et al., 1978):"*

C18) In Figure 2 and Eq. 9 it is introduced the concept of extinction efficiency, but this is of no use for the application of the proposed study. The important quantity that goes in Eq. 5 is Cext, not Qext. This analysis leads to another wrong statement at line 139. A single large drop contributes much more to attenuation than a single small one.
On the other hand, it is relevant for the study to analyze the relative attenuation of DSDs with small and large Dm and the same RR. In these conditions, it is true that large drops attenuate less because they produce smaller attenuation per unit mass (not per unit area). Also, larger drops fall faster, meaning for the same RR their volumetric concentration is lower.

*A18: Thank you, we are aware of that Qext alone is not sufficient to quantify contribution of large and small raindrops to total attenuation. However, our intention was to emphasize*

*different sensitivity to raindrops for frequencies used by E-band CMLs (73 and 83 GHz) and frequencies typically used in older CML networks (23 and 38 GHz), which can be demonstrated using extinction efficiency. For the sake of brevity, we have decided to remove the text explaining extinction-efficiency as well as figure 2, as they are not essential for further analyses. The statements that attenuation–rainfall model at E-band frequencies might be more sensitive to DSD are supported by the references to other studies.*

C19) Line 143. I think this is an important part and would be great to have it formulated mathematically as it has been done for RR attenuation. The combination of sections 2.3 and 2.4 models might result in better estimates of RR due to the consistent adjustment of WAA.

*A19: Thank you for this suggestion. However, formulating mathematically WAA is too complex problem and out of the scope of this paper. Instead, we have added at this place text referring to two recent studies numerically simulating WAA: "WAA can be modeled using EM full-wave simulators* (Mancini et al., 2019; Moroder et al., 2020) *solving numerically Maxwells's equations, nevertheless, such simulations are computationally demanding and require characterizing distribution of water (e.g. thin film, droplets, rivulets) and its volume on antenna radomes."*

C20) Section 3.1 and 3.2 are confusing. Wouldn't be better presenting the Duebendorf and the Prague datasets altogether? I think it makes much more sense to say what is each dataset scope, instrumentation, measuring periods, and available data instead of having these three pieces of information scattered around into two sections and three subsections.
The model to discriminate between stratiform and convective precipitation is a method well suited for the Duebendorf dataset part.
Dry and wet weather discrimination fits well the Prague dataset processing, also WAA estimation belongs to this section.

*A20: Agreed. We have merged sections 3.1 and 3.2 of the original MS.*

C21) Line 222 - Would be better to be quantitative here. What do the authors mean with mean MSL pressure? Could be an international standard atmosphere, Mid-latitude, mean MSL pressure in Prague, or others. Just reporting the number is sufficient.

*A21: Agreed. It is 1013 hPa.*

C22) Line 305 It is either "An are rainfall induced attenuations" or "A is rainfall induced attenuation"

*A22: Thank you, we have corrected this typo.*

C23) Figure 6 - The correlation coefficient (CC is of little use to evaluate the discrepancies between observed and simulated attenuation. What a high CC value tells is that if one quantity is increasing or decreasing, the other is doing the same. It does not provide information on constant biases and drifts of the two quantities.
I am not really sure of what is the information that I can get from the linear fits to the data (not discussed in the text).
In such scatterplots, it is usually more interesting to evaluate the deviations of the data from the 1:1 to analyze systematic biases and trends. The discussion up the correlation coefficients at lines 355-359 doesn't seem relevant to me, it also makes December look like the worstcase (smallest CC) while from a visual inspection it is probably the moth giving the best agreement between observed and simulated attenuation.

It seems that the theoretical attenuation is limited to values smaller than 3 dB while the observed ones go up to 6 dB, I wonder what could cause such discrepancies (uncertainties in the humidity measurements or in the evaluation of the measured gas attenuation perhaps). Comparing the distributions of attenuation values might be informative.

*A23: Indeed, the correlation does not provide information about constant biases and drifts. We have, therefore, decided to complete scatter plots (Figure 8 in the revised MS) with RMSE and mean deviation between theoretical and observed attenuation and comment on these results also in the text (lines 417-420 of the revised MS). The linear fits were removed and 1:1 lines in the color of grid lines are now shown in the scatterplots.*

*The possible causes of discrepancies between theoretical and observed attenuation are discussed in the Discussion section (lines 536-546 of the revised MS). The highest discrepancies occur during longer dry weather periods, mostly during nights and might be related to multipath propagation, or condensation of water on antennas causing WAA. Such high discrepancies are unlikely to be caused by uncertainties in humidity observations. Temperature during these events do not exceed 10° C and water vapor densities should be below 10 g m⁻³, causing at the long CML maximal theoretical gaseous attenuation not exceeding 0.45 dB/km. Moreover, humidity and temperature is measured independently at two sites and we did not observe differences which could explain such high deviations in observed attenuation.*

C24) Lines 426-430 - It is less than surprising that the results of the ITU and the stratiform-DSD-derived k-R model give similar results given the fact that for low-intensity precipitation they are almost indistinguishable (Fig. 8)

*A24: There are actually discrepancies between these two models also for low intensities, which can be seen in detailed view shown on the figure R3 below. The top panels show all four models and the bottom panels show difference between k-R model derived for stratiform rainfalls and ITU based model. The ITU model underestimates rainfalls compared to stratiform model for rainfall intensities up to about 2 mm/h. The highest underestimation compared to stratiform model (by 0.2 and 0.16 mm/h for 73.5 resp. 83.5 GHz frequencies) is reached for specific attenuation around 1.1 dB/km (Fig. R3, bottom). For stratiform rainfalls, this attenuation corresponds to rainfall intensity approx. 1.3 mm/h and 1.0 mm/h for 73.5 resp. 83.5 GHz frequency.*

[Figure]

*Figure R3: Difference between the k-R model derived for stratiform rainfall and the ITU-based k-R model.*

*The analysis estimating systematic and random errors of k-R models due to variable DSD was included into section 3.3 (lines 287-298). We have also added into Figure 8 (in the revised MS Figure 3) two insets with detail of low specific attenuations. The analysis of uncertainties provides in our view better insight than the table 4 presenting in the original MS k-R model discrepancies in terms of RMSE. We have, therefore, decided to remove this table from the revised MS. This resulted also in small changes in the Discussion section (lines 492-496 of the revised MS), where random and systematic errors are now discussed instead of RMSE.*

C25) Line 537 - It is indeed surprising that the rainfall estimation performance turns out so good. The reason is that as Fig. 8 demonstrates there is already at low intensity quite a huge spread of points derived from the DSD data which should translate in a high uncertainty of the k-R model. Moreover, even a perfect k-R model is very sensitive to the uncertainties in the estimated attenuation. The comparison of panels a and b of Fig 11 show qualitatively how the distance between antennas influences the uncertainty in the measured k, but I would like to see that quantity assessed and the influence on the retrieved RR quantified.
Especially by looking at the 70GHz panels in Fig 11b it is possible to see the points of the scatterplot lining up. This probably indicates that the uncertainty in the measurements of k are specific to each receiving station. The separation into shorter and longer CML helps the

qualitative assessment, but again the authors miss the opportunity to evaluate the uncertainty of the k measurement as a function of the distance between the stations.

*A25: We have separated deviation of CML QPEs into systematic and random component and related them to rainfall intensity (Fig. 10 of the revised MS). These deviations are then compared to expected systematic and random deviations i) due to deficits of the constant WAA model and, ii) due to DSD-related deficits of the ITU-based model. The evaluation of WAA considers explicitly length of CMLs. The results indicate, that DSD related errors are relatively small for short CMLs and for light rainfall intensities. On the other hand, the effect of DSD is significant for the long CML and can explain large part of random and systematic deviations in observed CML QPEs.*
*WAA seems to be crucial source of systematic error for shorter CMLs, which correspond to our results. The evaluation of uncertainties enabled us interpreting in detail, how CML path length influence its accuracy (lines 451 -453): "In general, sensitivity to rainfall, which is proportional to a CML path length, seems to be crucial characteristic influencing accuracy of CMLs when observing light rainfalls under 2 mm h$^{-1}$. For heavier rainfalls, other characteristic than path length influence the uncertainties of CMLs more significantly." We hypothesize that differences in WAA patterns are due to different aging of coating, nevertheless, the coating was not directly investigated in this study.*
*Furthermore, quantification of WAA (section 3.4 in the revised MS) is now extended by evaluation of systematic and random errors related to the modeling WAA as a constant offset.*

*OTHER CHANGES NOT REPORTED IN THE ANSWERS TO THE COMMENTS:*

Rev 26: Section 2.2 describing gaseous attenuation and its quantification explains briefly in the revised MS also principle behind water vapor retrieval and refers to two studies using CMLs for this purpose.

Rev 27: Coordinates of CMLs in the section 3.1 had to be removed due to concerns of mobile network operator. We, nevertheless, believe that Figure 2 provides sufficient information on the case study layout.

Rev 28: Discussion section was restructured (reordered) to reflect updated structure of Method and Result section.

Rev 29: Differences between rain gauges quantified in terms of standard deviation are reported at lines 508-510, where assumptions on rainfall spatial uniformity during quantification of WAA are discussed.

REFERENCES (ANSWERS):

Berne, A. and Uijlenhoet, R.: Path-averaged rainfall estimation using microwave links: Uncertainty due to spatial rainfall variability, Geophys. Res. Lett., 34(7), L07403, doi:10.1029/2007GL029409, 2007.

Jaffrain, J. and Berne, A.: Quantification of the Small-Scale Spatial Structure of the Raindrop Size Distribution from a Network of Disdrometers, J. Appl. Meteor. Climatol., 51(5), 941–953, doi:10.1175/JAMC-D-11-0136.1, 2012.

Mancini, A., Lebrón, R. M. and Salazar, J. L.: The Impact of a Wet S-Band Radome on Dual-Polarized Phased-Array Radar System Performance, IEEE Transactions on Antennas and Propagation, 67(1), 207–220, doi:10.1109/TAP.2018.2876733, 2019.

Meshesha, D. T., Tsunekawa, A., Tsubo, M., Haregeweyn, N. and Adgo, E.: Drop size distribution and kinetic energy load of rainfall events in the highlands of the Central Rift Valley, Ethiopia, Hydrological Sciences Journal, 59(12), 2203–2215, doi:10.1080/02626667.2013.865030, 2014.

Moroder, C., Siart, U., Chwala, C. and Kunstmann, H.: Modeling of Wet Antenna Attenuation for Precipitation Estimation From Microwave Links, IEEE Geoscience and Remote Sensing Letters, 17(3), 386–390, doi:10.1109/LGRS.2019.2922768, 2020.

Olsen, R., Rogers, D. and Hodge, D.: The aRbrelation in the calculation of rain attenuation, IEEE Transactions on Antennas and Propagation, 26(2), 318–329, doi:10.1109/TAP.1978.1141845, 1978.

Overeem, A., Leijnse, H. and Uijlenhoet, R.: Measuring urban rainfall using microwave links from commercial cellular communication networks, Water Resources Research, 47(12), doi:10.1029/2010WR010350, 2011.

Schleiss, M. and Berne, A.: Identification of Dry and Rainy Periods Using Telecommunication Microwave Links, IEEE Geoscience and Remote Sensing Letters, 7(3), 611–615, doi:10.1109/LGRS.2010.2043052, 2010.

Schleiss, M., Rieckermann, J. and Berne, A.: Quantification and Modeling of Wet-Antenna Attenuation for Commercial Microwave Links, IEEE Geoscience and Remote Sensing Letters, 10(5), 1195–1199, doi:10.1109/LGRS.2012.2236074, 2013.

SOME REFERENCES (COMMENTS):

Battaglia, A., M. O. Ajewole, and C. Simmer, 2006: Evaluation of Radar Multiple-Scattering Effects from a GPM Perspective. Part I: Model Description and Validation. J. Appl. Meteor. Climatol., 45, 1634–1647, https://doi.org/10.1175/JAM2424.1.

Berne, A. and Uijlenhoet, R.: Path-averaged rainfall estimation using microwave links: Uncertainty due to spatial rainfall
variability, Geophys. Res. Lett., 34(7), L07403, doi:10.1029/2007GL029409, 2007.

Fujiwara, M.: Raindrop-size Distribution from Individual Storms, J. Atmos. Sci., 22(5), 585–591, doi:10.1175/1520-0469(1965)022<0585:RSDFIS>2.0.CO;2, 1965

Illingworth, A. J., and T. M. Blackman, 2002: The Need to Represent Raindrop Size Spectra as Normalized Gamma Distributions for the Interpretation of Polarization Radar Observations. J. Appl. Meteor., 41, 286–297, https://doi.org/10.1175/1520-0450(2002)041<0286:TNTRRS>2.0.CO;2.

[revised manuscript text omitted]